# Differential SNARE chaperoning by Munc13-1 and Munc18-1 dictates fusion pore fate at the release site

Bhavya R. Bhaskar[1], Laxmi Yadav[1], Malavika Sriram [1], Kinjal Sanghrajka [1], Mayank Gupta[1], Boby K. V[1], Rohith K. Nellikka[1] & Debasis Das [1] ✉

The regulated release of chemical messengers is crucial for cell-to-cell communication; abnormalities in which impact coordinated human body function. During vesicular secretion, multiple SNARE complexes assemble at the release site, leading to fusion pore opening. How membrane fusion regulators act on heterogeneous SNARE populations to assemble fusion pores in a timely and synchronized manner, is unknown. Here, we demonstrate the role of SNARE chaperones Munc13-1 and Munc18-1 in rescuing individual nascent fusion pores from their diacylglycerol lipid-mediated inhibitory states. At the onset of membrane fusion, Munc13-1 clusters multiple SNARE complexes at the release site and synchronizes release events, while Munc18-1 stoichiometrically interacts with trans-SNARE complexes to enhance N- to C-terminal zippering. When both Munc proteins are present simultaneously, they differentially access dynamic trans-SNARE complexes to regulate pore properties. Overall, Munc proteins' direct action on fusion pore assembly indicates their role in controlling quantal size during vesicular secretion.

The timely and regulated release of chemical messengers from different cell types is the key to coordinated overall human body function. Abnormality in this process results in disorders like epilepsy, intellectual disabilities, autism, seizures, and neuromuscular disorders[1]. During exocytosis, when chemical messengers filled secretory vesicles fuse with the plasma membrane, ephemeral fusion pores form the first aqueous connection between the lumen of secretory vesicles and the extracellular space[2–9]. Neurotransmitters, hormones, cytokines, etc., are released through these pores to establish intercellular communication. The v-SNAREs (soluble N-ethylmaleimide–sensitive factor attachment protein receptor) present in the secretory vesicle membrane interact with the t-SNAREs present in the target membrane to catalyze the trans-SNARE complex formation[10–12]. These dynamic trans-SNARE complexes control the fusion pore properties[13]. The fusion pores dilate in a fast timescale (millisecond to second) as the trans complexes transition into the cis complexes, and the secretory vesicles fully fuse with the plasma membrane[14–16]. In neurons, synaptic vesicles (SVs) contain ~70 copies of v-SNAREs synaptobrevin2 (Syb2), whereas only 2–3 copies are sufficient to assemble functional fusion pores[17–19]. The SNARE copy numbers control the size and the kinetic properties of individual pores[13,18,20]. During rigorous presynaptic activity, how heterogeneously distributed SNARE complexes synchronize functional fusion pore assemblies is still unknown.

Although SNAREs serve as the minimal machinery, several regulatory proteins are critical in controlling membrane fusion's spatio-temporal dynamics[21]. In neurons, the docked neurotransmitter-filled SVs are primed to a release-ready state at the active zone in the absence of calcium[22,23]. After one set of vesicles fuse, the next set of SVs dock and prime at the release site in a precisely timed manner[21]. A few crucial regulatory proteins believed to operate at this stage are synaptotagmin1, complexin, Munc13-1, and Munc18-1[22,24]. Munc13-1 and Munc18-1 are known as SNARE chaperones[25,26]. Munc13-1 deletion formed ultrastructurally normal synapses[27]. However, AMPA-receptor-mediated EPSCs significantly reduced in Munc13-1-deficient glutamatergic neurons due to decreased RRP (readily releasable vesicle pool)[27].

[1]Department of Biological Sciences, Tata Institute of Fundamental Research, Mumbai 400005, India. ✉e-mail: debasis.das@tifr.res.in

Munc13-1/2 double knockout mice are perinatally lethal, and the hippocampal neurons showed a significant reduction in mEPSCs[28]. Whole-cell voltage-clamp recordings from Munc18-1 deficient neocortical neurons in slices at (embryonic) E18 showed complete loss of spontaneous synaptic activity, and this abolition of Munc18-1 led to postnatal death[29]. These cell-based studies indicated a stimulatory effect of SNARE chaperones on vesicular secretion. It is unknown whether this stimulatory action is a direct effect of these chaperones on fusion pore assembly at the release site.

Munc13-1 contains N-terminal C2A, C1 (phorbol ester Diacylglycerols or DAGs binding), and C2B ($PIP_2$ and $Ca^{2+}$ binding) domains; a central MUN (SNARE binding) domain; and a C-terminal C2C domain. It tethers SVs to the plasma membrane through its N- and C-terminal C1 and C2 domains and directly enhances SNARE assembly through the MUN domain[30,31]. DAGs and $PIP_2$ are the minor components of cell membranes and serve as the secondary messengers in diverse signaling pathways[32]. The exogenous addition of DAG enhanced both spontaneous and evoked neurotransmission from neurons[33–35]. $PIP_2$ interacts with several $Ca^{2+}$ sensors during neurotransmission[36–38]. It is, however, unknown whether these secondary messengers play a direct role in regulating fusion pore assembly during SNARE-catalyzed membrane fusion.

Munc13-1 cooperates with Munc18-1 to promote the accuracy of SNARE assembly and produce an α-SNAP/NSF-resistant complex[26,39]. The assembly pathway involves Munc18-1's binding to a self-inhibited 'closed' Syntaxin-1. Munc13-1 facilitates Syntaxin-1's opening to aid in SNARE complex formation. In the in vitro studies where Munc13-1/Munc18-1's effect on SNARE complex assembly was studied, on most occasions, SNAREs were used in the absence of a membrane environment[25,26,40]. In other reports, the SNARE-reconstituted liposomes were used to describe Munc13-1 and Munc18-1's action in vesicular secretion[30,39,41,42]. However, their direct role in affecting functional fusion pore assembly could not be traced.

Here we traced the phorbol ester-dependent localization of both Munc13-1 and Munc18-1 with the t-SNARE SNAP-25B, Syntaxin1a, and v-SNARE Syb2 in the presynaptic terminal of rat cortical neurons. To test the impact of these Munc proteins in functional fusion pore assembly, we used recently described in vitro ND-BLM reconstitution technique[13,43,44] and studied the μ-second dynamics of individual nascent fusion pores. The phorbol ester DAG binding protein Munc13-1 rescued fusion pores from their DAG-mediated inhibitory state. This action of Munc13-1 was mediated by clustering multiple SNARE complexes at the release site. These pores, however, were restricted to a low-conducting narrow conformation. Munc18-1 also rescued pores from their DAG-mediated inhibitory states by stoichiometrically interacting with SNARE complexes and stimulating N- to C-terminal zippering. These pores were significantly larger in size in comparison to those triggered open by Munc13-1. Munc18-1-triggered pores were, however, kinetically less stable at their open state than Munc13-1-triggered pores. These two Munc proteins differentially access the dynamic trans-SNARE complexes to regulate fusion pore opening. Overall, Munc13-1/Munc18-1 directly regulates chemical messengers' secretion by organizing fusion pore assembly in a timely and coordinated manner.

## Results

### Phorbol ester enhances SNAP-25B's localization with SNARE chaperones

In cells, how Munc13-1 and Munc18-1 organize release sites is unclear. To check if these Munc proteins engage differentially with the SNAREs and membrane lipids, we first performed immunocytochemistry on rat cortical neurons and studied the co-localization of Munc proteins with v- (Syb2) and t-SNAREs (Syntaxin1a and SNAP-25B) at the neurites (Figs. 1 and S1). To check what population of such co-localization occurs in the presynaptic terminal, we used an additional channel to probe for presynaptic marker synapsin (Figs. 1 and S1). We quantified

Manders Correlation Coefficients (MCCs) between individual Munc proteins and SNAREs in the presynaptic terminal (Figs. 1 and S1). The MCs M1 or M2 (M1 score for Munc proteins overlap with SNAP-25B, Syb2, and Syntaxin1a; M2 scores for overlap of individual SNAREs with these Munc proteins) indicated that Munc13-1 and Munc18-1 co-localized higher with Syb2 and Syntaxin1a, respectively. Interestingly, exogenous treatment of cultured neurons with a DAG mimic—PMA significantly altered both the M1 and M2 for SNAP-25B; an unrelated protein Hsp60 served as the control (Figs. 1b, d and S1). The MCs' alterations for Syb2 and Syntaxin1a were less significant (Fig. 1b, d). Both the Munc proteins showed a similar trend in this set of experiments (Fig. 1b, d). The phorbol ester treatment was previously shown to translocate Munc13-1 to the plasma membrane[45,46], and Munc13-1 recruits SNAP-25B at the release site[47,48]. Munc18-1 does not possess any phorbol ester binding sites, still, this Munc protein's co-localization with SNAP-25B was phorbol ester dependent. Because Munc18-1 interacts with Syntaxin and is already present at the presynaptic active zone[49–51], the enhanced co-localization of Munc18-1 with SNAP-25B was presumably due to SNAP-25B's translocation at the presynapse in a Munc13-1 dependent manner. Munc13-1 and Munc18-1 showed opposing effects in their co-localization with Syb2 as a result of exogenous PMA (Fig. 1), whereas Syntaxin1a did not show any appreciable change in co-localizing with any of the Munc proteins (Fig. 1). These results suggest a role of PMA in differentially recruiting individual SNAREs with Munc proteins at the presynapse.

Overall, this set of experiments raised the question of whether Munc13-1 and Munc18-1's presence at the presynapse has any consequence in directly affecting functional fusion pore assembly.

### DAG-mediated inhibitory action on fusion pores is rescued by Munc13-1

To test Munc13-1's direct role in membrane fusion, we employed a reconstitution-based approach and monitored glutamate release through the fusion pores as v-SNARE containing nanodisc (ND) fuses with the t-SNARE liposome (Fig. 2a). When $ND5_S$ (nanodisc containing ~5 copies of Syb2, subscript S represents small size NDs with diameter ~13 nm) was allowed to react with the glutamate-entrapped t-SNARE liposomes, glutamate release through the fusion pores was monitored using the glutamate sensor iGluSnFr (Fig. 2a). The presence of $PIP_2$ and DAG in the t-SNARE liposomes significantly reduced the percent (%) glu release (Fig. 2a). When we repeated this set of experiments in the presence of Munc13-1 or Munc18-1, keeping $PIP_2$ and DAG as part of the membrane lipid constituents, both the Munc proteins significantly enhanced % glu release (Fig. 2b). Munc13-1's $PIP_2$/DAG and SNARE binding and Munc18-1's SNARE binding properties might have contributed to that enhanced Glu release. The above Munc13-1 actions did not require $Ca^{2+}$, as all the experiments were done in the presence of BAPTA (Fig. S2a). The direct action of Munc13-1 or Munc18-1 on membrane lipids did not contribute to the enhanced content release, as none of them showed considerable SRB (sulforhodamine B) release when SRB entrapped t-SNARE liposomes were separately treated with these Munc proteins (Fig. S2b).

Next, to check if Munc13-1 directly affects the individual fusion pore properties, we employed a recently described ND-BLM assay and studied the submillisecond dynamics of individual nascent fusion pores[13,43]. $ND0_S$ (ND bearing no Syb2) failed to produce any detectable effect on BLM[13], while $ND5_S$ formed the bona fide fusion pores (Fig. 2c). Interestingly, the presence of $PIP_2$ and DAG in the BLM lipids significantly reduced the open-state stability of individual pores (Fig. 2c, Table S1). We observed fusion pores' transient opening in most of the traces in the presence of $PIP_2$/DAG (Fig. 2d). The inclusion of these lipids shifted the open dwell time distribution significantly to a shorter lifetime (Fig. 2e). In this report, open dwell time refers to the fully open state of the pores at their largest conductance states unless specifically mentioned otherwise. The rate of pore opening notably reduced in the

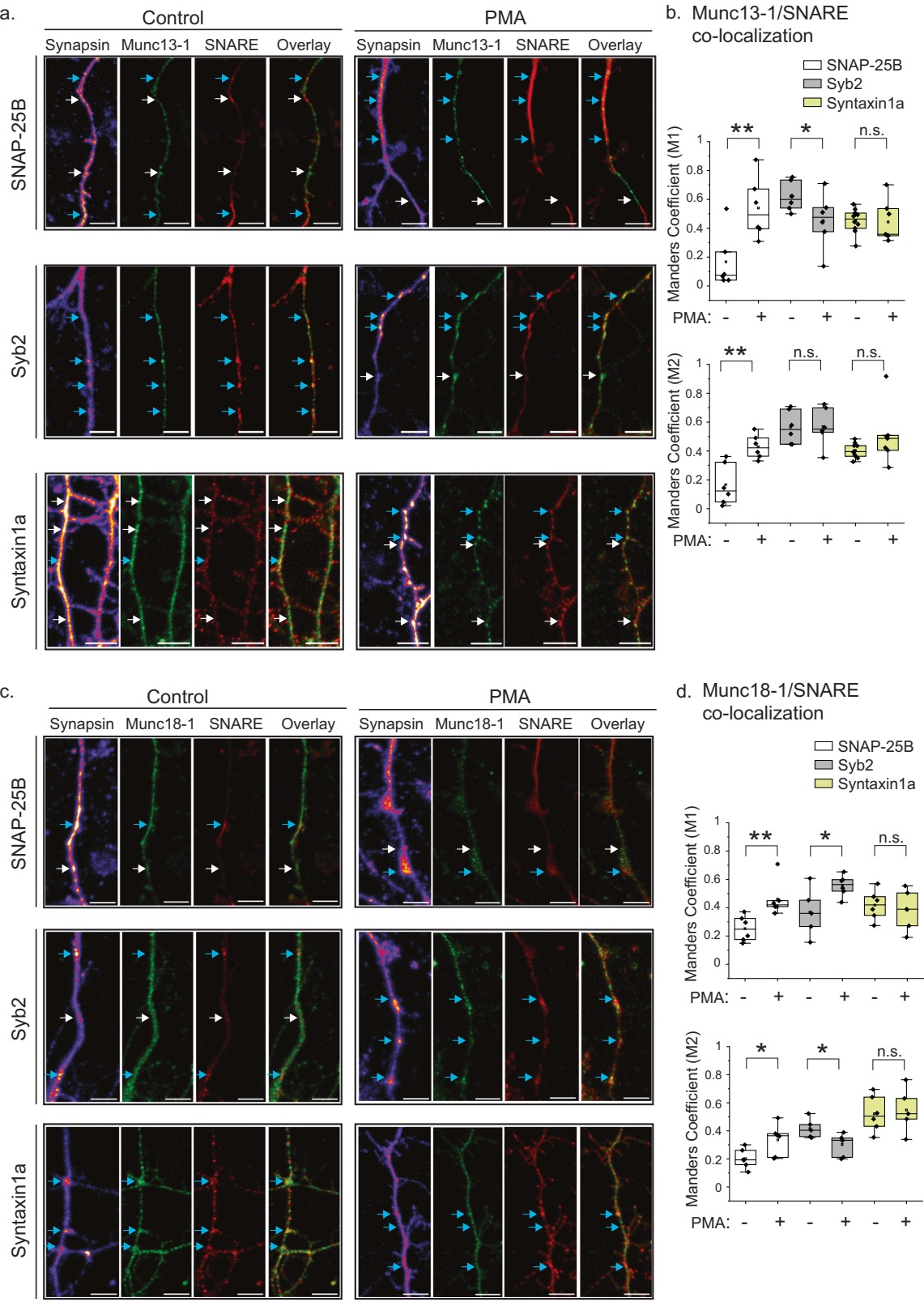

presence of these lipids, as quantified from the closed state cumulative distribution frequency (CDF) (Fig. 2f, Table S1). The inclusion of PIP$_2$ in PE, PC, and PS membranes did not alter the pore properties (Fig. S3, Table S1). The ensemble lipid mixing assay, demonstrated previously[11,18,52] also indicated that the inclusion of DAG in the PE, PC, PS containing membrane is responsible for reducing the membrane fusion efficiency (Fig. S3d).

The presence of DAG in PE, PC, and PS membranes significantly reduced the open-state stability of the pores (Fig. S4). We titrated [DAG] in the BLM and found that the low [DAG] is insufficient to yield an efficient pore inhibitory phenotype (Fig. S4). Similarly, lipid mixing percent also reduced gradually as we increased the DAG content of the acceptor membrane (Fig. S4c). Hence, DAG was responsible for this change in the fusion pore characteristics. We wondered whether this

**Fig. 1 | SNAP-25B co-localizes with SNARE chaperones in a phorbol ester-dependent manner. a** Left panel, representative confocal image of primary cortical neurons that were triple labeled for synapsin (Alexa 647-conjugated secondary antibody), SNARE proteins (Alexa 555-conjugated secondary antibody: SNAP-25B (top), Syntaxin1a (middle), Syb2 (bottom) and Munc13-1 (Alexa 488-conjugated secondary antibody). The overlay image shows regions where Munc13-1 coincided with individual SNAREs, as indicated. Blue arrows: co-localization; while arrows: no co-localization. Right panel, representative confocal images of PMA treated primary cortical neuron, under indicated conditions. **b** The box plots showing Manders correlation co-efficients M1 (Munc13-1 co-localizing with individual SNAREs as indicated) and M2 (individual SNAREs' co-localizing with Munc13-1); $n = 6$ (Munc13-1/SNAP-25B and Munc13-1/Syb2), $n = 7$ (Munc13-1/Syntaxin1a) individual neurons. $p = 0.005$ (Munc13-1/SNAP-25B; M1), 0.002 (Munc13-1/SNAP-25B; M2), 0.453 (Munc13-1/Syntaxin1a; M1), 0.113 (Munc13-1/Syntaxin1a; M2), 0.048 (Munc13-1/ Syb2; M1), 0.48 (Munc13-1/Syb2; M2). **c** The representative confocal images of primary cortical neurons that were triple labeled as described in (**a**), for Munc18-1 (Alexa 488-conjugated secondary antibody), with and without treating the neurons with PMA. **d** The box plots showing Manders correlation co-efficients M1 (Munc18-1 co-localizing with individual SNAREs as indicated) and M2 (individual SNAREs' co-localizing with Munc18-1); $n = 5$ (Munc18-1/Syntaxin1a and Munc18-1/Syb2), $n = 5$ (Munc18-1/SNAP-25B). $p = 0.004$ (Munc18-1/SNAP-25B; M1), 0.018 (Munc18-1/SNAP-25B; M2), 0.335 (Munc18-1/Syntaxin1a; M1), 0.387 (Munc18-1/Syntaxin1a; M2), 0.036 (Munc18-1/Syb2; M1), 0.016 (Munc18-1/Syb2; M2). The box plot minima and maxima represent the 25th and 75th percentiles, the lower and upper whiskers indicate the 5th and 95th percentiles, and the center line and square indicate median and mean, respectively; the Student's t-test (one-tailed) was performed to compare the two means; *$p < 0.05$, **$p < 0.01$, n.s. non-significance. Scale bar in **a** and **c** indicates 5 µm. Relevant source data are provided as a Source Data file.

pore inhibitory action was due to DAG-induced negative membrane curvature[53]. When we replaced DAG with another small head group containing lipid PA (phosphatidic acid), we observed a similar pore phenotype as observed in the presence of DAG (Fig. S5). The fusion pore inhibitory action was presumably the contribution of membrane lipid constituent that induces local negative curvature in the membrane.

Because Munc13-1 possesses a DAG binding domain[54], we added this Munc protein before the pore opening. Interestingly, in all recordings, the individual pore then yielded stable open states despite DAG's presence in the bilayer membrane (Fig. 3a, b). Munc13-1 stimulated the pore to interconvert between two different open states, a smaller conducting $O_1$ and a larger conducting dilated $O_2$ state (Fig. 3a–c). The pore's opening followed multi-exponential kinetics (Fig. 3d, Table S2). This action of Munc13-1 did not require $[Ca^{2+}]_{free}$, as the above set of experiments were performed in the presence of BAPTA. When we replaced DAG with PA, Munc13-1 could overcome the inhibitory effect of PA (Fig. S5). We compared the probabilities of the appearance of long closures under these conditions by analyzing the closed-state dwell time CDFs (Fig. S5); the mean closed-state dwell times ($\langle t_{c\text{-obs}} \rangle$) for DAG and PA were 0.54 (±0.17) s and 1.1 (±0.3) s, respectively. The probability of the appearance of long closures was higher in the presence of PA than in the presence of DAG (Fig. S5). This indicates a stabilizing function of Munc13-1/DAG interaction in pore opening and suggests that the inhibitory stimulatory mechanism is, perhaps, DAG specific. Overall, Munc13-1's interaction with the membrane lipids and SNAREs helps recover the pores from their inhibitory states.

To get additional molecular details, we performed reciprocal experiments where we kept the membrane lipid composition as—PE, PC, PS, PIP$_2$, and DAG but disrupted Munc13-1's DAG-binding ability by introducing a mutation in its DAG binding domain, yielding Munc13-1(H567K)[34]. When we repeated the above experiments in the presence of the mutated Munc13-1, a significant destabilization of the pores' open states was observed (Fig. 3e, f, h) as the mean closed-state dwell times ($\langle t_{c\text{-obs}} \rangle$) enhanced ~40 times in the case of Munc13-1(H567K) (0.38 (±0.17) s) from its wild type variant (0.014 (±0.003) s) (Fig. 3e). The mutated Munc13-1(H567K) yielded markedly larger pores in comparison to wild type (WT) Munc13-1, as evident from the conductance comparison (Fig. 3g). The rate of opening reduced drastically in the case of Munc13-1(H567K) (Fig. 3h, Table S2). We performed a co-flotation assay to assess Munc13-1(H567K)'s ability to interact with membrane lipids (Fig. S6). Interestingly, the mutated Munc13-1 showed considerable membrane interaction when compared to the WT (Fig. S6), presumably through its interaction with the other negatively charged membrane lipids. The Munc13-1(H567K), although interacted with the other membrane lipids, the absence of its efficient interaction with DAG prevented stable fusion pore opening. These results further confirmed the role of DAG in eliciting Munc13-1's pore modulatory action.

Because Munc13-1 interacts with both the membrane lipids and SNAREs, next, we explored whether its SNARE binding ability also contributes to the fusion pore regulation. We mutated the MUN domain of Munc13-1 (residues F1234, K1236), yielding Munc13-1(FKAA), which disrupts its interaction with the Syntaxin1a N-terminus[55,56]. Munc13-1(FKAA) interacts equally well with the membrane lipids as WT Munc13-1 (Fig. S6) but did not yield two open states of the fusion pores, as observed in the case of WT (Fig. 3i, j). The rate of opening of these pores is also reduced in comparison to the pores triggered open by WT Munc13-1 (Fig. 3k). Hence, Munc13-1 can access SNAREs in the dynamic trans-SNARE complex as fusion pores functionally assemble at the membrane fusion site.

The above actions of Munc13-1 were not dependent on $Ca^{2+}$, as all the experiments were performed in the presence of BAPTA. Because this Munc protein possesses $Ca^{2+}$ binding domain, we tested the effect of $Ca^{2+}$ on Munc13-1 triggered open pores. Interestingly, a 500 µM $Ca^{2+}$ significantly enhanced the pore conductance, while those pores showed kinetic destabilization in their open states, as indicated by the closed-state CDF analysis (Fig. S7). The same amount of $Ca^{2+}$ in the absence of regulatory protein did not alter pore properties, observed previously[43]. This indicates that the secondary messengers, like, $Ca^{2+}$ and DAG, can affect the function of Munc13-1 in regulating fusion pore properties.

Overall, this set of experiments indicates that Munc13-1's interactions with negative curvature-inducing membrane lipids and Syntaxin1a contribute to the stable fusion pore opening with multiple conductance states.

## Munc18-1's action on fusion pores is distinct from Munc13-1

To better understand the DAG-mediated inhibitory action on fusion pore properties, we inspected whether the other SNARE chaperone Munc18-1 can rescue pores from their inhibitory state. Munc18-1 stimulated stable opening of individual pores to a single conductance state, despite not possessing any lipid binding domain (Fig. 4a, b). Munc18-1 stimulated pores opened to a significantly larger conductance state than Munc13-1 (Fig. 4c), indicating their role in modulating the fusion pore structure. The in-depth kinetic analysis revealed a multi-exponential fusion pore closure driven by Munc18-1, yielding slow and fast kinetic components for pore closure (Fig. S8a and Table S3). The rate of pore closure increased as a function of [Munc18-1] (Fig. S8b). To better understand Munc18-1 mediated pore closure, we plotted the amplitudes of slow and fast kinetic components after fitting the open dwell times with multiexponential function (Fig. 4d, S8a, Table S3). Interestingly, the fast components increased, and the slow components decreased with the increase in [Munc18-1] (Fig. 4d). This set of experiments indicated that Munc18-1 stoichiometrically interacts with the multiple trans-SNARE complexes in fusion pore assembly. To further confirm whether this action of Munc18-1 was indeed through its interaction with SNAREs in the dynamic trans-SNARE complex, we mutated Munc18-1 residue that alters its

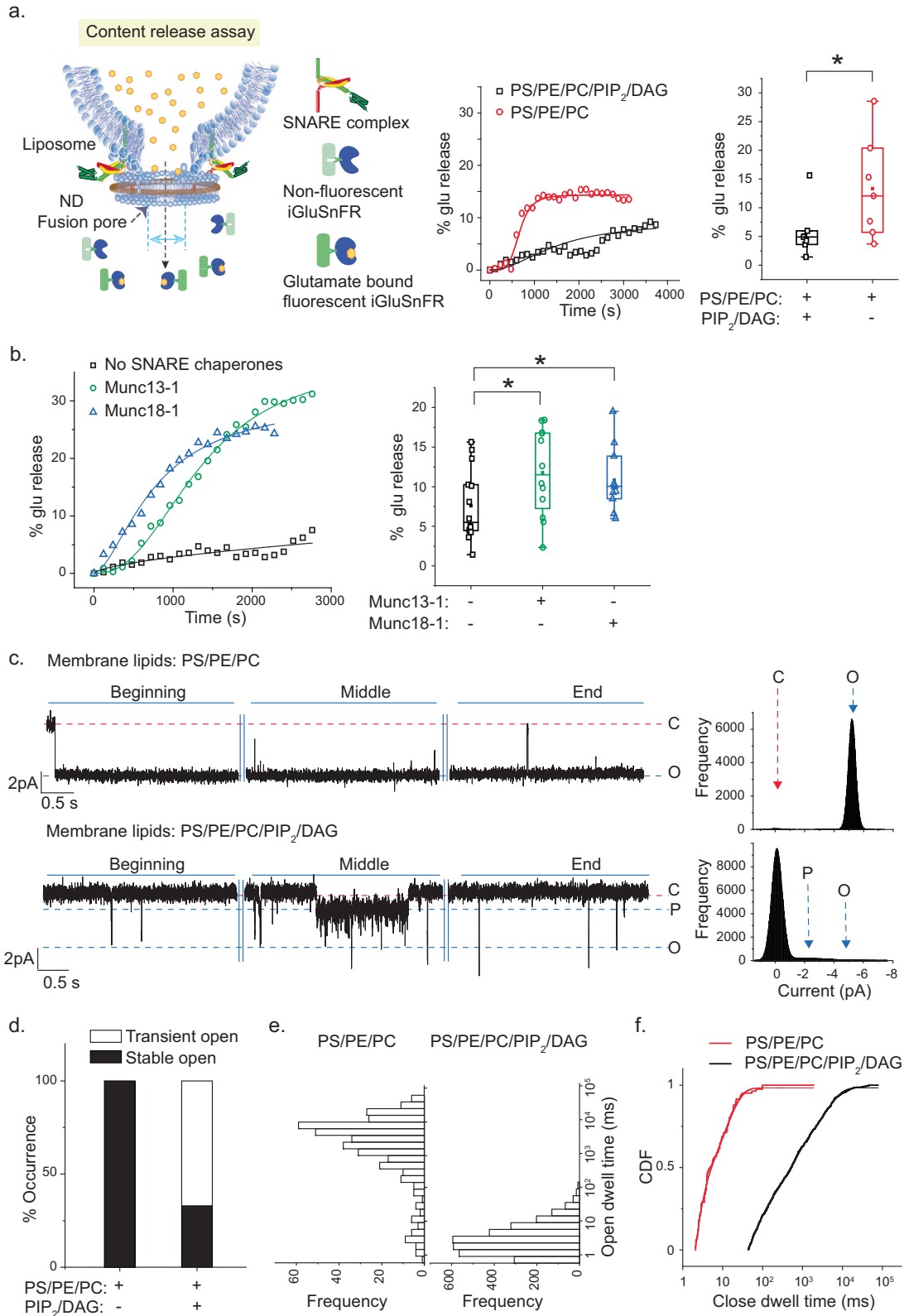

interaction with Syb2, yielding Munc18-1(D326K)[57] (Fig. 4e–g). We analyzed the close state dwell time CDFs for WT and mutant (Fig. 4g, Table S2). This mutation significantly increased the rate of pore opening compared to WT. Hence, Munc18-1 accessed dynamic trans-SNARE complexes to regulate the fusion pore properties.

Next, we probed the role of membrane lipids in Munc18-1's pore modulatory action. When we omitted $PIP_2$/DAG from the BLM membrane lipids, Munc18-1 kinetically enhanced the rate of pore opening to its single conductance state (Fig. S8c, Table S2), indicating that $PIP_2$/DAG reduced Munc18-1's efficiency of pore modulation. However, in comparison, this kinetic barrier was surmounted by the $PIP_2$/DAG interacting protein Munc13-1.

The above results indicated that Munc13-1 and Munc18-1 differentially regulate the pore properties.

**Fig. 2 | SNARE chaperones overcome the inhibitory effect of phorbol ester DAG during membrane fusion. a** Left, illustration for glutamate (glu) release assay. Middle, representative time course of glutamate released through the fusion pores in the absence (red) and presence (black) of PIP$_2$/DAG lipids in t-SNARE liposomes. Right, box plots showing pooled results to indicate half maximum percent (%) of glu released for indicated conditions; $p = 0.039$. **b** Left, representative time course as shown in (**a**), in the absence and presence of SNARE chaperones (Munc13-1 (green) and Munc18-1(blue)) keeping membrane lipid composition for t-SNARE liposomes (PS/PE/PC/PIP$_2$/DAG) fixed. Middle, box plots showing pooled results to indicate half maximum percent (%) of glu released for the indicated conditions; $p = 0.024$ (Munc13-1), 0.035 (Munc18-1). The box plot minima and maxima represent the 25th and 75th percentiles, the lower and upper whiskers indicate the 5th and 95th percentiles, and the center line and square indicate median and mean, respectively; $n = 7$ (for **a**) and $n = 10$ (for **b**) independent trials under each of the conditions mentioned; three independent sets of NDs were used. Student's $t$-test (one-tailed) was performed to compare the two means; *$p < 0.05$, n.s. non-significance. **c** Left, representative traces of single ND5$_S$ pores formed using the ND-BLM system. Three epochs (beginning, middle, and end) of the trace are shown. The membrane lipid composition of the BLMs is highlighted above the trace. Full closed (C) (red), partially open (P) (blue), and full open (O) (blue) states of the individual pores are indicated with respective currents. Right, current histograms of pores in the absence (top) and presence (bottom) of PIP$_2$/DAG in BLM lipids. **d** Percent (%) occurrence of trials for which pores were stably or transiently open to their full open conductance states for indicated experimental conditions. **e** Open dwell time histograms of pores in the absence and presence of PIP$_2$/DAG in BLM lipids are shown for each experimental condition. **f** Cumulative distribution functions (CDF) of closed dwell times for each experimental condition. $n = 3$ independent BLMs; three sets of NDs were used for each of the conditions. Relevant source data are provided in the Source Data file.

## Munc13-1 and Munc18-1 differentially organize trans-SNARE assembly

We checked the ability of Munc proteins to alter trans-SNARE assembly. When ND5$_S$ or ND25$_L$ [~5 or ~25 copies of v-SNAREs were reconstituted in 13 (ND5$_S$) or 30 (ND25$_L$) nm NDs, respectively] was allowed to react with t-SNARE liposomes in the presence of increasing [Munc13-1], a gradual increase in the SNARE complex formation was observed in the immunoblots (Fig. 5a, left panel and S9a). In parallel, we resolved the SNARE complexes in SDS-PAGE after heating at ~100 °C, yielding a similar amount of individual monomers for SNAP-25B, Syb2, and Syntaxin1a (Fig. S9b), indicating that the above increase of SNARE complexes was indeed an effect of Munc13-1. Further analysis yielded an EC$_{50}$ of 0.07 ($\pm$0.02) μM (for ND5$_S$), 0.3 ($\pm$0.1) μM (for ND25$_L$) & Hill coefficients of 4.5 ($\pm$0.9) (for ND5$_S$), 4.4 ($\pm$0.5) (for ND25$_L$) (Fig. 5a, left panel and S9c). These results indicated that Munc13-1 cooperatively increases trans-SNARE complex formation as v-SNARE NDs approach the t-SNARE liposomes. The above cooperativity can originate due to individual SNARE complexes' cooperative zippering from N- to C-terminus or due to the cooperative clustering of several SNARE complexes at the release site (Fig. S9d). To delineate further, we varied Syb2 copy numbers in the NDs, yielding ND3$_S$[13]. The ND3$_S$ and ND5$_S$ were separately allowed to react with the t-SNARE liposomes in the presence of increasing [Munc13-1]. Interestingly, the Hill coefficients derived increased significantly with an increase in Syb2 copy number (Fig. 5b, top panel and S9e). These results indicated that the cooperativity during SNARE complex formation was indeed originating from Munc13-1 mediated cooperative clustering of multiple SNARE complexes at the release site. That, however, does not exclude the possibility of Munc13-1 mediated cooperative zippering of SNARE complexes from N- to C-terminus[26].

Next, we inspected the effect of increasing [Munc18-1] in SNARE complex formation, which also showed cooperative enhancement of SNARE complex formation, quantified from the immunoblots (Fig. 5a, right panel). The Hill coefficients derived, however, did not increase with an increase in the Syb2 copy number (Fig. 5b, bottom panel and S9e). This indicates that the cooperativity, in this case, was originating due to individual SNARE complexes' cooperative zippering from N- to C-terminus, also reported previously[26,58,59].

We inspected whether the action of Munc13-1 and Munc18-1 on SNARE complexes was membrane lipid dependent. We omitted PIP$_2$ and DAG from the lipid mixture and allowed ND5$_S$ to react with t-SNARE liposomes in the presence of increasing [Munc13-1] or [Munc18-1]. The derived Hill coefficients were significantly reduced in the case of Munc13-1 but remained the same for Munc18-1 (Figs. 5c and S9f). Munc13-1's effect on SNARE complex clustering was membrane lipid dependent (Fig. 5c, left panel and S8f).

Next, we wondered if Munc proteins' cooperative action provides any kinetic benefit to the SNARE complex organization. To test that, we first engineered the cytosolic domains of Syb2 and Syntaxin1a to retain only one cysteine (C) at the SNARE motif's N- or C- termini. These mutations had no effect in yielding SNARE complexes (Fig. S10a). We labeled the Syb2 and the Syntaxin1a variants with the fluorescence dyes cy3 and cy5, respectively. These cy3 labeled Syb2 were reconstituted in NDs, yielding ND5$_S^{cy3N}$ and ND5$_S^{cy3C}$; while cy5 labeled Syntaxin1a was reconstituted as a part of t-SNAREs in liposomes, yielding t-SNARE liposome$^{cy5N}$ and t-SNARE liposome$^{cy5C}$ (Fig. S10b). We observed the kinetics of SNARE complex organization as ND5$_S^{cy3N}$ was allowed to react with t-SNARE liposome$^{cy5N}$ in the presence of Munc13-1 or Munc18-1, and the cy5 emission was traced as a function of time (after exiting at cy3 excitation wavelength). None of the Munc proteins altered the rate of SNARE complex zippering at the N-terminus, as evident from the quantified $T_{1/2}$ (half-time) (Fig. 5d). In contrast, when ND5$_S^{cy3C}$ and t-SNARE liposome$^{cy5C}$ were allowed to react, Munc proteins significantly altered the rate of SNARE complex zippering at the C-terminus (Fig. 5e). This, however, does not exclude the possibility that the observed results were due to enhanced SNARE pairing by Munc proteins.

This set of experiments indicated that Munc13-1 and Munc18-1 differentially organize trans-SNARE assembly (Fig. 5f).

## Munc13-1 and Munc18-1 synchronize vesicular secretion at the release site

To probe the importance of SNARE complex clustering in cargo release, we allowed ND3$_S$, ND5$_S$, and ND7$_S$ to react with the glutamate-entrapped t-SNARE liposomes in a separate set of experiments. DAG and PIP$_2$ were included in the t-SNARE liposomes. The % glu release increased linearly with a slope of 0.12 ($\pm$0.001) as the v-SNARE copy numbers increased from three to seven (Figs. 5g and S10c). This increase resulted from the increased number of trans-SNARE complexes recruited at the membrane fusion site, reported earlier[13]. When we repeated these experiments in the presence of Munc13-1, the v-SNARE copy number dependence of % glu release was abrogated, as the above slope reduced significantly to 0.02 ($\pm$0.01) (Figs. 5g and S10c). The presence of Munc18-1 also reduced the slope to 0.07 ($\pm$0.06) (Figs. 5g and S10c). Similarly, the kinetics of Glu release also became independent of the v-SNARE copy number in the presence of both the Munc proteins (Fig. 5g).

We speculated whether Munc13-1 mediated cooperative clustering of trans-SNARE complexes assists in fusion pore opening. We performed a competition experiment between Munc13-1 and the cytoplasmic domain of Syb2 (cd-Syb2)[13,43], each of which independently alters open state stability of individual pores by accessing the dynamic trans-SNARE complex[13,43]. In the ND-BLM assay, after Munc13-1 triggered the ND5$_S$ pores to yield stable open states (membrane lipid composition—PE, PC, PS, PIP$_2$, and DAG), a ~20 μM cd-Syb2 was unable to show any alteration in the pore properties (Fig. S10). When we omitted Munc13-1 from this reaction (membrane lipid composition—PE, PC, PS), the same [cd-Syb2] significantly destabilized the pore open

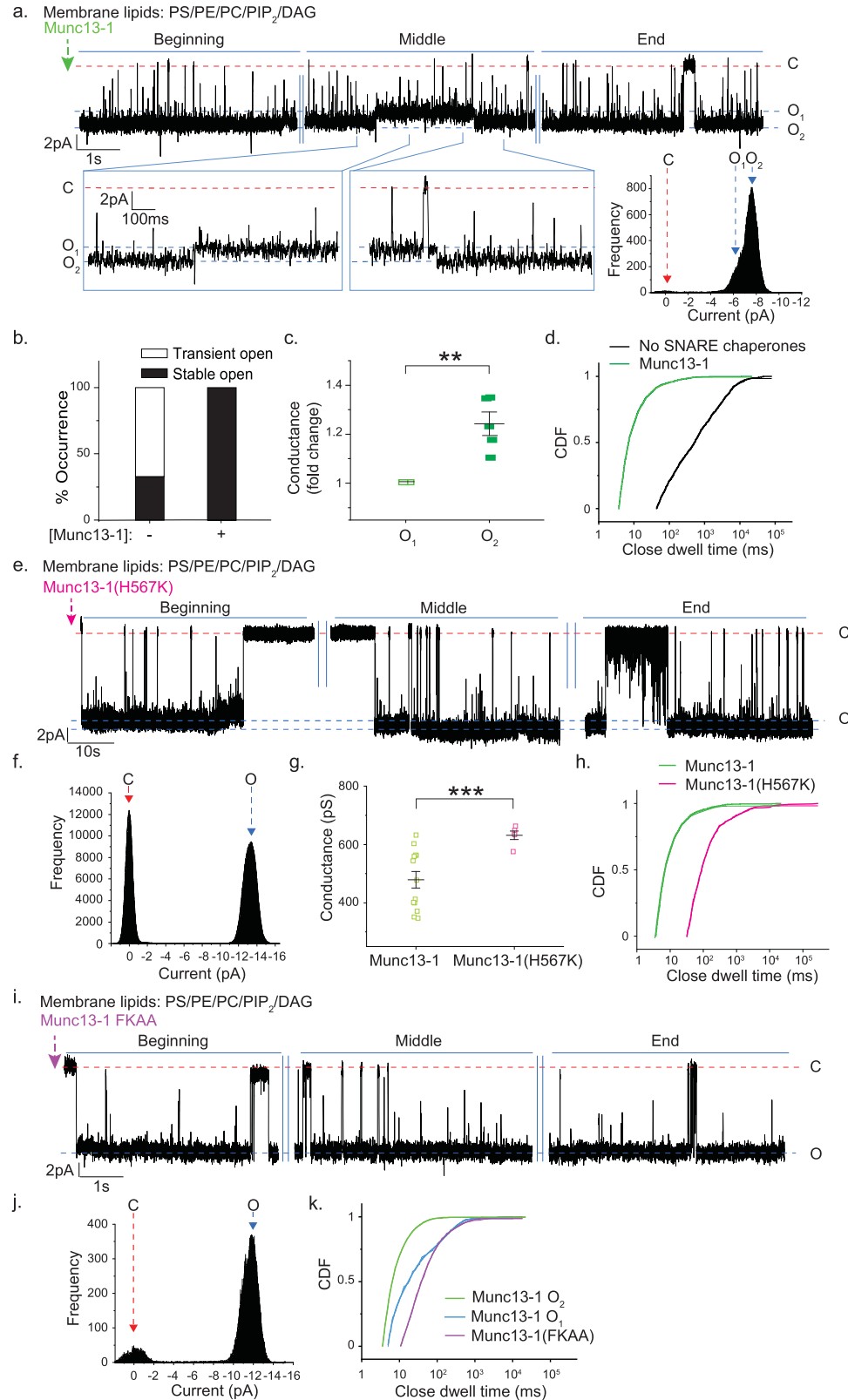

states and closed the individual pores (Fig. S11), also shown previously[13,43]. Munc13-1's presence organizes more trans-SNARE complexes at the release site. Hence the same [cd-Syb2] was insufficient to titrate out all the trans-SNARE complexes. We reduced the Syb2 copy number in NDs, yielding ND3$_S$, and observed that ~20 μM cd-Syb2 was sufficient to yield long pore closures in the presence of Munc13-1 (membrane lipid composition—PE, PC, PS, PIP$_2$ and DAG)

(Fig. S11). This set of experiments showed the importance of Munc13-1 mediated clustering of SNARE complexes in yielding stable open pores. To get additional insight, in a separate set of experiments, we mutated Munc13-1 residue 1358, yielding Munc13-1(D1358K), which perturbs its oligomeric organization as well as its interaction with Syb2[60–62]. This mutated Munc13-1 was unable to trigger stable opening of fusion pores from the DAG-induced inhibitory state (Fig. S12),

**Fig. 3 | Munc13-1 directly alters nascent fusion pore properties. a** Top, representative trace shows the effect of Munc13-1 on an ND5$_S$ pore. The membrane lipid composition of the BLMs is mentioned. Closed (C) and multiple open states (O$_1$/O$_2$) are indicated with the respective currents. Bottom inset, current and time axes were magnified from the trace shown above. Bottom right, current histogram of the pore; closed (C) and open (O$_1$/O$_2$) states are indicated by red and blue arrows respectively. **b** Percent (%) occurrence of trials for which pores were stably open or transiently open to their full open conductance states for indicated experimental conditions are shown. **c** Scatter plots showing fold change in conductance of two open states of the pore, as observed in (**a**); $n = 5$ independent BLM recordings; $p = 0.004$. **d** CDF of closed state dwell times for indicated experimental conditions. **e** Representative trace of single ND5$_S$ pore triggered open by Munc13-1(H567K), as described in (**a**). Closed (C) and open states (O) are indicated with the respective currents. **f** Current histogram of the trace shown in (**e**), closed (C) and open (O) states are indicated by red and blue arrows, respectively. **g** Scatter plots show pooled data for pore conductance under indicated experimental conditions; $n = 13$ (for Munc13-1) and $n = 5$ (for Munc13-1(H567K)) single pore currents were analyzed; $p = 0.0002$. **h** CDF of closed state dwell times for indicated experimental conditions. **i** Representative trace of single ND5$_S$ pore triggered open by Munc13-1(FKAA) as described in (**a**). **j** Current histogram of the trace shown in (**i**), closed (C), and open (O) states are indicated by red and blue arrows, respectively. **k** CDF of closed state dwell times for indicated experimental conditions. $n = 9$ independent BLMs for (**a**–**d**); three sets of NDs were used. $n = 3$ independent BLMs for (**e**–**k**); two sets of NDs were used. Data are presented as mean ± SEM; Student's $t$-test (one-tailed (for **c**) and two-tailed (for **g**)) was performed to compare the two means; **$p < 0.01$, ***$p < 0.001$, n.s. non-significance. Relevant source data are provided as a Source Data file.

indicating the importance of Munc13-1's oligomerization in yielding functional fusion pore assembly[60,61].

These results suggest that both the proteins offset the degree of heterogeneity in cargo release that arose due to a varied number of SNARE complexes engaged at the release site (Fig. 5f). Hence, both the Munc proteins assist in synchronizing vesicular secretion during membrane fusion.

### Sequential action of SNARE chaperones on fusion pores differentially augments vesicular secretion

Because both the Munc proteins provided kinetic benefit in synchronized vesicular secretion, we wondered how their sequential action dictates fusion pores' fate at the release site. We first used Munc18-1 to trigger the fusion pores to stably open from the DAG-mediated inhibitory state (Fig. 6a, b). Pores showed stable open as well as closed state transitions, with the appearance of intermittent long closures (Fig. 6a, b). The addition of Munc13-1 to that pore significantly stabilized the open states (Fig. 6a, b). This Munc13-1 addition, however, did not alter the pore conductance (Fig. 6c), indicating that the pore diameter was unaffected. The rate of pore opening was enhanced significantly upon Munc13-1 addition, as evident from the closed-state CDF (Fig. 6d).

Then we performed the reciprocal experiments where we first allowed the pores to stably open from the DAG-mediated inhibitory state using Munc13-1 (Fig. 6e–h). The addition of Munc18-1 to those pores neither altered the open state stability (Fig. 6e–h) nor altered conductance (Fig. 6g), indicating that Munc18-1 had no effect in altering pore properties under this condition. The rate of pore opening also remained unaltered, as evident from the closed state CDF (Fig. 6h).

Next, we added a mixture of Munc13-1 and Munc18-1 at the beginning of pore formation, which yielded pores that interconvert between two open states (O$_1$ and O$_2$) (Fig. 6i, j). The conductance comparison indicated that these pores had dimeters comparable to that of pores triggered open by Munc18-1 alone (Fig. 6k). The closed state CDF analysis showed that the rate of opening of these pores is more compared to either Munc13-1 alone or Munc18-1 alone triggered pores (Fig. 6l).

Overall, this set of experiments suggests that the Munc13-1 and Munc18-1 differently access the fusion pores to modulate their properties. When both the chaperones are simultaneously available at the release site, it ensures maximum cargo release by both kinetically stabilizing the pore open states as well as structurally yielding pores with large diameters.

### Discussion

Spatiotemporal dynamics of membrane fusion control the amount of chemical messengers released from cells. The SNAREs serve as the minimal machinery for fusion[10,63] and catalyze the fusion pore formation, through which chemical messengers escape in a timely and precise manner. The regulated fusion pore assembly is the key to

synchronizing overall human body function, which gets disrupted under disease conditions[1]. Several regulatory proteins either act on SNAREs and/or on membrane lipids to directly control membrane fusion[21]. It is unclear how these regulators organize the SNAREs and regulate the functional fusion pore assembly. Here we have investigated the role of SNARE chaperones Munc13-1 and Munc18-1 in arranging fusion pores at the release site.

The membrane association of Munc13-1 was reported to be DAG-dependent[54]. Here we first investigated phorbol ester's effect in the localization of SNARE chaperones with the individual SNAREs in rat cortical neurons. Our results indicated that SNAP-25B is recruited at the presynapse by the phorbol ester binding protein Munc13-1 (Fig. 7). This event presumably brings SNAP-25B adjacent to Munc18-1, which is already present in the Syntaxin clusters at the release site (Fig. 7)[47–49,64]. Hence, the exogenous PMA treatment enhanced the localization of Munc18-1 with SNAP-25B, although Munc18-1 does not contain any phorbol ester binding domain. The PMA treatment in our experiments might also activate PKC (Protein Kinase C) pathways[65], which has been shown to phosphorylate Munc18-1[66,67], which may not affect synaptic transmission[68]. This set of experiments, however, did not provide direct evidence of how these Munc proteins organize fusion pores at the release site during regulated secretion.

To understand the role of Munc13-1 and Munc18-1 in fusion pore assembly, we traced glu release through the pores in the reconstituted system and serendipitously discovered the inhibitory role of phorbol ester DAG in glutamate efflux. The presence of DAG as a membrane lipid constituent significantly reduced the rate of individual fusion pores' opening, yielding long closures (Fig. 2). Hence, the localized presence of DAG in the membrane can serve as the membrane fusion clamp. Because we could functionally replace DAG with PA (Fig. S5), presumably, the inhibitory action of DAG could be a consequence of its ability to alter membrane curvature. Interestingly, SNARE chaperones Munc13-1 and Munc18-1 assisted in overcoming this inhibitory effect and stimulated the glu release during membrane fusion.

Munc13-1 affects both the size and kinetic properties of individual nascent fusion pores. It stimulates pore opening through its specific interaction with DAG. Munc13-1 translocates to the plasma membrane via its interaction with DAG, which is a crucial step to stably open the fusion pores but restricts the pores in structurally small conformation. This action of Munc13-1 is dependent on its ability to cooperatively cluster multiple SNARE complexes at the release site, which we observed is also DAG dependent. This SNARE chaperone altered the required number of cd-Syb2 molecules to titrate out the dynamic trans-SNARE complexes. Because an increasing number of SNARE complexes at the release site increases the probability of cargo release through the fusion pores[13,18,20], it was unknown how this heterogeneous release gets attuned during sustained release activities. Munc13-1 synchronizes the release probabilities by cooperatively organizing multiple SNARE complexes at the release site. The Munc13-1 nano-assembles were observed at the active zone which in turn can

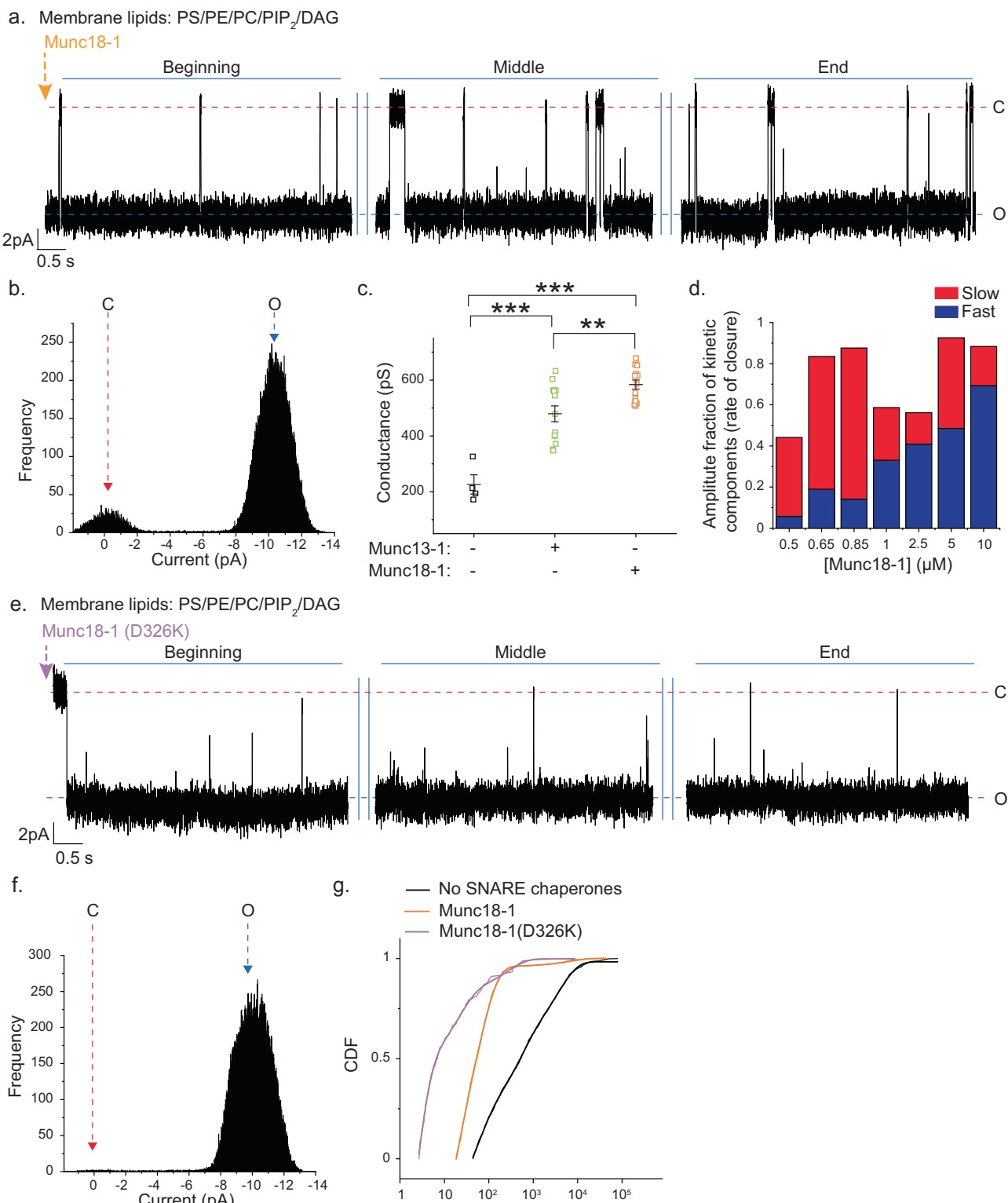

control the number of release sites[69–71]. The direct pore modulatory action of Munc13-1 is also dependent on its self-oligomerization[60,72] at the release site and on its interaction with the N-terminal regulatory domain of Syntaxin1a[56].

The other SNARE chaperone Munc18-1 stoichiometrically interacts with trans-SNARE complexes and stimulates individual SNARE complexes' zippering from N- to C-terminus. The Munc18-1 protein-triggered pores showed larger conductance than the Munc13-1-triggered pores. However, those pores were kinetically less stable

than Munc13-1-triggered pores. The Munc18-1 mediated long closures presumably appeared due to a dynamic open/close transition of SNAREs within the fusion pore assembly. This Munc protein has been shown to disassemble t-SNARE heterodimers and stabilize the closed conformation of Syntaxin1a[41,73]. This could lead to transient SNAP-25B displacement from the SNARE complexes, hence resulting in long pore closures (Fig. 7). The Munc18-1 mutation (D326K) that showed reduced SNAP-25B displacement[73] significantly enhanced the open state stability of pores, with the appearance of multiple open states.

**Fig. 4 | Munc18-1 differentially alters fusion pore properties compared to Munc13-1. a** Representative trace for ND5$_S$ pore triggered open by -1 μM Munc18-1. Three epochs (beginning, middle and end) of the trace are shown, membrane composition mentioned; closed (C) and open (O) states are indicated with the respective currents. **b** Current histogram of the trace shown in (**a**), closed (C), and open (O) states are marked by red and blue arrows, respectively. **c** Scatter plot shows a comparison of pore conductances between no SNARE chaperones (black), Munc13-1 (-0.3 μM) (green), and Munc18-1 (-1 μM) (orange) triggered pores; $n = 4$ (for no SNARE chaperones), $n = 13$ (for Munc13-1), $n = 15$ (for Munc18-1) single pores were analyzed; $p = 0.0006$ (No SNARE chaperones/Munc13-1), 0.0004 (No SNARE chaperones/Munc18-1), 0.005 (Munc13-1/Munc18-1). **d** Stacked column plots show amplitudes of fast and slow kinetic components (as indicated) for pore closures, derived from open dwell time CDFs exponential fit. These individual pores were triggered open by different [Munc18-1], as shown. **e** Representative trace of single ND5$_S$ pore triggered open by Munc18-1(D326K), BLM lipids used is mentioned. Three epochs (beginning, middle, and end) of the trace are shown, closed (C) and open states (O) are indicated with the respective currents. **f** Current histogram of the trace shown in (**e**), closed (C) and open (O) states are indicated by red and blue arrows, respectively. **g** Closed state dwell time CDFs of pores formed under indicated experimental conditions. $n = 7$ (for Munc18-1), $n = 3$ (for no SNARE chaperones and Munc18-1(D326K) independent BLMs; at least three sets of NDs were used for each experimental condition. Data are presented as mean ± SEM; Student's $t$-test (two-tailed) was performed to compare the two means; **$p < 0.01$, ***$p < 0.001$. Relevant source data are provided as a Source Data file.

The amount of cargo released during membrane fusion is dependent on both the size and the duration of the fusion pore opening, which has ramifications in differentially activating the downstream receptors. The above results indicated that Munc13-1 and Munc18-1 can differentially control the release probabilities by differently altering the fusion pore assemblies at the release sites. When both the SNARE chaperones engage at the membrane fusion site, fusion pores open in a structurally large conformation and in a kinetically stable open state. We also observed enhanced C-terminal assembly of Syntaxin1a and Syb2 in the presence of these Munc proteins, indicating their contribution in kinetically overcoming the energetic barrier of membrane fusion. The co-existence of these Munc proteins together allows stable association of SNAP-25B with the other two SNAREs to hold the pores open at the release sites. How these properties dictate the cellular physiology under normal and pathological conditions will be the subject of future studies.

## Methods
Our research complies with all relevant ethical regulations under the institutional biosafety committee (IBSC, TIFR-Mumbai) and Committee for Supervision and Care of Experimental Animals (CPCSEA), Government of India.

### Animals
Sprague-Dawley rats (Rattus were bred in the Tata Institute of Fundamental Research (TIFR) animal facility on a 12 h light–dark cycle with ad libitum access to food and water. The experimental procedures described below were in accordance with the guidelines of the Committee for Supervision and Care of Experimental Animals (CPCSEA), Government of India, and were approved by the TIFR Institutional Animal Ethics Committee (TIFR/IAEC/2020-4). For neuronal cell culturing, 8–12 postnatal pups (P0–P4) were used for each round of culturing, and sex was not considered while selecting the pups.

### Materials
1,2-dioleoyl-sn-glycero-3-phospho-L-serine (sodium salt) (PS), 1-palmitoyl-2-oleoyl-sn-glycero-3-phospho-(1′-rac-glycerol) (sodium salt) (PG), 1,2-dioleoyl-sn-glycero-3-phosphoethanolamine (PE), 1-Palmitoyl-2-oleoyl-sn-glycero-3-phosphocholine (PC), brain derived Phosphatidylinositol 4,5-bisphosphate or PtdIns(4,5)P2 (PIP$_2$) and 1-palmitoyl-2-oleoyl-sn-glycerol (DAG) were purchased from Avanti polar lipids; DDM (n-dodecyl β-D-maltoside) and OG (n-octyl glucoside) were obtained from Gold Biotechnology; IPTG (Isopropyl β- d-1-thiogalactopyranoside), Triton X 100, HEPES, KCl, Imidazole, β-mercaptoethanol, Thrombin, Tween-20, L-Glutamic acid potassium salt monohydrate, N,N-Dimethylformamide anhydrous, Tris(2-carboxyethyl)phosphine hydrochloride (TCEP), Ampicillin sodium salt, Bovine Serum Albumin (BSA), DNase I, Calcium Chloride, Phorbol 12-myristate 13-acetate (PMA) and 1,2-bis(o-Aminophenoxy)ethane-N,N,N′,N′-tetraacetic Acid (BAPTA), 3-sn-Phosphatidic acid sodium salt from egg yolk lecithin (kind gift from Prof. Roop Malik, IIT Bombay) were from Sigma Aldrich; Glutathione Sepharose 4 B, Ni-Sepharose 6 Fast Flow and Benzamidine Sepharose 4 Fast Flow (low sub) were obtained from GE Healthcare; SnakeSkin Dialysis tubing and Fluo-4 pentapotassium salt (kind gift from Prof. Sudipta Maiti's lab in DCS, TIFR-Mumbai) were from Thermo Fischer Sc.; and Bio-beads SM2 were from BIO-RAD. Non-fat milk powder and SignalFireTM ECL Reagent (Cell Signaling Technology), Glycerol (Honeywell), Accudenz (Axell), Protease Inhibitor Cocktail (Roche). Kanamycin Sulfate, 1× HBSS (no calcium, no magnesium, no phenol red), 1× TrypLETM Express Enzyme (no phenol red), Poly-D-Lysine (PDL), DPBS (Calcium, Magnesium), NeurobasalTM Medium, L-Glutamine, B-27TM Supplement (50×, Serum-free), Penicillin-Streptomycin (5000U/ml), and Paraformaldehyde (PFA) were obtained from Thermo Fischer Sc. (GibcoTM), NuncTM Cell-Culture Treated Multidishes (12 well culture plates), CyDye Maleimides (Amersham/GE Healthcare), 364-well flat black plate (Costar). The antibodies used in this study are listed in Table S4.

### Cell culture and PMA treatment
Briefly, P0–P4 Sprague-Dawley littermates were sacrificed, and cortices were microdissected and kept in 1× HBSS until trypsinization. Cortices were incubated with 0.5× TrypLE™ Express Enzyme for 20 min at 37 °C and then washed with complete growth medium (Neurobasal medium supplemented with B-27, Penicillin–Streptomycin and L-Glutamine). Following trituration, cells were added to PDL-coated coverslips in 12 well-sterile cell culture plates. At DIV 17, 1.5 μM PMA was added to the cells and incubated for 15–30 min, following which cells were washed and fixed for immunocytochemistry.

### Immunocytochemistry
Coverslips of cortical neurons from Sprague-Dawley postnatal pups were washed with 1× sterile PBS and then fixed with freshly prepared 4% PFA for 15 min at room temperature. The cells were incubated in 0.1% Triton X in 1× PBS for 10 min and then kept for blocking in 1× PBS containing 3% BSA for 30 min. Coverslips were then incubated with primary antibodies (Table S4) against Munc13-1 (1:500, rabbit) or Munc18-1(1:500, rabbit) and SNAP-25 (1:200, mouse) or Syntaxin1 (1:500, mouse) or Syb2 (1:500, mouse), and Synapsin (1:1000, guinea pig) in PBS containing 0.1% BSA at 4 °C overnight. Coverslips were then washed with 1× PBS and incubated with fluorescent secondary antibodies (AlexaFluor488 conjugated anti-rabbit IgG, AlexaFluor555 conjugate anti-mouse IgG, and AlexaFluor647 conjugated anti-guinea pig IgG) for 1 h at room temperature. Cells were then washed with 1× PBS and mounted on glass slides.

### Confocal imaging and colocalization analysis
Fixed slides were imaged on an Olympus FV3000 confocal microscope equipped with a 60× 1.4NA oil objective with laser power (18 μW for 488 lasers, 2.4 μW for 561 lasers, and 34.3 μW for 640) and offset (4%) settings kept same for all experiments. The signal was detected using a spectral detector and using a multi-band dichroic mirror 405//488/ 561/633 nm. 512 × 512 pixel images were acquired using a High Sensitive GaAsP PMT detector at a detector gain of 550 and at a zoom of 4×.

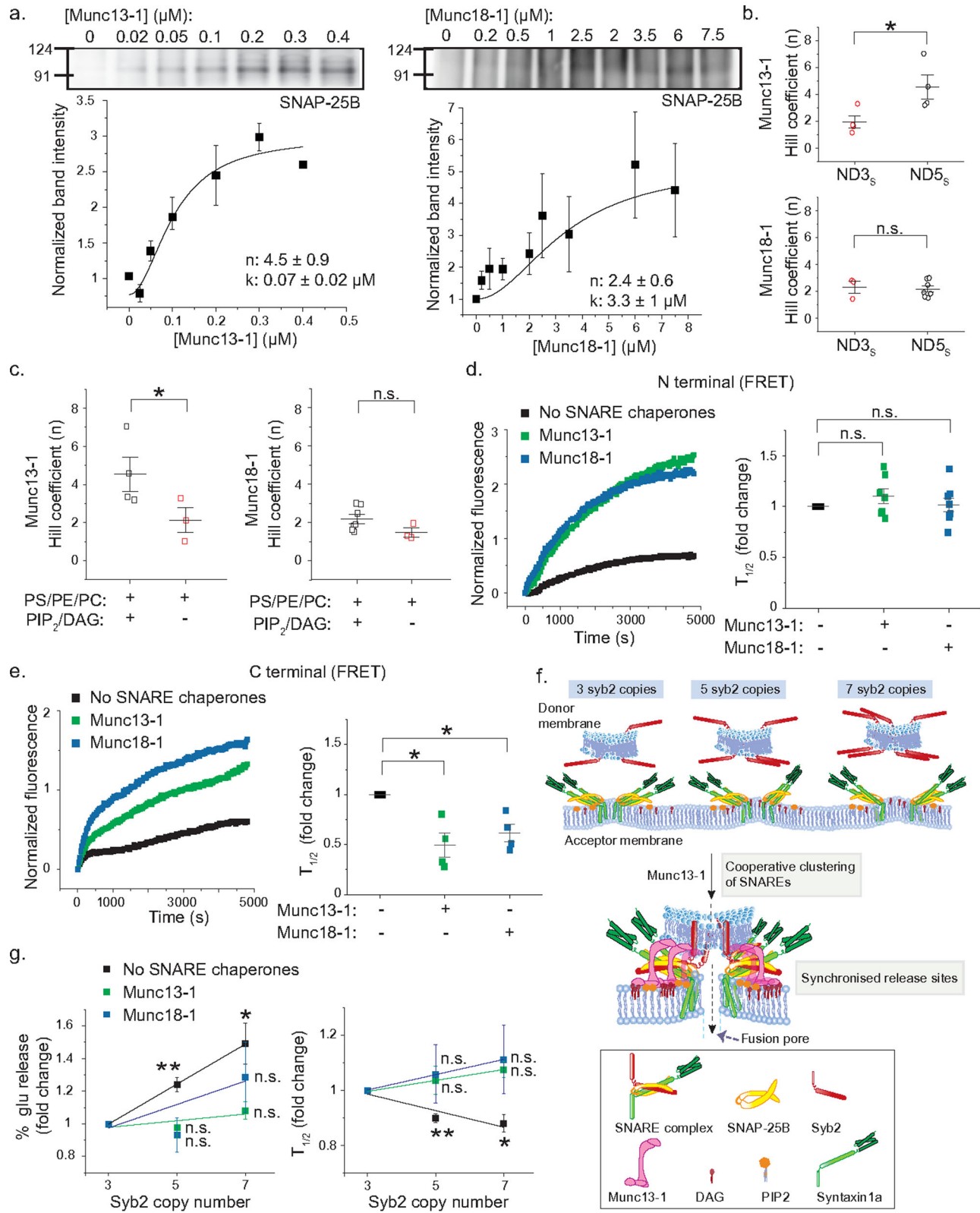

A frame average of 2 was applied to both channels. For image analysis, the Z project of the z stack was used to mark boundaries of individual cells and cell bodies removed before thresholding for Synapsin signal. The auto local threshold was applied using the Bernsen method with a radius of 50 to select regions in the neurites with Synapsin signal. The selected regions were then used for colocalization using the JACoP plugin in ImageJ. Manders' coefficient M1 indicates a fraction of

Munc13-1 or Munc18-1 overlapping with an individual SNARE protein as indicated, while M2 indicates a fraction of SNARE proteins overlapping Munc13-1 or Munc18-1.

### Protein purification

cDNA for the following proteins was derived from *Rattus norvegicus* and expressed in *E. coli* BL21(DE3) cells under T7 promoter control in

**Fig. 5 | Munc13-1 and Munc18-1 synchronize vesicular secretion by differentially organizing trans-SNARE assembly. a** Representative immunoblot showing trans-SNARE complexes with increasing [Munc13-1] (top, left) and [Munc18-1] (top, right). Fitted scatter plots showing fold change in band intensity for trans-SNARE complex formation as indicated. The hill coefficient ($n$) and $EC_{50}$ ($k$) are indicated. t-SNARE liposomes lipids−PS/PE/PC/PIP$_2$/DAG. **b** Scatter plot showing pooled data for hill coefficients (top−Munc13-1, bottom−Munc18-1) obtained from (**a**) using ND3$_S$ and ND5$_S$; $n = 4$ (for ND3$_S$/Munc13-1), $n = 5$ (for ND5$_S$/Munc13-1), $n = 3$ (for ND3$_S$/Munc18-1), $n = 7$ (for ND5$_S$/Munc18-1) independent blots; $p = 0.027$ (Munc13-1), 0.403 (Munc18-1). **c** Scatter plot showing pooled data for hill coefficients (left−Munc13-1, right−Munc18-1) obtained from (**a**), as indicated; $n = 3$ (for ND5$_S$/Munc13-1) and $n = 4$ (for ND5$_S$/Munc18-1) independent blots for membrane lipid PS/PE/PC; $p = 0.041$ (Munc13-1), 0.347 (Munc18-1). **d** Left, representative time course of corrected Cy5 emission, as ND5$_S$$^{cy3N}$ reacts with t-SNARE liposome$^{cy5N}$ as indicated. Right, scatter plot shows fold change in time at which FRET intensity is half of the maximum ($T_{1/2}$) as indicated; $n = 7$ independent trials, $p = 0.108$ (Munc13-1), 0.421

(Munc18-1). **e** Left, representative time course of corrected Cy5 emission, as ND5$_S$$^{cy3C}$ reacts with t-SNARE liposome$^{cy5C}$ as indicated. Right, scatter plot shows fold change in time at which FRET intensity is half of the maximum ($T_{1/2}$) as indicated; $n = 4$ independent trials; $p = 0.012$ (Munc13-1), 0.011 (Munc18-1). **f** Model illustrates Munc13-1 mediated cooperative clustering of SNARE complexes, which synchronizes release events. **g** Fold change in the maximum glu release percent (average) (left) and the average time at which % glu released reaches the half-maximum ($T_{1/2}$) (right), for ND3$_S$, ND5$_S$ and ND7$_S$, in the absence and presence of Munc13-1 or Munc18-1; $n = 5$ independent trials for each condition, three different ND preparations; $p = 0.005$ (ND5$_S$/No SNARE chaperones), 0.356 (ND5$_S$/Munc13-1), 0.276 (ND5$_S$/Munc18-1), 0.017 (ND7$_S$/No SNARE chaperones), 0.099 (ND7$_S$/Munc13-1), 0.127685534 (ND7$_S$/Munc18-1). Molecular weights (in kDa) are shown. Data presented as mean ± SEM; Student's $t$-test (one-tailed) was performed to compare the ND5$_S$ or ND7$_S$ means with ND3$_S$. *$p < 0.05$, **$p < 0.01$, n.s. non-significance. Relevant source data are provided as a Source Data file.

the pET28a vector: His6-tagged full-length Syntaxin-1a and SNAP-25B and His6-tagged Munc18-1; while GST tagged full-length Syb2 was cloned into pGEX vector. His6-tagged rat Munc13-1 fragment (residues 529–1735, spanning C1, C2B, MUN, and C2C domains; with 1408–1452 replaced by EF motif) cloned in a pET28a vector for bacterial expression was a kind gift from Prof. Jose Rizo-Rey at UT Southwestern. In-Fusion cloning was done with primers (Table S5) to introduce the H567K mutation, double mutation F1234A, K1236A (FKAA), and D1358K mutations to the Munc13-1 pET28a construct, and D326K mutation to Munc18-1 pET28a construct.

SNAREs, Syb2, and t-SNARE heterodimers comprising Syntaxin-1A and SNAP-25B and Munc18-1 were expressed at 37 °C for 4–5 h with corresponding antibiotic resistance, while Munc13-1 WT and H567K mutant were expressed at 20 °C overnight, post induction with 0.5 mM IPTG. BL21 DE3 bacterial pellets were resuspended in resuspension buffer (25 mM HEPES−KOH pH 7.4, 400 mM KCl, 10% glycerol, and 10 mM β-mercaptoethanol). Protease inhibitor cocktail and EDTA-free DNase I were then added, and 2% Triton X 100 (only for SNAREs) was added prior to sonication (3 × 30 s cycle with 60% frequency). Cell lysates containing overexpressed SNARE proteins were incubated overnight with rotation at 4 °C before centrifugation of the cell lysate at 37,044×$g$ for 1 h in a JA-25.5 rotor (Beckman Coulter). The supernatant was then incubated for >2 h at 4 °C with Ni-NTA agarose or GST Sepharose beads (GE Healthcare; 1 ml of a 50% slurry per 2 l of cell culture) equilibrated in resuspension buffer containing 10 mM imidazole. The slurry was loaded onto a column and washed extensively with resuspension buffer containing 1% Triton X-100 and 20 mM imidazole, then washed with OG wash buffer (25 mM HEPES-KOH, pH 7.4, 400 mM KCl, 50 mM imidazole, 10% glycerol, 5 mM β-mercaptoethanol, 1% OG). Elution of His6-tagged proteins was done in a resuspension buffer containing 500 mM Imidazole. For GST-tagged Syb2 elution, the beads were incubated with thrombin (0.15 mg/ml, as per manufacturer's instructions) at 4 °C overnight. Thrombin was then removed by incubating the elutes with Benzamidine Sepharose beads for 1.5 h at 4 °C. Purity of proteins was assessed using SDS PAGE after staining with Coomassie brilliant blue. Munc13-1, Munc13-1 (H567K), Munc13-1 (D1358K), Munc13-1 (FKAA), Munc18-1, Munc18-1 (D326K), MSPE3D1 and iGluSnFR were also purified as His6-tagged proteins, as described above, except all detergents were omitted from the wash buffers and reconstitution buffer.

### Liposomes preparation

t-SNARE liposomes were prepared[13,44] with the following lipid compositions for different experiments−(1) 32% PE, 52% PC, 16% PS; (2) 31% PE, 51% PC, 15% PS, 1.5% PIP$_2$, 1.5% DAG; (3) 75% PE, 25% PG. Dried lipids were rehydrated in reconstitution buffer (25 mM HEPES-KOH, pH 7.4, 100 mM KCl, 2 mM β-mercaptoethanol, and 0.3% OG) with or without t-SNAREs heterodimers. For t-SNARE liposomes containing glutamate,

an additional 50 mM glutamate was used during rehydration. The ratio of t-SNARE to lipid molecules used was 14:3000 for a theoretical yield of 100 copies of t-SNAREs per liposome. Dialysis was performed against reconstitution buffer overnight at 4 °C, followed by the isolation of t-SNARE liposomes by density gradient (using ACCUDENZ gradient) ultracentrifugation at 171,500×$g$ for 2 h (4 °C) in Optima MAX-XP ultracentrifuge (Beckman Coulter).

### Nanodiscs preparation

Reconstitution of Syb2 into NDs was performed[13,44] by rehydrating dried lipids with Syb2, MSP1E3D1 (MSP variant to generate 13 nm NDs) or NW30 (MSP variant to generate 30 nm NDs) in reconstitution buffer containing 0.02% DDM. The detergent was slowly removed with Bio-Beads (1/3 volume) with gentle rotation overnight at 4 °C. The NDs were purified by gel filtration (AKTA) using a Superdex 200 10/300 GL column, equilibrated in a reconstitution buffer. The ratio of MSP to lipid molecules used was 3:180 for 13 nm NDs and 2:1000 for 30 nm NDs; while the MSP to Syb2 ratios were 3:1.5 (ND3$_S$), 3:6 (ND5$_S$), 3:9 (ND7$_S$) and 2:8 (ND25$_L$). The lipid composition of 16% PS, 32% PE, and 52% PC was used for all experiments.

### Co-floatation assay

Liposomes (30 µl) with membrane lipid composition of 31% PE, 51% PC, 15% PS, 1.5% PIP$_2$, and 1.5% DAG without any t-SNAREs reconstituted in them were mixed with Munc13-1(0.5 µM) or different mutated Munc13-1 proteins (0.5 µM) as indicated. The reaction volume was made up to 60 µl with reconstitution buffer containing 25 mM HEPES-KOH, pH 7.4, 100 mM KCl, and 2 mM β-mercaptoethanol and kept at 37 °C for 1 h. Samples were then mixed with 60 µl of 80% ACCUDENZ, and a density gradient was created with 60 µl 30% ACCUDENZ and 18 µl reconstitution buffer (0% ACCUDENZ) in a tube and centrifuged at 61,824×$g$ for 1.5 h at 4 °C using TLA-100 ultracentrifugation rotor. In total, 30 µl of liposomes were collected from the 0–30% interface, and 30 µl of the bottom layer was collected from each tube. The samples were prepared and resolved by SDS-PAGE and immunoblotting for Munc13-1.

### BAPTA titration assay

To assay for the minimum [BAPTA] to be used in experiments for calcium chelation, varying BAPTA concentrations were prepared in reconstitution buffer (25 mM HEPES-KOH pH-7.4, 100 mM KCl). The calcium indicator Fluo-4 was then mixed with the samples, and its fluorescence (Ex: 494 nm and Em: 516 nM) was recorded using the Tecan SPARKcontrol Magellan microplate reader.

### BLM electrophysiology

Planar lipid bilayer (BLM) recordings were performed using a BLM Workstation from Warner Instruments (USA)[13]. Briefly, 30 mg/ml total lipids (BLM lipid composition: (1) 16% PS, 32% PE, 52% PC, (2) 31% PE,

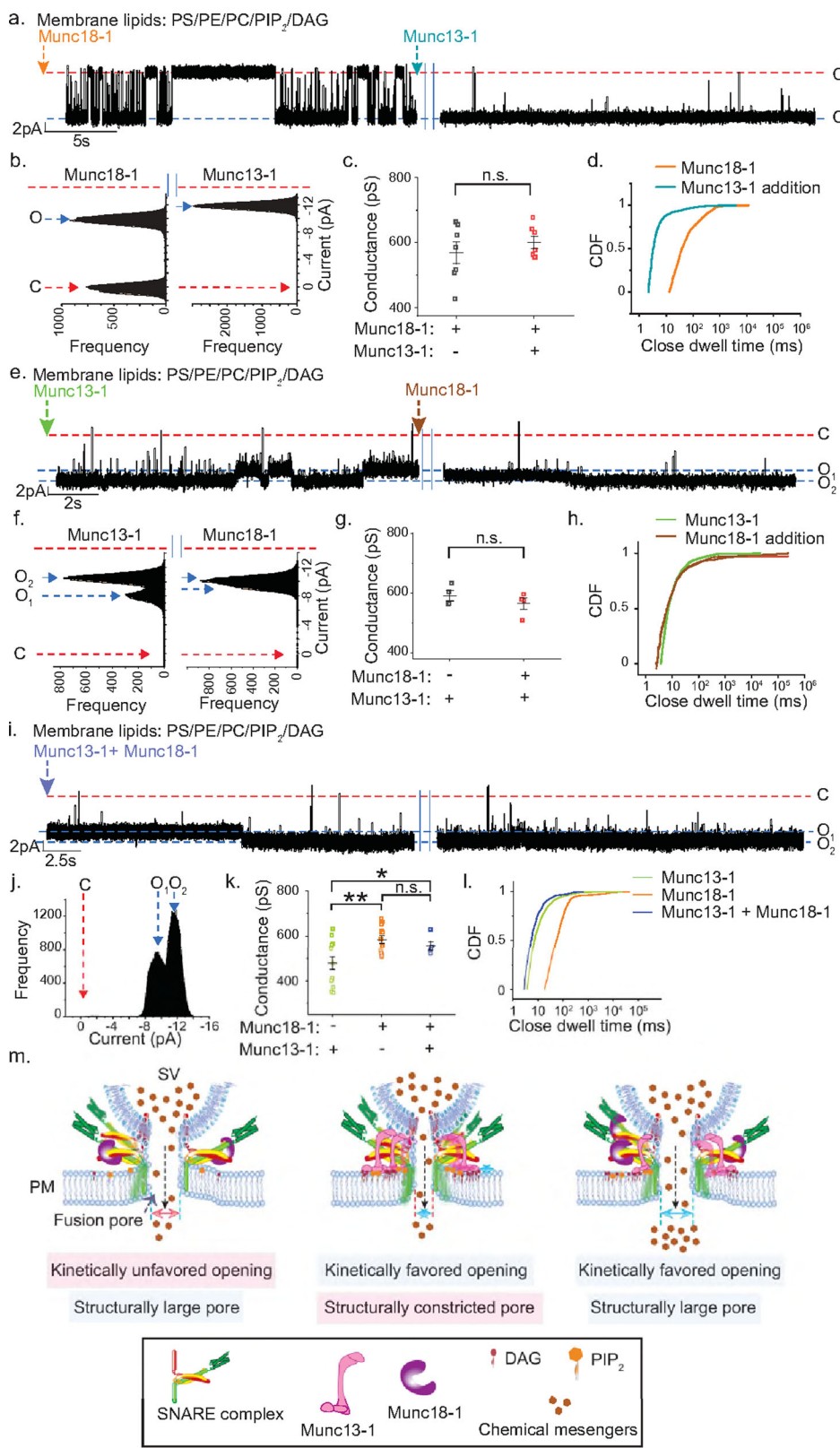

51% PC, 15% PS, 1.5% PIP$_2$, 1.5% DAG, (3) 31% PE, 52% PC, 16% PS, 1% PIP$_2$, (4) 31.5% PE, 51.5% PC, 15.5% PS, 1.5% DAG, (5) 31.5% PE, 51.5% PC, 15.5% PS, 1.5% PA, (6) 30.3% PE, 50.3% PC, 14.3% PS, 5% DAG, and (7) 31.97% PE, 51.97% PC, 15.97% PS, 0.1% DAG) were dried and dissolved in n-decane and then painted onto a 150-µm aperture in a 1 ml polystyrene cup (Warner Instruments) and dried for 10 min. The cup is added onto a bilayer cell forming a cis and trans chamber. Silver-silver chloride electrodes were connected to each chamber wherein a cis buffer (25 mM HEPES-KOH, pH 7.4 and 100 mM KCl) was added in the cis chamber, and a trans buffer (25 mM HEPES-KOH, pH 7.4 and 10 mM KCl was added in the trans chamber. The BLM lipids in n-decane were gently re-applied to the aperture with a brush and repeatedly worked on with air bubbles until a conductance-blocking seal was formed and the desired capacitance was achieved.

**Fig. 6 | Fusion pore fate is dictated by the sequential action of SNARE cha-perones. a** Representative trace of single ND5$_S$ pore triggered open by Munc18-1 (orange), addition of Munc13-1(cyan) is indicated. Closed (C) and open (O) states; respective currents are indicated. **b** Current histogram of the pore shown in (**a**). Closed (C) and open (O) states are indicated by red and blue arrows, respectively. **c** Scatter plots show pooled data for pore conductance as indicated; $p = 0.213$. **d** CDF of closed dwell times as indicated. **e** Representative trace of single ND5$_S$ pore triggered open by Munc13-1 (green), the addition of Munc18-1 to that pore is indicated (brown). Closed (C) and multiple open states (O$_1$/O$_2$) of the individual pores are indicated. **f** Current histogram of the pore shown in (**e**). Closed (C) and open (O$_1$/O$_2$) states are indicated by red and blue arrows, respectively. **g** Scatter plots show pooled data for pore conductance as indicated; $p = 0.172$. **h** CDF of closed dwell times as indicated. **i** Representative trace of single ND5$_S$ pore triggered open by both Munc13-1 and Munc18-1. **j** Current histogram of the pore shown in (**i**).

Closed (C) and open (O$_1$/O$_2$) states are indicated by red and blue arrows, respectively. **k** Scatter plots show pooled data for pore conductance triggered open by either Munc13-1 (green) or Munc18-1 (orange) or both (blue); $n = 13$ (for Munc13-1), $n = 15$ (for Munc18-1) and $n = 5$ (for Munc13-1 + Munc18-1) single pores were analyzed; $p = 0.02$ (Munc13-1/Munc13-1 + Munc18-1), 0.002 (Munc13-1/Munc18-1), 0.141 (Munc18-1/Munc13-1 + Munc18-1). **l** CDF of closed dwell times comparing indicated experimental conditions. **m** Model illustrating how Munc13-1 and Munc18-1 can differentially control the cargo release probabilities at the release site by altering the fusion pore size and kinetics. For **a**–**d**, $n = 7$ independent BLMs; three sets of NDs were used. For **e**–**h**, $n = 4$ independent BLMs; two sets of NDs were used. For **i**–**l**, $n = 3$ independent BLMs; two sets of NDs were used. Data are presented as mean ± SEM; Student's $t$-test (one-tailed) was performed to compare the two means; n.s. non-significance. Relevant source data are provided in the Source Data file.

## Single fusion pore measurements

Single pore measurements were performed as following[44]. After the formation of BLM of desired capacitance, t-SNARE liposomes (lipid composition: PE/PG) were added to the cis chamber, which spontaneously fused with the planar bilayer depositing t-SNAREs into the BLM at a density of 0.4 molecules per μm² [43]. Fusion pores were formed at room temperature within 10-40 min after nanodiscs (ND5$_S$) were added to the cis chamber. Current was monitored for >60 min using Bilayer Clamp Amplifier BC-535 (Warner Instrument) and a Digidata 1550B (with Humsilencer) acquisition system (Molecular Devices Corp.), in the absence and presence of indicated amount of Munc13-1 or Munc18-1 or in combination or mutated Munc proteins as indicated. All recordings were done in the presence of BAPTA (10 μM) in the cis chamber. Wherever indicated, 500 μM [Ca²⁺]$_{free}$ was added to the cis chamber while recording. Single-channel recordings were acquired at 10 kHz using pCLAMP 11 (Molecular Devices, LLC.) software and were filtered at 5 kHz using a multisection Bessel filter. $\Delta\psi \equiv \psi_{cis} - \psi_{trans}$ ($\psi_{trans} \equiv 0$ V). Pore formation and dynamics were studied at $\Delta\psi = -20$ mV.

## Analysis of single fusion pore unitary currents

Single-channel data were analyzed using Clampfit 10.7 (Molecular Devices) and Origin 2020b (OriginLab). In all Figs. showing BLM recordings, the representative traces were filtered twice at 0.5 and 0.3 kHz for display purposes.

Current histograms were plotted using CLAMPFIT 10.7 and fitted with Gaussian functions to calculate mean closed-state currents ($I_C$) and open-state currents ($I_O$). The conductance in pico-siemens (pS) was calculated using the equation:

$$G = (I_C - I_O)/\Delta\psi \tag{1}$$

Dwell times corresponding to the fully open and fully closed states were measured in individual records using the event detector in CLAMPFIT10.7. The fraction of time fully closed ($f_c$) was calculated using the equation:

$$\text{Fraction of time closed} = (\text{closed dwell time})/ \\ (\text{open dwell time} + \text{closed dwell time}) \tag{2}$$

## Kinetic analysis of single pore data

Kinetic analysis of single channel data was performed as previously reported[43]. Pooled closed state dwell times ($\langle t_{c\text{-}obs} \rangle$) across all trials were calculated in Origin 2020b. These were further statistically analyzed by CDFs [defined by the probability $P(t_{closed} \leq t)$] using Origin 2020b and fitted with different exponential fits[43]. CDFs and their corresponding exponential fits generated from the closed state dwell times provide information about the existence of kinetic intermediates for fusion pore opening in different experimental conditions. For

closed-state CDF comparison under different conditions, we analyzed the slow kinetic components. Similarly, CDFs were obtained and analyzed for open dwell times of pores formed with different concentrations of Munc18-1.

## Immunoblotting

Immunoblotting of the samples was performed after mixing the samples with 1× (final concentration) Laemmli sample buffer, run on SDS PAGE, and blotted onto a PVDF membrane. The membrane was blocked with 5% skimmed milk in Tris-buffered saline containing 0.1% Tween 20 (TBST) and incubated with primary antibodies (Table S4) overnight at 4 °C. The blots were washed with TBST and incubated with corresponding HRP conjugated secondary antibody (Table S4) for 1 h at room temperature. The blots were developed using enhanced chemiluminescence. All uncropped immunoblots of cropped blots shown in relevant figures are supplied in the Source Data file.

## Steady-state SNARE complex formation assay

t-SNARE liposomes (5 nM) and NDs (100–150 nM) were incubated with different concentrations of Munc13-1 or Munc18-1 (as indicated) in reconstitution buffer (25 mM HEPES-KOH pH-7.4, 100 mM KCl, 2 mM β-mercaptoethanol) overnight at 4 °C. The overnight reaction samples were split into two fractions to probe for SNARE complex and for using as heated controls. For SNARE complex blots, the overnight reaction samples were mixed with 1× Laemmli sample buffer without heating, and SDS-PAGE was run, followed by immunoblotting with SNAP-25 antibody for quantifying SNARE complexes. For control blots, the reaction samples for SDS-PAGE and immunoblotting were prepared by heating at 95 °C for 15 min to disrupt SNARE complexes formed and to check individual levels of t-SNAREs, Syb2 and Munc13-1 or Munc18-1. The blots were analyzed using ImageJ software and the band intensity of SNARE complexes formed was quantified under indicated conditions. The graph was plotted as a fold change in band intensity to its corresponding Munc13-1/Munc18-1 concentrations. Fitting of the data was done using the modified Hill equation where $y_{min}$ is the minimum SNARE complex formed, $y_{max}$ is the maximum SNARE complex formed, $k$ is the EC$_{50}$, and $n$ is the Hill coefficient:

$$y = y_{min} + \frac{x^n}{k^n + x^n}(y_{max} - y_{min}) \tag{3}$$

## Ensemble FRET assay for SNARE assembly

Cysteine-free mutants of SNAREs were engineered using infusion cloning, and single cysteine mutations were introduced at the following sites: (1) N terminal–Syntaxin 1a I203C, Syb2 Q33C and (2) C terminal–Syntaxin 1a V241C, Syb2 A72C. Following purification of the engineered constructs as mentioned above, the proteins were buffer exchanged using 25 mM HEPES, 400 mM KCl, 10% glycerol, 1%OG, and 0.1 mM TCEP. Single cysteine mutants were labeled by mixing Cy3

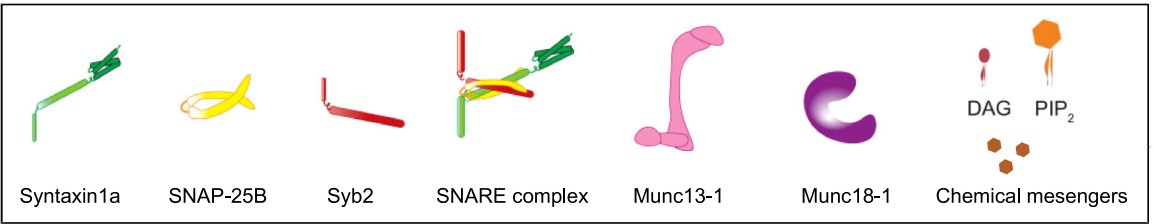

**Fig. 7 | Model for Munc13-1/Munc18-1's action on fusion pore.** Illustration shows the action of Munc13-1/Munc18-1 on dynamic trans-SNARE complexes to cooperatively cluster and synchronize fusion pore assembly at the release site.

maleimide (donor) with Syb2 and Cy5 maleimide (acceptor) with Syntaxin 1a at protein: dye molar ratios of 1:25 as per manufacturer's instructions (Amersham). Labeled SNAREs were further reconstituted into v-SNARE NDs and t-SNARE liposomes. For the FRET assay, 5 nM t-SNAREs liposome (Cy5) was mixed with 25 nM v-SNARE NDs (Cy3) in a 364-well flat black plate in the absence and presence of regulatory factors (0.5 μM Munc13-1, 5 μM Munc18-1) as indicated. For control, ND5 (Cy3) only and t-SNAREs liposome (Cy5) only samples were added to the black plate for fluorescence measurement. FRET was measured by exciting the donor at 510 nm and acceptor emission at 670 nm was recorded with time at 30 °C in Tecan SPARKcontrol Magellan microplate reader. Bleach correction was done by exponentially fitting

donor ND5 (Cy3) only fluorescence data and obtaining the bleaching ratio. The FRET measurements obtained were divided by the bleaching ratio of donor fluorescence for each time point (Nathalie B Vicente et al. 2007 J. Phys.: Conf. Ser. 90 012068). After correcting for photobleaching of donor fluorescence, acceptor bleed-through was corrected by subtracting t-SNAREs liposome (Cy5) only fluorescence data. The corrected fluorescence data was plotted as $(F - F_0)/F_0$ versus time, where $F_0$ is the initial corrected fluorescence.

**Content release assay**

For glutamate release assay, 5 nM t-SNARE liposome entrapped with glutamate was mixed with 100 nM NDs and 1 μM iGluSnFR glutamate

sensor[18,74] in the absence or presence of regulatory factors (0.5 µM Munc13-1, 5 µM Munc18-1) as indicated. The iGluSnFR fluorescence (Ex: 480 nm and Em: 510 nM) was recorded with time at 30 °C using the Tecan SPARKcontrol Magellan microplate reader. After each run, 5 µl of 5% Triton X-100 was added to each reaction, and data were collected for another 15 min to obtain maximum fluorescence of iGluSnFR. The data was analyzed after correcting for basal iGluSnFR fluorescence for liposomes in each trial, and normalized release was calculated as a percentage of maximum release.

### Lipid mixing assay
For the lipid mixing assay, FRET pair labeled ND5$_s$ was prepared as described above with lipid composition: 29% PE, 16% PS, 52% PC, 1.5% NBD-PE, 1.5% Rhod-PE. t-SNARE liposomes with varying lipid composition as indicated, were mixed with FRET pair labeled ND5$_s$. NBD fluorescence was measured at 30 °C using a Tecan SPARKcontrol Magellan microplate reader for 1 h (Ex: 460 nm and Em: 538 nm). After each run, 5 µL of 5% Triton X-100 was added to each reaction, and data were collected for 10 min.

### Leakage assay
t-SNARE liposomes were prepared as described above, and 20 mM sulforhodamine B (SRB) was incorporated into it during rehydration. 5 nM t-SNARE liposomes entrapped with SRB were incubated in the absence or presence of SNARE chaperones (0.5 µM Munc13-1, 5 µM Munc18-1) as indicated. SRB release from t-SNARE liposomes was recorded at 30 °C using a Tecan SPARKcontrol Magellan microplate reader for 1 h by measuring SRB fluorescence dequenching (Ex: 532 nm and Em: 586 nm). After each run, 5 µL of 5% Triton X-100 was added to each reaction, and data were collected for 10 min. For positive control to indicate membrane fusion, t-SNARE liposomes were incubated with ND5$_s$, and SRB fluorescence was measured.

### Statistical analysis
The number of independent trials is provided in the figure legends, along with the statistical tests that were performed. Error bars represent SEM.

### Reporting summary
Further information on research design is available in the Nature Portfolio Reporting Summary linked to this article.

## Data availability
All data necessary to support and validate the experiments have been provided in the text and in the Supplementary Information file. Source data are provided in this paper.

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

## Acknowledgements

We thank Das lab members for their suggestions and comments regarding this manuscript. We thank Dr. Shital Suryavanshi, Darshana Kapri, Sashaina Fanibunda (TIFR), and Shreya Joshi (St. Xavier's College Mumbai) for technical assistance and animal house usage. We thank Dr. Sundar R. Naganathan and Sumit Sen (TIFR) for the helpful discussion on image processing. This study was supported by a grant (19P0119) from the Department of Atomic Energy (Govt. of India) and TIFR-Mumbai to D.D.

## Author contributions

D.D. conceived of the project, and D.D. and B.B. designed the experiments. B.B., L.Y., M.S., M.G., K.S., B.K.V., R.K.N., and D.D. performed the experiments. D.D. supervised the projects. D.D. and B.B. wrote the paper.

## Competing interests

The authors declare no competing interests.
