## [Peer Review File · Nature Communications]

Differential SNARE chaperoning by Munc13-1 and Munc18-1 dictates fusion pore fate at the release siteReviewers' Comments:

Reviewer #1 (Remarks to the Author):

The authors study the roles of Munc13 and Munc18 proteins on SNARE mediated membrane fusion. Although they make some potentially interesting observations, many of their results are hard to explain and/or inconsistent with basic expectations. I do not think this paper should be published anywhere.

Here are some more details:

1) The results are presented in a strange way, giving the wrong impression that the authors studied fusion pores in neurons. There is no mention of the main methodology used until the second subsection in Results.

2) The only experiments with rat cortical neurons are studies of colocalization of SNAP25 with Munc13 or Munc18 in the presence or absence of the DAG mimic PMA.

It is not clear why SNAP25 is chosen for these experiments. The obvious choice would have been Syntaxin, since both Munc13 and Munc18 interact with it.

Importantly, small changes in colocalisation cannot be taken seriously in the absence of controls. How specific are the antibodies? How does PMA affect antibody labeling? Does it induce similar minor changes in colocalization of unrelated proteins? By how much?

As the authors note, "Because Munc18-1 doesn't possess any phorbol ester binding sites, it was unclear why this Munc protein's co-localization with SNAREs would be phorbol ester dependent."

Note that DAG is a surfactant and DAGs are used as emulsifiers. It is not surprising their use may cause slight changes in antibody labeling.

Finally, there is no clear link between these colocalization studies and the fusion pore measurements reported later in the paper.

3) There are many statements that are hard to understand. For example, p.4, penultimate paragraph: "The above Munc13-1 actions were Ca²⁺ independent, as all the experiments were done in presence of BAPTA (Fig. S2a)." How can you deduce the effect is calcium independent if calcium is not present? I suppose the authors mean to say the effect did not require calcium.

4) In the nanodisc-liposome fusion experiments, controls for leakage or liposome aggregation are missing.

Fig. 2 a: please try to draw the schematics to scale. Is there really enough space to accommodate all the proteins in the ND?

The plot of "fold change" is meaningless, as GLu release from the PS/PE/PC/PIP2/DAG liposomes continues almost linearly after release from the PS/PE/PC liposomes has completely leveled off. That is, if

one waits past ~4000 s, there would be more release from the former.
Is the use of student's t-test justified? (are the data normally distributed?)

5) The main results are obtained using an assay in which nanodiscs (NDs) are fused with planar suspended bilayers. This assay is sensitive to single fusion pores and has very good time resolution.

DAG is found to shorten pore lifetimes. This is not surprising, given that DAG is a surfactant with a negative curvature. It is expected to shorten pore lifetimes (e.g. see Karatekin...Brochard-Wyart, *Biophys. J.* 2003).

6) Adding Munc13 restores longer pore lifetimes to some extent. Munc13 has a DAG binding C1 domain. A Munc13 mutant that is unable to bind DAG (H567K) is expected to have NO effect on pore dynamics compared with pores induced by SNAREs alone. Yet, the mutant has an even stronger effect on pore conductance than WT Munc13, and causes longer pore lifetimes compared to SNARE-alone pores. It is completely unclear what to make of these results.

Similarly, it is not clear how to interpret the results about SNARE copy numbers and the effects of calcium.

7) The effects of Munc18 are most simply explained by Munc18 (a soluble protein) transiently blocking the pores. As the Munc18 concentration is increased, the probability of the block increases. Many controls would need to be made to exclude the possibility of Munc18 blocking pores.

8) The suspended bilayer method is an interesting approach, but the presence of the solvent (n-decane here) dramatically affects membrane properties (e.g. see Santinho...Thiam, *Biophys. J.* 2021, Campillo...Sykes, *Biophys. J.* 2013; Melikov...Chernomordik *Biophys. J.* 2001). Thus one needs to be very careful in interpreting results, especially when results are unexpected and hard to explain.

9) There are many self citations, but other key literature is omitted. For example, increased pore size as a function of increasing SNARE copy numbers was reported well before the report by Bao et al *Nature* 2018, using nanodisc-liposome fusion (by the Rothman lab) and nanodisc-cell fusion (Karatekin lab) assays.

Reviewer #2 (Remarks to the Author):

The manuscript entitled "Differential SNARE chaperoning activity determines quantal size during membrane fusion", by Bhaskar et al. provides the functional evidence of SNARE chaperons Munc13-1 and Munc18-1 in rescuing nascent fusion pore from DAG-mediated inhibitory states. Munc13-1 and Munc18-1 differentially regulate pore size and kinetics. Munc13-1 restricts the pore in a narrow low

conductance state while Munc18-1 triggers pores in their dilated large conductance states. They also provided data showing that Munc13-1 and Munc18-1 allow a cooperative zippering of SNARE from N- to C- terminal and assist in synchronizing content release during membrane fusion, indicating the crucial role of Munc proteins in modulating fusion pore assembly.

Addressing the following concerns would improve the study significantly.

Major Points:

1. DAG (diacylglycerol) and PIP2 (phosphatidylinositol 4,5-bisphosphate) are consistently observed in nearly all experimental investigations mentioned within the manuscript. However, the literature review related to their significance is apparently lacking. Therefore, the association between DAG and PIP2 on SNARE proteins and fusion pore should be introduced.
2. In Figure 1, the author initially explored the organization of release sites by Munc13-1 and Munc18-1. However, it is essential to investigate the involvement of additional components of the fusion site, syntaxin. It is recommended to employ the same methodology to examine the co-localization of Munc proteins and syntaxin, shedding light on their potential collaborative roles in synaptic vesicle fusion.
3. As depicted in Figure 1, it is difficult to observe the increase of SNAP25 after PMA treatment. In order to enhance visual clarity and provide a more detailed representation, it is recommended to replace the current image with a higher-quality version and enlarge specific details, as demonstrated in Supplementary Figure S1.
4. In Figure 1c and 1d, the difference in Munc18-1/SNAP25B colocalization is much more significant than that of Munc13-1/SNAP25B, and at the end of the first paragraph of the Results, the authors also mentioned that: "Because Munc18-1 doesn't possess any phorbol ester binding sites.....in a phorbol ester dependent manner", the experimental results reveal a novelty that warrants an extended discussion in the manuscript.
5. The consistency between the data depicted on the left side of Figure 2b and the accompanying statistical graph in the middle appears to be questionable, suggesting a potential inconsistency in their depiction. To address this issue, a more comprehensive explanation is needed regarding the methodology employed to generate Figure 2b. And it is better to plot the averaged traces of Figure 2a middle and Figure 2b left, respectively.
6. In Figure 2c, the authors demonstrated that the presence of PIP2/DAG in the BLM lipid exerted a significant impact on the stability of individual pores. However, it remains crucial to investigate how DAG, as an independent entity, influences the state of the fusion pore. The experiment may provide a more thorough understanding of the underlying mechanisms involved.
7. In the experiment shown in Figure 4, an explanation for two concentrations of Munc18-1 that were applied is needed. And why is the Fraction-close of 5uM Munc18-1 lower than that of 10uM Munc18-1? These interesting observations require speculative explanations in the Discussion section of the manuscript. The authors can explore potential factors that may contribute to the observed Fraction-close differences. These may include changes in Munc18-1 binding affinity at different concentrations, potential allosteric effects on the fusion machinery, or differential interactions with other regulatory proteins involved in fusion pore dynamics.
8. For Fig 5a and Fig S5a and S5b, the methods for immunoblots of SNARE complex and individual proteins were not described clearly. How is the fraction of liposome-nanodisc reaction processed for blotting? Please revise the method section to make it easier to follow. Is it native PAGE for the detection

for SNARE complex? Please clarify it. For the SNARE complex detection, it is not enough to only show SNAP-25 band and re-blot for Munc13-1/Munc18-1/Syb2/Syntaxin-1a are quite necessary to demonstrate the composition of the complex. Besides, the ladders for protein size should be labeled in blots for the detection of both complex and individual proteins. In addition, why is the concentration of Munc13-1 and Munc18-1 indicated in blots different from that in plot for band intensity measurement? Please explain it.

9. As depicted in Figure 3a, Munc13-1 stimulated the pore to interconvert between two different open states, O1 and O2, and the Current between -6 to -9 pA. But in Figure 6e and 6f, only one open state of Munc13-1 was shown, and the Current between -9 to -13 pA. The data looks inconsistent. Please give a reasonable explanation.

10. Many groups of data were collected from only 3-6 BLMs, which are a bit risky considering that variations can be large. For example, the group of Munc18-1 in Figure 2b right and the group of Munc13-1 in Figure 4e show large variation. I suggest that the authors increase the cell number to enhance the significant difference.

11. The part on the molecular mechanism of the pore regulation by Munc13-1 and Munc18-1 is still weak. Following points need to be addressed: It's important to know which core domain they use to control the size and the open state stabilities of the fusion pore, therefore, some experiments for mutations of Munc13-1 and Munc18-1 need to be presented.

Minor concerns:

1. In Figure 2b middle and right, the label for Munc13-1 and Munc18-1 should be consistent within Figure 2b left.
2. In Figure 4c, the concentration of Munc18-1 and Munc13-1 need to be stated in the legend.
3. The scales bar of the current trace in Figure 3a is not consistent with that in Figure 3e, is it possible for the author to make all the scales bar consistent in all the figures.

Reviewer #3 (Remarks to the Author):

Munc18-1 and Munc13-1 are well-established components that organize SNARE complex assembly and neurotransmitter release in neurons and neuroendocrine cells. A great amount of data has shown that Munc18-1 and Munc13-1 cooperatively chaperone SNARE complex assembly. However, it is still enigmatic whether the two chaperones would function in opening and dilation of fusion pores during the last step of SNARE-mediated membrane fusion. In this manuscript, by using previous established nanodisc-planar lipid bilayer electrophysiology approach, Bhaskar et al. revealed that Munc18-1 and Munc13-1 could individually rescue nascent fusion pores from their DAG-mediated inhibitory states with different mechanisms. Meanwhile, the presence of both Munc18-1 and Munc13-1 would further stabilize the fusion pore size and open-close dynamic, illustrating that they cooperatively orchestrate the final step of SNARE-mediated membrane fusion. Taken together, the authors presented interesting and potentially important data for the function of Munc18-1 and Munc13-1 in SNARE-mediated membrane fusion. However, there still exists numerous imperfections and defects that hinder the manuscript from being published. Overall, the manuscript may need to undergo an intensive revision to meet the requirement of a high-quality work. Some critical concerns are listed below.

Major concerns:

1. Despite the diacylglycerol-like phorbol ester PMA is commonly applied to inducing membrane localization of Munc13-1, the authors should acknowledge that PMA would also activate PKC pathways. In other words, PMA treatment would cause a variety of cellular effect, e.g., vesicle transportation and protein expression (though the treatment was lasting only 15 min as indicated by the authors). Given that the authors proposed that Munc18-1 and Munc13-1 would function in opening and dilation of fusion pores, is it more appropriate to use high-potassium treatment to induce calcium influx in this assay?
2. It is also unclear whether SNARE complex assembly would increase upon PMA treatment. The authors need to check the colocalization between the other two SNAREs (i.e., SNAP-25 and VAMP2) and Munc18-1/Munc13-1 to further confirm their conclusions.
3. Under normal circumstances, SNAREs (i.e., Syx1, SNAP-25, and VAMP2) and SNARE-chaperones exert their functions mainly in presynaptic regions. These proteins in soma and non-dendritic regions may not in their 'working state' and may be the freshly synthesized fractions. I am not quite sure how the authors quantified the colocalization. My suggestion is that adding an additional channel of presynaptic marker (e.g., synapsin) to indicate presynaptic regions, analyzing the MCC between SNAREs and Munc18-1/Munc13-1 of presynaptic marker-positive region. Meanwhile, the effects of PMA treatment in individual proteins were vague. For instance, Munc13-1 seems to transit into more smeared distribution along the neurites upon PMA treatment (Fig. 1a central panel and Fig. S1a), which contradicts with previous observations that Munc13-1 translocate to membrane structures (Duncan et al., 1999, PMID: 10488064; Houy et al., 2022, PMID: 36214779). The authors need to comment on the issue and analyze the localization of individual proteins before and after PMA treatment.
4. It is interesting that colocalization of Munc18-1-SNAP-25 could also response to PMA treatment although it does not contain any lipid binding domains. However, the authors did not make enough interpretations and discussions. A quick look at previous literatures, it has been reported that phosphorylation of Munc18-1 by PKC regulates exocytosis kinetics (Barclay et al., 2003, PMID: 12519779; Craig et al., 2003, PMID: 12950453), but the phosphorylation may affect Syx1 binding of Munc18-1. Another recent literature reported that phorbol esters and PTP enhance synaptic transmission primarily by mechanisms that are independent of PKC phosphorylation of Munc18-1 (Wang et al., 2021, PMID: 34290081). These results again indicate that the effect of PMA treatment is complicated (see point#1).
5. The authors interpreted that DAG instead of PIP2 caused the inhibitory action on fusion pore. Despite they analyzed the individual effect of PIP2 by nanodisc-bilayer electrophysiology, no direct data shows that DAG inhibits opening and dilation of fusion pore. This is an important issue since the data below were all based on this precondition. Also, the authors should analyze the individual effect of DAG and PIP2 in bulk liposome-nanodisc fusion assays.
6. Another important issue: the mechanism of the inhibitory action on fusion pore by DAG need to be illustrated or proposed. Does DAG cause fusion-incompatible membrane curvature (as DAG possesses rather small head group)? Or DAG 'cages' one or more SNAREs? Meanwhile, how do Munc13-1 and/or Munc18-1 relieve the inhibitory effect? As a reader, I could only acknowledge that Munc13-1 binds to DAG through its C1 domain according to the manuscript, however, this is probably not the case for the inhibitory-stimulatory mechanism. To better reconcile the inhibitory effect, the authors need to titrate the reconstitution% of DAG in bulk liposome-nanodisc fusion and nanodisc-bilayer electrophysiology assays. For the inhibitory mechanism, the authors may use other lipids with small headgroup (e.g., PA,

PE, ..etc.) to replace DAG and analyze the effect of fusion pore formation.

7. Munc18-1 and Munc13-1 are reported to cooperatively chaperone SNARE assembly. The authors also agree with the notion. I could understand that the authors tried to dissect the individual function of Munc18-1 and Munc13-1, however, they did not show the effect with both Munc18-1 and Munc13-1 in most of the experiments. For the bulk liposome-nanodisc fusion assay in Fig. 2b and SNARE assembly assays in Fig. 5a-e, it is recommended to include an experiment that containing both Munc18-1 and Munc13-1.

8. Similar issue: the sequential actions of Munc18-1 and Munc13-1 on fusion pores were analyzed by the authors, however, they did not show the effect with both Munc18-1 and Munc13-1 simultaneously.

9. The pore sizes (or the conductance) were not shown for the experiments in Fig. 2c. I think it is important to compare the pore sizes formed by only SNAREs (with or without DAG/PIP2) and SNAREs+Munc18-1/Munc13-1.

10. The authors need to be cautious about the bulk liposome-nanodisc fusion data. Munc13-1 is reported to directly bind to lipids and membrane structures. The C2C domain of Munc13-1 associates with membrane through hydrophobic insertion with no lipid preference (Quade et al., 2019, PMID: 30816091), which means that in vitro membrane fusion assays in the presence of Munc13-1 may be perturbed by content leaking. The authors should perform control assays for bulk liposome-nanodisc fusion assays and nanodisc-bilayer electrophysiology assays to confirm the effects were genuinely caused membrane fusion, rather than content leakage (e.g., use NDSO or NDLO).

11. It is strange that Munc13-1H567K mutant displayed markedly larger pores compared to Munc13-1WT. The authors concluded that 'These results indicated that the Munc13-1, while rescuing the pores from their DAG-induced inhibitory state, keeps the pores in a narrow-constricted conformation. Pores are kinetically stable in open states despite such structural constraints (Figs. 3c-h). This structural arrest is mediated through Munc13-1's interaction with DAG in the membrane lipids.' The interpretations were vague and incomprehensible. I suppose that the authors meant that the 'narrow pore' is caused by Munc13-1 binding to DAG. However, this is literally not a 'mechanism' but an experimental observation. If the authors believed that the binding of Munc13-1 to DAG instead of the membrane (I mean, not specific to DAG) is responsible for the effect, they should confirm whether Munc13-1H567K would still attach to membrane, since Munc13-1 C2B could also interact with PIP2. Alternatively, they may construct a chimeric Munc13-1H567K with an N-terminal membrane binding module that does not bind to DAG and check the function of the chimera in pore formation.

12. It is also incomprehensible that over-dose Munc18-1 would affect pore formation. The authors still did not present a mechanism and potentially reasonable explanation (and they did not show the conductance comparison between 5 and 10 μ M Munc18-1). Does over-dose Munc18-1 impair SNARE assembly? According to Fig. 5a, left panel, it seems not. Does over-dose Munc18-1 affect the lipid dynamics? The authors need to carry out additional experiments to further interpret the results, otherwise the data using different [Munc18-1] make no sense. Incidentally, does over-dose Munc13-1 affect pore formation as well?

13. Given that the Hill coefficient (n) of ND5S is larger than ND3S, the authors did not present the parameter k for comparison (also lacking representative traces, see minor concerns). It is important to judge the cooperative effect with full fitting parameters. Similar issues were found in Fig. 5c.

14. I would doubt about the interpretation of $T_{1/2}$. The authors wrote that (e.g. Fig. 2b) '...indicate fold change in time at which % glutamate released is half of the maximum ($T_{1/2}$)'. If the authors meant to compare the thermodynamical efficiencies among different groups, the $T_{1/2}$ value may overlap with fold

change in maximum percent (%) and be meaningless. If the authors meant to compare the kinetic efficiencies, there are more appropriate ways, e.g., fit the scatters with exponential association functions and use kinetic parameters to interpret the data. Additionally, in Fig.5e, it is clear that the 3 traces displayed multiple exponential kinetics, it is hard to conclude whether Munc18-1/Munc13-1 accelerate C-terminal assembly kinetics or simply facilitate SNARE pairing. The authors are obligated to check the above issues and carefully interpret the data since these may be important to the central conclusion.

15. A critical concern: in Fig.3, the authors concluded that 'Munc13-1 stimulated the pore to interconvert between two different open states, a smaller conducting O1 and a larger conducting dilated O2 state.' and 'These pores, however, were restricted to a low-conducting narrow conformation. Munc18-1 also rescued the pores from their DAG mediated inhibitory state and the pores were significantly larger in size in comparison to those triggered open by Munc13-1.' However, in Fig. 6e and g, the pore sizes stimulated by Munc13-1 before Munc18-1 addition were literally larger than that shown in Fig.3, and Munc18-1 addition did not induce any change in sizes. This seems to be an apparent contradiction. Please check and comment on the issue.

16. Also, I noticed that the data of Fig.3g (green scatters) and Fig.6g (green scatters) are identical if I am not mistaken. In my view, the data in Fig.6e is independent to that in Fig.3a, therefore, the statistical data should be different. Please carefully check the issue and present reasonable explanations.

17. In Fig.S7a, the authors concluded that 'The Ca²⁺ addition downstream to Munc13-1 had no appreciable effect on stabilizing the pore open states.' According to the CDF curve, Ca²⁺ addition after Munc13-1 even decrease the open state dynamic of the fusion pore. In contrast, in Fig.S7b, the authors concluded that 'The Ca²⁺ addition downstream to Munc18-1 shifted the multi exponential kinetics of pore opening to largely single exponential (Fig. S7b), suggesting a role of Ca²⁺ in stimulated fusion pore opening.' Judging from the data, Ca²⁺ addition after Munc18-1 also decrease the open state dynamic of the fusion pore. Theoretically, if I am not mistaken, nanodisc-bilayer fusion would never produce a post-fusion state as liposome-bilayer fusion. Thus, it is unlikely that Ca²⁺ addition leads to a post-fusion state. Then, why Ca²⁺ addition decrease the open state dynamic of the fusion pore? Munc13-1 C2B domain is the only one that could response to calcium ions in this system, is Munc13-1 C2B responsible for the effect? If so, what is the underlying mechanism? Would disruption of calcium-binding sites of C2B rescue the effect? The authors also interpreted that the 'largely single exponential kinetics' suggests 'a role of Ca²⁺ in stimulated fusion pore opening', which puzzles me a lot. Could the 'kinetics' of close dwell time CDF trace be used for interpreting pore opening/dilation kinetics?

18. I do not know why the authors explored the effect of Ca²⁺ here in the absence of the calcium sensor Synaptotagmin-1. Given that Munc13-1 C2B domain is proposed to be a potential calcium sensor (Xu et al., 2017, PMID: 28177287), physiological fusion pore opening and dilation may be dominated by Synaptotagmin-1. Please comment on this.

19. The major conclusion that Munc18-1 and Munc13-1 function in fusion pore opening and dynamics largely relies on the system build on DAG-inhibitory effect. Unfortunately, the underlying mechanism of DAG-inhibitory effect was not fully addressed by the authors. Therefore, additional data were needed to confirm the effect of Munc18-1 and Munc13-1 in fusion pore opening and dilation. Given that the fusion pores were efficiently and stably formed without DAG and PIP2, the authors may utilize a C-terminal truncated SNAP-25 ($\Delta 9$, BoNT/A cleaved product) to clamp the final fusion (Lu et al., 2015, PMID: 26157630; Li et al., 2018, PMID: 29485200). In fact, similar function(i.e., driving the terminal stage of SNARE-dependent membrane fusion) was found in HOPS-regulated vacuolar SNAREs-mediated fusion (Song et al., 2021, PMID: 33944780; D'Agostino et al., 2017, PMID: 29088698). The authors need to

discuss these analogous results in the discussion section.

20. Finally, I would express my concern about the title. The title was partially overstated and not clear. On one hand, it feels like that the authors analyzed multiple SNARE chaperones more than Munc18-1 and Munc13-1, but in fact, they mainly focused on Munc18-1 and Munc13-1; On the other hand, the authors did not fully resolve the 'activity' of Munc18-1 and Munc13-1 in the fusion pore formation during SNARE-mediated membrane fusion, thus, to say '...activity determines...' should be inappropriate. The authors may utilize words that reflect their ground truth data, instead of making the title vague.

Minor concerns:

1. The fractions (M1 and M2) of Manders' colocalization coefficient are not very friendly to the readers and may cause potential misunderstandings. The authors are suggested to use 'A to B' and 'B to A' to interpret the colocalization data.
2. The CDFs of closed state dwell times were not shown for different concentration of Munc18-1 in Fig. 4. Please check.
3. Some of the representative traces were missing (e.g., Fig.5b, the traces for ND3S; Fig.5c, the trace for PS/PE/PC; Fig.5g, all; Fig.S2b, all; Fig.S5c, the traces for ND25L; Fig.S7, all). It is important to present raw representative traces to meet the requirement of transparency process.
4. In pg.8, texts cited Fig.5h, however, Fig.5h could not be found in the figure c

Reviewers' comments:

Reviewer #1 (Remarks to the Author):

The authors study the roles of Munc13 and Munc18 proteins on SNARE mediated membrane fusion. Although they make some potentially interesting observations, many of their results are hard to explain and/or inconsistent with basic expectations. I do not think this paper should be published anywhere.

Here are some more details:

1) The results are presented in a strange way, giving the wrong impression that the authors studied fusion pores in neurons. There is no mention of the main methodology used until the second subsection in Results.

>> Because fusion pores are not only formed in neurons but in many cell types, in our ms we introduced fusion pores in general. In our in vitro experiments, all the SNAREs and Munc isoforms used are of neuronal origin; hence we wanted to investigate how these SNAREs' localization with Munc proteins is affected by exogenous phorbol ester treatment of neurons.

To resolve the issue, we have modified the introduction section of the manuscript. The revised introduction now states:

"Here we traced the phorbol ester dependent localization of both Munc13-1 and Munc18-1 with the t-SNARE SNAP-25B, syntaxin1a and v-SNARE syb2 in the presynaptic terminal of rat cortical neurons. To test the impact of these Munc proteins in functional fusion pore assembly, we used recently described in vitro ND-BLM reconstitution technique^{13,43,44} and studied the μ -second dynamics of individual nascent fusion pores."

2) The only experiments with rat cortical neurons are studies of colocalization of SNAP25 with Munc13 or Munc18 in the presence or absence of the DAG mimic PMA.

It is not clear why SNAP25 is chosen for these experiments. The obvious choice would have been Syntaxin, since both Munc13 and Munc18 interact with it.

>> The rationale behind choosing SNAP25 was that SNAP25 is a cytosolic protein, unlike Syntaxin. Previous reports also indicated the role of Munc13-1 in recruiting SNAP-25B at the release site in the presynaptic terminal (PMID: 37343100, 33222163). In our experiments, SNAP25's localization with the Munc13-1 and Munc18-1 (both the Munc proteins are also cytosolic) would mean the presence of Munc proteins associated with the SNARE complex assembly at the presynaptic terminal.

To resolve the issue, in the revised ms, we have now incorporated the co-localization data of both syntaxin and syb2 with the Munc proteins, at the presynapse. The revised data is shown below for ready reference:

Figure 1.

Fig. 1: SNAP-25B co-localizes with SNARE chaperones in a phorbol ester dependent manner

a, Left panel, representative confocal image of primary cortical neuron that were triple labeled for synapsin (Alexa 647-conjugated secondary antibody), SNARE proteins (Alexa 555-conjugated secondary antibody: SNAP-25B (top), syntaxin1a (middle), syb2 (bottom) and Munc13-1 (Alexa 488-conjugated secondary antibody). The overlay image shows regions where Munc13-1 coincided with individual SNAREs, as indicated. Blue arrows: co-localization; while arrows: no co-localization. Right panel, representative confocal images of PMA treated primary cortical neuron, under indicated conditions. **b**, The box plots showing Manders correlation co-efficients M1 (Munc13-1 co-localizing with individual SNAREs as indicated) and M2 (individual SNAREs' co-localizing with Munc13-1) **c**, The representative confocal images of primary cortical neurons that were triple labeled as described in **a**, for Munc18-1 (Alexa 488-conjugated secondary antibody), with and without treating the neurons with PMA. Data are presented as mean \pm SEM from neurites of 6-7 individual cortical neurons for each of the conditions mentioned; the Student's T test was performed to compare the two means; * $p < 0.05$, ** $p < 0.01$, n.s. - non-significance.

Importantly, small changes in colocalisation cannot be taken seriously in the absence of controls. How specific are the antibodies?

>> The antibodies used for this set of experiments are very specific. We performed immunoblot on the neuronal lysates (treated with PMA) to resolve the issue, shown below for the ready reference:

Figure: Immunoblots for syb2 (left) syntaxin-1a (middle) and SNAP-25B (right), using the neuronal lysates treated with and without PMA. The antibody dilutions used are as follows – syb2 (1:2000), syntaxin-1a (1:2000), and SNAP-25B (1:2000).

How does PMA affect antibody labeling? Does it induce similar minor changes in colocalization of unrelated proteins? By how much?

>> The effect of PMA on colocalization of an unrelated protein Hsp60 has been included in the revised Fig.1, which is shown below for the ready reference.

Fig. S1

Fig. S1: Co-localization of Hsp60 with Munc13-1 and Munc18-1 in rat cortical neurons

a, Representative overlay confocal image of neurites that were triple labeled for synapsin (Alexa 647-conjugated secondary antibody), Hsp60 (Alexa 555-conjugated secondary antibody), and the SNARE chaperone Munc13-1 (upper panel), Munc18-1 (lower panel) (Alexa 488-conjugated secondary antibody), in the absence (left) and presence (right) of the phorbol ester PMA. Blue arrows: co-localization; while arrows: no co-localization. **b**, The scatter plots showing Manders correlation coefficients M1 and M2 for Hsp60's co-localization with Munc13-1 (left) and Munc18-1 (right), under indicated conditions. Data are presented as mean \pm SEM from neurites of 6-7 individual cortical neurons for each of the conditions mentioned; the Student's T test was performed to compare the two means; n.s. - non-significance.

As the authors note, "Because Munc18-1 doesn't possess any phorbol ester binding sites, it was unclear why this Munc protein's co-localization with SNAREs would be phorbol ester dependent."
Note that DAG is a surfactant and DAGs are used as emulsifiers. It is not surprising their use may cause slight changes in antibody labeling.

>> Please see the above experiments, which contradict the reviewer's interpretation.

Finally, there is no clear link between these colocalization studies and the fusion pore measurements reported later in the paper.

>> The above colocalization studies were done to support our in vitro findings that Munc13-1 and Munc18-1 can be localized with the SNAREs within neurons. The detailed functional consequence of that co-localization was studied in the manuscript, as to how these Munc proteins alter SNARE complex assembly, fusion pore organization etc.

If it is deemed necessary, we can move the data to supplementary.

3) There are many statements that are hard to understand. For example, p.4, penultimate paragraph: "The above Munc13-1 actions were Ca²⁺ independent, as all the experiments were done in presence of BAPTA (Fig. S2a)." How can you deduce the effect is calcium independent if calcium is not present? I suppose the authors mean to say the effect did not require calcium.

>> We have revised the sentence, which now reads:

"The above Munc13-1 actions did not require Ca²⁺, as all the experiments were done in presence of BAPTA (Fig. S2)."

4) In the nanodisc-liposome fusion experiments, controls for leakage or liposome aggregation are missing.

>> We wanted to point out that these control experiments were already done in the lab, as this assay has been published previously by us and others (PMID: 34413185, 29420480, 26656855). However, to resolve the issue, we are providing below one such result, using ND0 as a control, for the reference of the reviewer.

Figure: Representative time course of % glu release, using t-SNARE liposomes (lipid composition - PS/PE/PC/PIP2/DAG) and either ND5 or ND0, as indicated. The ND0 means NDs bearing no syb2.

It is to be noted that none of the Munc proteins has been reported to cause liposome leakage in the bulk lipid mixing or content release experiments (PMID: 33468652, 34779770).

Please see below (response #11 of reviewer #3), where we have provided ND-BLM data to show the effect of only Munc13-1 or Munc18-1 in the BLM membrane. None of the proteins cause any detectable change in current through the BLM membrane.

Fig. 2 a: please try to draw the schematics to scale. Is there really enough space to accommodate all the proteins in the ND?

>> The schematic has been modified as per the suggestion, shown below for ready reference.

Fig.2: a, Left panel, illustration shows the glutamate release assay through fusion pores as ND5_S interacts with the glutamate entrapped t-SNARE liposomes. The glutamate released into the reaction medium is measured by an increase in the fluorescence signal of the glutamate sensor (iGluSnFR).

Please note – the radius of SNARE TMD is ~0.35 nm (PMID: 20512644) and ND radius is ~6.9 nm (the AFM image for that is provided below for ready reference). From the simple calculation, ND surface area is ~150 nm² and the SNARE TMD surface area is ~0.38 nm². The ND surface area is ~400 times larger than a single SNARE TMD. Hence, there is space in the ND to accommodate the indicated number of SNAREs.

Figure legend. Left, middle panels - representative AFM images showing the morphology and size of the nanodiscs (ND5_S and ND25_L). Right panel - size (diameter) distribution (from AFM images) of the nanodiscs. Data analysis was performed by the JPK Data Processing software. The height profile of all individual objects (NDs) were obtained by manually drawing vertical lines along them. These cross-section profiles were fitted with Gauss equation in Origin software and FWHM (Full Width Half Maximum) for each individual NDs was obtained. The size (diameter) distribution of the NDs was plotted (right panel). n = 100. Error bar is S.D.

The plot of "fold change" is meaningless, as GLu release from the PS/PE/PC/PIP2/DAG liposomes continues almost linearly after release from the PS/PE/PC liposomes has completely leveled off. That is, if one waits past ~4000 s, there would be more release from the former.

>> The purpose of this assay was to check the effect of membrane lipids PIP2/DAG on content release. To resolve the issue, in the revised manuscript we have modified the "fold change" plot to "half maximum of % glu release". The revised figure is provided below for ready reference.

Fig. 2a (right panel): Box plots showing pooled results to indicate half maximum percent (%) of glutamate (glu) released during fusion reaction between ND5_s and glutamate entrapped t-SNARE liposomes, in absence and presence of PIP2/DAG lipids in t-SNARE liposomes.

Is the use of student's t-test justified? (are the data normally distributed?)

>> We have performed Q-Q plot, provided below, to show that the content release data are indeed normally distributed.

5) The main results are obtained using an assay in which nanodiscs (NDs) are fused with planar suspended bilayers. This assay is sensitive to single fusion pores and has very good time resolution.

DAG is found to shorten pore lifetimes. This is not surprising, given that DAG is a surfactant with a negative curvature. It is expected to shorten pore lifetimes (e.g. see Karatekin...Brochard-Wyart, Biophys. J. 2003).

>> Please note - **the paper cited has no relation with the SNARE-catalyzed membrane fusion and hence is not relevant to our current study.** We are confused, about what the reviewer meant - ‘.shorten pore lifetimes’, while the reference has no SNAREs at all.

The referred paper used only DOPC lipid, whereas we have a totally different lipid composition, hence this comment of the reviewer stands irrelevant.

More importantly, **the referred paper does not use DAG, while the reviewer assumed that the effect of DAG and a surfactant (they used tween 20) on the bilayer membrane would be the same.** DAG is constantly being produced inside the cell by the phospholipase C family members, which makes it highly physiologically relevant to check the direct effect of DAG in fusion pore modulation.

6) Adding Munc13 restores longer pore lifetimes to some extent. Munc13 has a DAG binding C1 domain. A Munc13 mutant that is unable to bind DAG (H567K) is expected to have NO effect on pore dynamics compared with pores induced by SNAREs alone. Yet, the mutant has an even stronger effect on pore conductance than WT Munc13, and causes longer pore lifetimes compared to SNARE-alone pores. It is completely unclear what to make of these results.

>> We are a little confused with the following statement of the reviewer - “A Munc13 mutant that is unable to bind DAG (H567K) is expected to have NO effect on pore dynamics compared with pores induced by SNAREs alone.” **It is well established that the MUN domain of Munc13-1 interacts with the SNAREs,** even if the DAG binding activity is lost in the mutation. **The other functional domains of Munc13-1, like C2B and C2C, are also active.** Please note that the SNARE binding ability of Munc13-1 might be critical here, that was one motivation to perform this set of experiments. Please see the sentence - “It was, however, unclear whether Munc13-1’s lipid and/or SNARE binding properties were responsible for the above pore phenotype.”

In the revised ms, we have also incorporated data for Munc13-1’s SNARE binding mutation. The relevant text and data are provided below for the ready reference:

“Because Munc13-1 interacts with both the membrane lipids and SNAREs, next, we explored whether its SNARE binding ability also contributes to the fusion pore regulation. We mutated MUN domain of Munc13-1 (residues F1234, K1236) yielding Munc13-1(FKAA), which disrupts its interaction with the syntaxin1a N-terminus^{54,55}. Munc13-1(FKAA) interacts equally well with the membrane lipids as WT Munc13-1 (Fig. S6) but did not yield two open states of the fusion pores, as observed in case of WT (Figs. 3i, j). The rate of opening of these pores also reduced in comparison to the pores triggered open by WT Munc13-1 (Fig. 3k). Hence, Munc13-1 can access SNAREs in the dynamic trans-SNARE complex, as fusion pores functionally assemble at the membrane fusion site.”

Fig. 3: i, Representative trace of single ND_{5s} pore triggered open by Munc13-1(FKAA), BLM lipids used is mentioned above the trace. Three epochs (beginning, middle and end) of the trace are shown. Closed (C) and open states (O) of the individual pores are indicated with the respective currents. j, Current histogram of the ND_{5s} pore formed in the presence of Munc13-1(FKAA). Closed (C) and open (O) states are indicated by red and blue arrows respectively. k, CDF of closed state dwell times for indicated experimental conditions. n = 3 independent BLMs; two sets of NDs were used.

Similarly, it is not clear how to interpret the results about SNARE copy numbers and the effects of calcium.

>> To resolve the issue regarding SNARE copy numbers, we have performed a series of additional experiments with the cytoplasmic domain of syb2 (cd-syb2). The relevant text is provided below for the ready reference. The revised ms does not contain calcium data, as that does not affect the central conclusion of this ms.

“We speculated whether Munc13-1 mediated cooperative clustering of trans-SNARE complexes assists in fusion pore opening. We performed a competition experiment between Munc13-1 and cytoplasmic domain of syb2 (cd-syb2)^{13,43}, each of which independently alters open state stability of individual pores by accessing the dynamic trans-SNARE complex^{13,43}. In the ND-BLM assay, after Munc13-1 triggered the ND_{5s} pores to yield stable open states (membrane lipid composition – PE, PC, PS, PIP₂ and DAG), a ~20 μM cd-syb2 was unable to show any alteration in the pore properties (Fig. S10). When we omitted Munc13-1 from this reaction (membrane lipid composition – PE, PC, PS), the same [cd-syb2] significantly destabilized the pore open states and closed the individual pores (Fig. S10), also shown previously^{13,43}. Munc13-1’s presence organizes more trans-SNARE complexes at the release site, hence the same [cd-syb2] was insufficient to titrate out all the trans-SNARE complexes. We reduced the syb2 copy number in NDs, yielding ND_{3s}, and observed that ~20 μM cd-syb2 was sufficient to yield long pore closures in presence of Munc13-1 (membrane lipid composition – PE, PC, PS, PIP₂ and DAG) (Fig. S10). This set of experiments showed importance of Munc13-1 mediated clustering of SNARE complexes in yielding stable open pores. To get additional insight, in a separate set of experiments we mutated Munc13-1 residue 1358, yielding Munc13-1(D1358K), which perturbs its oligomeric organization as well as its interaction with syb2⁵⁹⁻⁶¹. This mutated Munc13-1 was unable to trigger stable opening of fusion pores from the DAG induced inhibitory state (Fig. S11), indicating the importance of Munc13-1’s oligomerization in yielding functional fusion pore assembly^{59,60}.”

Fig. S10

Fig. S10: a-c, Representative traces of ND5_s and ND3_s pores, in absence and presence of Munc13-1, membrane lipid composition are mentioned. The addition of different [cd-syb2] is indicated by the arrow. The corresponding current histograms from traces shown in (a) are plotted. The close (C) and open (O) states are indicated. N = 3 independent BLM recordings.

7) The effects of Munc18 are most simply explained by Munc18 (a soluble protein) transiently blocking the pores. As the Munc18 concentration is increased, the probability of the block increases. Many controls would need to be made to exclude the possibility of Munc18 blocking pores.

>> Here we would like to first refer to a series of recent ND-BLM studies (from our laboratory and others, published in well-reputed journals including *Nature*, *Nat. Comm.*), which clearly demonstrated the pore modulatory action of membrane fusion regulators, through their interaction with the dynamic trans-SNARE complex (PMID: 34413185, 29420480, 31932584). We presume Munc18-1 affects the pore properties similarly.

To resolve this issue, **we have now used a Munc18-1 mutant that does not alter pore properties as observed by the WT Munc18-1.** That data is provided below for ready reference, along with the revised text.

“To further confirm whether this action of Munc18-1 was indeed through its interaction with SNAREs in the dynamic trans-SNARE complex, we mutated Munc18-1 residue that alters its interaction with syb2, yielding Munc18-1(D326K)⁵⁶ (Figs. 4e-g). We analysed the close state dwell time CDFs for WT and mutant (Fig. 4g, Table S2). This mutation significantly increased the rate of pore opening compared to WT. Hence, Munc18-1 accessed dynamic trans-SNARE complexes to regulate the fusion pore properties.”

Fig. 4: e, Representative trace of single ND5_s pore triggered open by Munc18-1(D326K), BLM lipids used is mentioned above the trace. Three epochs (beginning, middle and end) of the trace are shown. Closed (C) and open states (O) of the individual pores are indicated with the respective currents. f, Current histogram of the ND5_s pore formed in the presence of Munc18-1(D326K). Closed (C) and open (O) states are indicated by red and blue arrows respectively. g, CDF of closed state dwell times of pores formed during indicated experimental conditions. n = 3 independent BLMs; three sets of NDs were used.

8) The suspended bilayer method is an interesting approach, but the presence of the solvent (n-decane here) dramatically affects membrane properties (e.g. see Santinho...Thiam, *Biophys. J.* 2021, Campillo...Sykes, *Biophys. J.* 2013; Melikov...Chernomordik *Biophys. J.* 2001). Thus one needs to be very careful in interpreting results, especially when results are unexpected and hard to explain.

>> Thanks for the suggestion. Please note – at least one author (if not more) of this ms has demonstrated the ability to design, analyze, and report the ND-BLM data in journals of good repute (PMID: 34413185, 29420480, 31932584, 33582136). In none of the previous reports we observed any detectable change of measured currents due to solvent, which was n-decane in all those studies.

More importantly, two of the three references provided by the reviewer (Santinho...Thiam, *Biophys. J.* 2021, Campillo...Sykes, *Biophys. J.* 2013) used absolutely different methods for artificial membrane formation than what we have used in our current ND-BLM recordings; hence those references are irrelevant. The third reference (Melikov...Chernomordik *Biophys. J.* 2001) showed the effect of voltage in pore formation using BLM, which is also irrelevant in our case, as we use only one voltage in our entire study, which does not alter the BLM properties.

9) There are many self citations, but other key literature is omitted. For example, increased pore size as a function of increasing SNARE copy numbers was reported well before the report by Bao et al *Nature* 2018, using nanodisc-liposome fusion (by the Rothman lab) and nanodisc-cell fusion (Karatekin lab) assays.

>> Please note – the Rothman lab paper using ND-liposome fusion (PMID: 22422984) has already been cited. To resolve the issue, we have now included the Karatekin lab paper in the revised manuscript. (PMID: 28346138).

Reviewer #2 (Remarks to the Author):

The manuscript entitled "Differential SNARE chaperoning activity determines quantal size during membrane fusion", by Bhaskar et al. provides the functional evidence of SNARE chaperons Munc13-1 and Munc18-1 in rescuing nascent fusion pore from DAG-mediated inhibitory states. Munc13-1 and Munc18-1 differentially regulate pore size and kinetics. Munc13-1 restricts the pore in a narrow low conductance state while Munc18-1 triggers pores in their dilated large conductance states. They also provided data showing that

Munc13-1 and Munc18-1 allow a cooperative zippering of SNARE from N- to C- terminal and assist in synchronizing content release during membrane fusion, indicating the crucial role of Munc proteins in modulating fusion pore assembly.

Addressing the following concerns would improve the study significantly.

>> We thank the reviewer for all the suggestions. We have now included all the suggestions in our revised manuscript. Please see our response below.

Major Points:

1. DAG (diacylglycerol) and PIP2 (phosphatidylinositol 4,5-bisphosphate) are consistently observed in nearly all experimental investigations mentioned within the manuscript. However, the literature review related to their significance is apparently lacking. Therefore, the association between DAG and PIP2 on SNARE proteins and fusion pore should be introduced.

>> We appreciate the reviewer's suggestion. In the revised ms, we have extensively edited the introduction and discussion section to include the association of DAG and PIP2 on SNAREs and fusion pores. The revised text is provided below for ready reference.

"The DAGs and PIP₂ are the minor components of cell membranes, and serve as the secondary messengers in diverse signalling pathways³². The exogeneous addition of DAG enhanced both spontaneous and evoked neurotransmission from neurons³³⁻³⁵. PIP₂ interacts with several Ca²⁺ sensors during neurotransmission³⁶⁻³⁸. It is, however, unknown whether these secondary messengers play a direct role in regulating fusion pore assembly during SNARE catalyzed membrane fusion."

2. In Figure 1, the author initially explored the organization of release sites by Munc13-1 and Munc18-1. However, it is essential to investigate the involvement of additional components of the fusion site, syntaxin. It is recommended to employ the same methodology to examine the co-localization of Munc proteins and syntaxin, shedding light on their potential collaborative roles in synaptic vesicle fusion.

>> Again, we appreciate the reviewer's suggestion. In the revised ms, we have included the colocalization data of syntaxin and syb2 with Munc proteins. Please note, the co-localization studies were performed to understand which of the SNAREs engage in the presynaptic membrane because of enhanced DAG signal. This might have implications at the very early stages of membrane fusion.

Please see response #2 of reviewer #1.

3. As depicted in Figure 1, it is difficult to observe the increase of SNAP25 after PMA treatment. In order to enhance visual clarity and provide a more detailed representation, it is recommended to replace the current image with a higher-quality version and enlarge specific details, as demonstrated in Supplementary Figure S1.

>> In the revised ms, we have replaced Fig. 1 with new high-resolution images.

4. In Figure 1c and 1d, the difference in Munc18-1/SNAP25B colocalization is much more significant than that of Munc13-1/SNAP25B, and at the end of the first paragraph of the Results, the authors also mentioned that: "Because Munc18-1 doesn't possess any phorbol ester binding sites....in a phorbol ester dependent manner", the experimental results reveal a novelty that warrants an extended discussion in the manuscript.

>> Thank you for pointing this out. In the revised ms, we have discussed phorbol ester-dependent co-localization of Munc18-1/SNAP25B. The revised text section is provided below for the ready reference of the reviewer.

"The PMA treatment in our experiments might also activate PKC (Protein Kinase C) pathways⁶³, which has been shown to phosphorylate Munc18-1^{64,65} which may not affect synaptic transmission⁶⁶."

5. The consistency between the data depicted on the left side of Figure 2b and the accompanying statistical graph in the middle appears to be questionable, suggesting a potential inconsistency in their depiction. To

address this issue, a more comprehensive explanation is needed regarding the methodology employed to generate Figure 2b. And it is better to plot the averaged traces of Figure 2a middle and Figure 2b left, respectively.

>> We apologize for the confusion. In the revised figure, we have kept one representative trace for "% Glu release" in Fig. 2a (middle) and Fig. 2b (left). In the accompanying statistical graphs, we are now showing the average data for half maximum % glutamate release for indicated conditions, along with SEM.

Fig. 2a (right panel): Box plots showing pooled results to indicate half maximum percent (%) of glutamate released during fusion reaction between ND5_s and glutamate entrapped t-SNARE liposomes, in absence and presence of PIP₂/DAG lipids in t-SNARE liposomes.

Fig. 2b (middle panel): Box plots showing pooled results to indicate half maximum percent (%) of glutamate (glu) released during fusion reaction between ND5_s and glutamate entrapped t-SNARE liposomes with membrane lipids PS/PE/PC/PIP₂/DAG, in absence and presence of SNARE chaperones.

6. In Figure 2c, the authors demonstrated that the presence of PIP₂/DAG in the BLM lipid exerted a significant impact on the stability of individual pores. However, it remains crucial to investigate how DAG, as an independent entity, influences the state of the fusion pore. The experiment may provide a more thorough understanding of the underlying mechanisms involved.

>> We agree with the reviewer. In the revised ms we have now included the single pore recording in the presence of membrane lipids - PE/PC/PS/DAG. The revised Figure is provided below for the ready reference of the reviewer.

Fig. S4

a. Membrane lipids: PS/PE/PC/DAG

Fig. S4: a,b, Representative raw traces of ND_{5s} pores (a) and the corresponding current histograms (b) are shown, for different [DAG] in the BLM, as indicated in the figure. N = 3 independent BLMs; two sets of NDs were used for each of the conditions.

7. In the experiment shown in Figure 4, an explanation for two concentrations of Munc18-1 that were applied is needed. And why is the Fraction-close of 5 μ M Munc18-1 lower than that of 10 μ M Munc18-1? These interesting observations require speculative explanations in the Discussion section of the manuscript. The authors can explore potential factors that may contribute to the observed Fraction-close differences. These may include changes in Munc18-1 binding affinity at different concentrations, potential allosteric effects on the fusion machinery, or differential interactions with other regulatory proteins involved in fusion pore dynamics.

>> Thank you for pointing this out. In the revised Figure 4, we have now included data for additional Munc18-1 concentrations, which shows a stoichiometric interaction of Munc18-1 with the SNAREs in functional fusion pore assembly. The revised Figure 4 and the associated text from the revised ms are provided below for ready reference.

“The in-depth kinetic analysis revealed a multi-exponential fusion pore closure driven by Munc18-1, yielding slow and fast kinetic components for pore closure (Fig. S7a and Table S3). The rate of pore closure increased as a function of [Munc18-1] (Fig. S7b). To better understand Munc18-1 mediated pore closure, we plotted the amplitudes of slow and fast kinetic components, after fitting the open dwell times with multiexponential function (Figs. 4d, S7a, Table S3). Interestingly, the fast components increased and the slow components decreased with the increase in [Munc18-1] (Fig. 4d). This set of experiments indicated that Munc18-1 stoichiometrically interacts with the multiple trans-SNARE complexes in fusion pore assembly.”

Fig. S7: a, Cumulative distribution functions (CDF) of open dwell times for indicated Munc18-1 concentrations. $n = 3$ independent BLMs; two to three sets of NDs were used. b, The open state CDFs from a, were fitted with three exponential equations and the slowest kinetic rate constants ($k_{\text{closure(slow)}}$) are plotted as a function of [Munc18-1]. The data was fitted with a Hill equation and the derived parameters n (Hill coefficient) and EC_{50} are shown.

Fig.4: d, Stacked column plot showing amplitude fraction of fast and slow rate of pore closures derived from exponential fitting of open dwell time CDFs of pores triggered open by different [Munc18-1] as indicated.

8. For Fig 5a and Fig S5a and S5b, the methods for immunoblots of SNARE complex and individual proteins were not described clearly. How is the fraction of liposome-nanodisc reaction processed for blotting? Please revise the method section to make it easier to follow. Is it native PAGE for the detection for SNARE complex? Please clarify it.

>> We apologize for the confusion. In the revised ms, we have thoroughly edited this method section, which is provided below for ready reference.

"Steady state SNARE complex formation assay

t-SNARE liposomes (5nM) and NDs (100-150nM) were incubated with different concentrations of Munc13-1 or Munc18-1 (as indicated) in reconstitution buffer (25 mM HEPES-KOH pH-7.4, 100 mM KCl, 2mM β -mercaptoethanol) overnight at 4°C. The overnight reaction samples were split into two fractions to probe for SNARE complex and for using as heated controls. For SNARE complex blots, the overnight reaction samples were mixed with 1X Laemmli sample buffer without heating and SDS-PAGE was run followed by immunoblotting with SNAP-25 antibody for quantifying SNARE complexes. For control blots, the reaction samples for SDS-PAGE and immunoblotting were prepared by heating at 95°C for 15 minutes to disrupt SNARE complexes formed and to check individual levels of t-SNAREs, Syb2 and Munc13-1 or Munc18-1. The

blots were analyzed using ImageJ software and the band intensity of SNARE complexes formed was quantified under indicated conditions. The graph was plotted as fold change in band intensity to its corresponding Munc13-1/Munc18-1 concentrations. Fitting of the data was done using the modified Hill equation where y_{min} is the minimum SNARE complex formed, y_{max} is the maximum SNARE complex formed, k is the EC_{50} and n is the Hill coefficient:

$$y = y_{min} + \frac{x^n}{k^n + x^n} (y_{max} - y_{min})$$

For the SNARE complex detection, it is not enough to only show SNAP-25 band and re-blots for Munc13-1/Munc18-1/Syb2/Syntaxin-1a are quite necessary to demonstrate the composition of the complex.

>> We appreciate the reviewer for this suggestion. Please note that our immunoblots are from SDS-PAGE; because our SNARE complexes molecular weight does not change in SDS-PAGE after Munc13-1/Munc18-1 treatment, we believe Munc proteins' interaction with SNARE complexes are not SDS-resistant.

To resolve the issue, we have now probed the blots for both SNAP-25B and syb2. The syb2 immunoblot is provided below for the ready reference of the reviewer.

Figure legend: Immunoblot showing the gradual increase of SNARE complexes (indicated by red arrow), as a function of increasing [Munc13-1]. Antibody used syb2, also mentioned. Molecular weight (in kDa).

Besides, the ladders for protein size should be labeled in blots for the detection of both complex and individual proteins.

>> In the revised ms, we have now included the molecular weight marker as suggested by the reviewer.

In addition, why is the concentration of Munc13-1 and Munc18-1 indicated in blots different from that in plot for band intensity measurement? Please explain it.

>> We again apologize for the confusion. In the revised ms, we have now included better representative blots with the band intensity measurements, to remove the inconsistency regarding concentration of Munc proteins. The revised Fig. is provided below for ready reference.

Fig. 5: a, Representative immunoblot showing trans SNARE complexes during t-SNARE liposomes' fusion with ND5_s with increasing [Munc13-1] (top, left) and [Munc18-1] (top, right). Scatter plots fitted with modified hill equation showing fold change in band intensity for trans-SNARE complexes formed as [Munc13-

1] (bottom, left) and [Munc18-1] (bottom, right) increases. The hill coefficient (n) and EC_{50} (k) are indicated in the plot. The membrane lipid composition for the t-SNARE liposomes was PS/PE/PC/PIP₂/DAG.

9. As depicted in Figure 3a, Munc13-1 stimulated the pore to interconvert between two different open states, O₁ and O₂, and the Current between -6 to -9 pA. But in Figure 6e and 6f, only one open state of Munc13-1 was shown, and the Current between -9 to -13 pA. The data looks inconsistent. Please give a reasonable explanation.

>> Thanks for pointing this. We apologize for not choosing representative epochs from the raw traces in Fig.6e. In the revised Figs. 6e and 6f, now the pore interconvert between O₁ and O₂ states. The revised Figs. 6e and 6f are provided below for ready reference.

Fig. 6: e, Representative trace of single ND_{5s} pore triggered open by Munc13-1 (green arrow), addition of Munc18-1 to that pore is indicated (brown arrow). Closed (C) and open (O₁ and O₂) states of the individual pores are indicated, with their respective currents. f, Current histogram of the pore shown in e. Closed (C) and open (O₁ and O₂) states are indicated by red and blue arrows respectively.

In all the figures the current histograms are shown only for the representative raw traces, shown in the figures.

Please also note – we have plotted conductance of all the Munc13-1 pores in Figure 3G, which does show trace-to-trace variation in the currents. That, however, does not change the central conclusions from this figure.

10. Many groups of data were collected from only 3-6 BLMs, which are a bit risky considering that variations can be large. For example, the group of Munc18-1 in Figure 2b right and the group of Munc13-1 in Figure 4e show large variation. I suggest that the authors increase the cell number to enhance the significant difference.

>> In the revised ms we have now – 1) included additional data in Figure 4e; and 2) we have modified the Figure 2b. The revised Figures are shown below for ready reference.

Fig. 4: c, Scatter plot shows comparison of pore conductances between pores triggered open in the absence and presence of SNARE chaperones Munc13-1 and Munc18-1. N = 4 to 14 independent BLMs; three or more sets of NDs were used for each experimental condition. Data are presented as mean \pm SEM; Student's T test was performed to compare the two means; **p < 0.01, ***p < 0.001.

Fig. 2b (middle panel): Box plots showing pooled results to indicate half maximum percent (%) of glutamate (glu) released during fusion reaction between ND5S and glutamate entrapped t-SNARE liposomes with membrane lipids PS/PE/PC/PIP2/DAG, in absence and presence of SNARE chaperones.

11. The part on the molecular mechanism of the pore regulation by Munc13-1 and Munc18-1 is still weak. Following points need to be addressed: It's important to know which core domain they use to control the size and the open state stabilities of the fusion pore, therefore, some experiments for mutations of Munc13-1 and Munc18-1 need to be presented.

>> We appreciate the reviewer for this comment. To resolve the issue, in the revised ms, we have now included a series of Munc13-1 and Munc18-1 mutation data, along with additional new experiments to obtain insight into the molecular mechanism of the process. The relevant Figures and the text are provided below for ready reference of the reviewer:

"When both the Munc proteins are present at the active zone with t-SNAREs and the v-SNARE containing donor membrane approaches, the two Munc proteins differentially assemble trans-SNARE complexes to directly regulate fusion pore opening (Fig. 7). The localized presence of DAG defines Munc13-1's site of action in the active zone.

Munc13-1 clustered multiple trans-SNARE complexes at the release site (Fig. 7). This action of Munc13-1 presumably is dependent on its self-oligomerization^{59,60} (Fig. 7), which stimulates stable opening of fusion pores from DAG mediated inhibitory states (Fig. 7). The inhibitory action of DAG could be a consequence of its ability to alter membrane curvature. We mutated the Munc13-1 residue (D1358) which has been previously implicated in its self-oligomerizing behaviour⁵⁹ and was responsible for its interaction with the C-terminal juxtamembrane domain of syb2 to catalyze trans-SNARE complex formation⁶⁰. This mutated Munc13-1 was unable to stabilize the pores in their open conformation. Munc13-1 also interacts with the N-

terminal regulatory domain of syntaxin1a⁵⁵. When we perturbed this interaction in Munc13-1(FKAA) mutation, it significantly reduced the stable opening of pores. Munc13-1 altered the required number of cd-syb2 molecules to titrate out the dynamic trans-SNARE complexes, further indicating this Munc protein's importance in clustering multiple trans-SNARE complexes (Fig. 7). Hence, Munc13-1's action on individual SNAREs and on trans-SNARE complexes play crucial roles during fusion pore assembly. Munc13-1 mutation (H567K) that reduces its DAG binding ability also showed significantly reduced open state stability of the pores, further indicating the importance of DAG mediated inhibitory-stimulatory mechanism in controlling pore properties.

The SNARE chaperone Munc18-1 stoichiometrically interacts with trans-SNARE complexes, and stimulates individual SNARE complexes' zippering from N- to C-terminus. This action of Munc18-1 overcomes the inhibitory effect of DAG to trigger pore opening into a more stable open state. However, Munc18-1 triggered pores still showed long closures, presumably due to a dynamic open/close transition of SNAREs within the fusion pore assembly. This Munc protein has been shown to disassemble t-SNARE heterodimers and stabilized the closed conformation of syntaxin1a^{41,67}. This could lead to transient SNAP-25B displacement from the SNARE complexes, hence resulting in long pore closures (Fig. 7). The Munc18-1 mutation (D326K) that showed reduced SNAP-25B displacement⁶⁷, showed significant enhancement in the open state stability of the pores, with the appearance of multiple open states. The presence of Munc13-1 along with Munc18-1 at the release site allows SNAP-25B to stably associate with the trans-SNARE complex, to yield kinetically stable and structurally large pores.

The amount of cargo released during membrane fusion is dependent on both the size and the duration of fusion pore opening, which has ramifications in differentially activating the downstream receptors. The two crucial SNARE chaperones Munc13-1 and Munc18-1 control both the pore properties, as revealed in our current study. How these properties dictate the cellular physiology under normal and pathological conditions, will be the subject of future studies."

Also, please see the response #6, 7, of reviewer #1.

Fig. S11

Membrane lipids: PS/PE/PC/PIP₂/DAG

Munc13-1 D1358K

Fig. S11: Representative trace of ND5_s pores where Munc13-1(D1358K) was added at the beginning of pore opening. The close (C) and open (O) states are indicated. N = 3 independent BLM recordings.

Minor concerns:

1. In Figure 2b middle and right, the label for Munc13-1 and Munc18-1 should be consistent within Figure 2b left.

>> We have modified the Figure 2b in our revised manuscript. The revised Figure is shown below.

Fig. 2b: Left, representative time course of percent (%) glutamate (glu) released during fusion reaction between ND₅s and glutamate entrapped t-SNARE liposomes with membrane lipids PS/PE/PC/PIP₂/DAG, in absence and presence of SNARE chaperones (Munc13-1 and Munc18-1). Middle, box plots showing pooled results to indicate half maximum percent (%) of glutamate (glu) released for the indicated conditions. Data are presented as mean \pm SEM from ten independent trials under each of the conditions mentioned; three independent sets of NDs were used. Student's T test was performed to compare the two means; * $p < 0.05$

2. In Figure 4c, the concentration of Munc18-1 and Munc13-1 need to be stated in the legend.

>> We apologize for this mistake. The Figure 4 has been modified and the legend has been updated now in the revised ms.

3. The scales bar of the current trace in Figure 3a is not consistent with that in Figure 3e, is it possible for the author to make all the scales bar consistent in all the figures.

>> In the revised ms, the current scale bars are now consistent in all figures.

Reviewer #3 (Remarks to the Author):

Munc18-1 and Munc13-1 are well-established components that organize SNARE complex assembly and neurotransmitter release in neurons and neuroendocrine cells. A great amount of data has shown that Munc18-1 and Munc13-1 cooperatively chaperone SNARE complex assembly. However, it is still enigmatic whether the two chaperones would function in opening and dilation of fusion pores during the last step of SNARE-mediated membrane fusion. In this manuscript, by using previous established nanodisc-planar lipid bilayer electrophysiology approach, Bhaskar et al. revealed that Munc18-1 and Munc13-1 could individually rescue nascent fusion pores from their DAG-mediated inhibitory states with different mechanisms. Meanwhile, the presence of both Munc18-1 and Munc13-1 would further stabilize the fusion pore size and open-close dynamic, illustrating that they cooperatively orchestrate the final step of SNARE-mediated membrane fusion. Taken together, the authors presented interesting and potentially important data for the function of Munc18-1 and Munc13-1 in SNARE-mediated membrane fusion. However, there still exists numerous imperfections and defects that hinder the manuscript from being published. Overall, the manuscript may need to undergo an intensive revision to meet the requirement of a high-quality work. Some critical concerns are listed below.

Major concerns:

1. Despite the diacylglycerol-like phorbol ester PMA is commonly applied to inducing membrane localization of Munc13-1, the authors should acknowledge that PMA would also activate PKC pathways. In other words, PMA treatment would cause a variety of cellular effect, e.g., vesicle transportation and protein expression (though the treatment was lasting only 15 min as indicated by the authors).

>> We agree with the reviewer's concern. To resolve the issue, in the revised ms, we have discussed this point in the discussion section.

Please see our response #4 for reviewer #2.

Given that the authors proposed that Munc18-1 and Munc13-1 would function in opening and dilation of fusion pores, is it more appropriate to use high-potassium treatment to induce calcium influx in this assay?

>> Thank you for pointing this out. We would like to mention, however, that our in vitro studies described in the ms suggest that the Munc18-1 and Munc13-1's action on pore opening and dilation does not require the presence of Ca²⁺. Once the pores are open, Munc13-1's interaction with Ca²⁺ presumably affects open state stability of the individual pores. Hence, in the current ms, we decided not to induce Ca²⁺ influx in the assay with cultured neurons. This set of experiments will be a part of a future publication.

2. It is also unclear whether SNARE complex assembly would increase upon PMA treatment. The authors need to check the colocalization between the other two SNAREs (i.e., SNAP-25 and VAMP2) and Munc18-1/Munc13-1 to further confirm their conclusions.

>> We agree with the reviewer's concern. To resolve the issue, in the revised ms, we have now included colocalization data of Munc13-1/Munc18-1 with syx2 and syntaxin. Please note – the motivation to do neuronal work was not to check the SNARE complex formation as a function of PMA treatment. Our aim was to check if PMA treatment allows SNAREs to differentially populate the presynapse, with the Munc proteins. This would be a very early stage of membrane fusion at the presynapse.

Please see our response #2 for reviewer #1.

3. Under normal circumstances, SNAREs (i.e., Syx1, SNAP-25, and VAMP2) and SNARE-chaperones exert their functions mainly in presynaptic regions. These proteins in soma and non-dendritic regions may not in their 'working state' and may be the freshly synthesized fractions. I am not quite sure how the authors quantified the colocalization. My suggestion is that adding an additional channel of presynaptic marker (e.g., synapsin) to indicate presynaptic regions, analyzing the MCC between SNAREs and Munc18-1/Munc13-1 of presynaptic marker-positive region.

>> We appreciate the reviewer for this suggestion. We ignored the cell body and only analyzed the neurites in the current version of the ms. In the revised ms, we have used the presynaptic marker synapsin in an additional channel and analyzed MCC between SNAREs and Munc proteins. The revised Fig. and the associated text are provided above (response #2, reviewer #3) for the ready reference of the reviewer.

Meanwhile, the effects of PMA treatment in individual proteins were vague. For instance, Munc13-1 seems to transit into more smeared distribution along the neurites upon PMA treatment (Fig. 1a central panel and Fig. S1a), which contradicts with previous observations that Munc13-1 translocate to membrane structures (Duncan et al., 1999, PMID: 10488064; Houy et al., 2022, PMID: 36214779). The authors need to comment on the issue and analyze the localization of individual proteins before and after PMA treatment.

>> The redistribution of Munc13-1 puncta after PMA treatment can originate due to phorbol ester dependent translocation of cytosolic Munc13-1 randomly into the membrane, which indeed supports all the previous papers that the reviewer has pointed out.

To resolve the issue, in the revised ms, 1) we have provided high-resolution images to show the effect of PMA on Munc protein's distribution; 2) we have discussed the results extensively in the discussion section. Please see above (response #2, reviewer #3), the high-resolution image to show PMA's effect.

4. It is interesting that colocalization of Munc18-1-SNAP-25 could also respond to PMA treatment although it does not contain any lipid binding domains. However, the authors did not make enough interpretations and discussions.

>> Thank you for pointing this out. In the revised ms, now we have included this in the discussion section. Please see our response to reviewer 2, response 4, above.

A quick look at previous literatures, it has been reported that phosphorylation of Munc18-1 by PKC regulates exocytosis kinetics (Barclay et al., 2003, PMID: 12519779; Craig et al., 2003, PMID: 12950453), but the phosphorylation may affect Syx1 binding of Munc18-1. Another recent literature reported that phorbol esters and PTP enhance synaptic transmission primarily by mechanisms that are independent of PKC phosphorylation of Munc18-1 (Wang et al., 2021, PMID: 34290081). These results again indicate that the effect of PMA treatment is complicated (see point#1).

>> Thank you for pointing this out. We have included these references in the discussion section of the revised ms.

5. The authors interpreted that DAG instead of PIP2 caused the inhibitory action on fusion pore. Despite they analyzed the individual effect of PIP2 by nanodisc-bilayer electrophysiology, no direct data shows that DAG inhibits opening and dilation of fusion pore. This is an important issue since the data below were all based on this precondition.

>> We agree with the reviewer. In the revised ms, we have now provided ND-BLM data for PE/PC/PS/DAG (without PIP2). Now, we have also included data for DAG titration in the BLM. Together, the results described in the revised ms demonstrate the inhibitory role of DAG in fusion pore opening and dilation.

Please see our response #6 for reviewer #2.

Also, the authors should analyze the individual effect of DAG and PIP2 in bulk liposome-nanodisc fusion assays.

>> Because in the revised ms we have clearly demonstrated the inhibitory effect of DAG on fusion pore opening and dilation, in the single pore resolution, we have not done the bulk fusion assays to demonstrate the same thing. However, if the reviewer thinks it is deemed necessary, we will incorporate the data.

6. Another important issue: the mechanism of the inhibitory action on fusion pore by DAG need to be illustrated or proposed. Does DAG cause fusion-incompatible membrane curvature (as DAG possesses rather small head group)? Or DAG 'cages' one or more SNAREs? Meanwhile, how do Munc13-1 and/or Munc18-1 relieve the inhibitory effect?

>> We appreciate the reviewer for this comment. In the revised ms, we have now included the data with PA, as a part of the BLM lipids. Because both the DAG and PA showed similar inhibitory effects on the fusion pore properties, we presume that the pore inhibitory action, in this case, is contributed by the membrane curvature.

In the revised manuscript we have now incorporated a series of new data and proposed a model of how Munc13-1/Munc18-1 relieves the inhibitory effect.

As a reader, I could only acknowledge that Munc13-1 binds to DAG through its C1 domain according to the manuscript, however, this is probably not the case for the inhibitory-stimulatory mechanism. To better reconcile the inhibitory effect, the authors need to titrate the reconstitution% of DAG in bulk liposome-nanodisc fusion and nanodisc-bilayer electrophysiology assays.

>> To better understand the inhibitory-stimulatory mechanism of membrane fusion, mediated by DAG and Munc proteins, we performed a series of additional experiments, now included in the revised version of the ms. Because the ND-BLM assay has higher resolution, we did not perform DAG titration using bulk fusion assay. If it is deemed necessary, we will include that.

For the inhibitory mechanism, the authors may use other lipids with small headgroup (e.g., PA, PE, ..etc.) to replace DAG and analyze the effect of fusion pore formation.

>> We thank the reviewer for this suggestion. We have used PA in the revised version of the ms. Please see the data and relevant text from the revised ms, provided above for ready reference.

"When we replaced DAG with another small head group containing lipid PA (phosphatidic acid), we observed the similar pore phenotype as observed in presence of DAG (Fig. S5). The fusion pore inhibitory action was presumably the contribution of membrane lipid constituent that induces local negative curvature in the membrane."

Fig. S5

a. Membrane lipids: PS/PE/PC/PA

Fig. S5: a,b, Representative trace (a) and the corresponding current histograms (b) of ND_{5s} pore, before and after Munc13-1 addition. The BLM lipid composition is mentioned. The close (C) and open (O) states are indicated. n = 3 independent BLM recordings.

7. Munc18-1 and Munc13-1 are reported to cooperatively chaperone SNARE assembly. The authors also agree with the notion. I could understand that the authors tried to dissect the individual function of Munc18-1 and Munc13-1, however, they did not show the effect with both Munc18-1 and Munc13-1 in most of the experiments. For the bulk liposome-nanodisc fusion assay in Fig. 2b and SNARE assembly assays in Fig. 5a-e, it is recommended to include an experiment that containing both Munc18-1 and Munc13-1.

>> We appreciate the reviewer's comment. We have now performed added both the Munc13-1 and Munc18-1 in ND-BLM and SNARE complex assembly assays. Because ND-BLM recordings yield high-resolution data, we have incorporated that in our revised ms. The ensemble SNARE complex assembly data are provided here for the reference of the reviewer.

Figure legend: Representative immunoblots showing trans SNARE complexes during t-SNARE liposomes' fusion with ND_{5s} with increasing [Munc18-1] keeping [Munc13-1] constant (left) and with increasing [Munc18-1] keeping [Munc13-1] constant (right). Scatter plots fitted with modified hill equation showing fold change in band intensity for trans-SNARE complexes formed as [Munc13-1] (bottom, left) and [Munc18-1] (bottom, right) increases while keeping the other Munc constant respectively.

Fig. 6: i, Representative trace of single ND5s pore triggered open by both Munc13-1 and Munc18-1. Closed (C) and multiple open states (O₁/O₂) of the individual pores are indicated with the respective currents. j, Current histogram of the pore shown in i. Closed (C) and open (O₁) and (O₂) states are indicated by red and blue arrows respectively. k, Scatter plots show pooled data for pore conductance comparing pores triggered open by either Munc13-1 or Munc18-1 or both. l, CDF of closed dwell times comparing indicated experimental conditions.

8. Similar issue: the sequential actions of Munc18-1 and Munc13-1 on fusion pores were analyzed by the authors, however, they did not show the effect with both Munc18-1 and Munc13-1 simultaneously.

>> Please see our response #7 above for reviewer #3.

9. The pore sizes (or the conductance) were not shown for the experiments in Fig. 2c. I think it is important to compare the pore sizes formed by only SNAREs (with or without DAG/PIP2) and SNAREs+Munc18-1/Munc13-1.

>> We appreciate the reviewer's comment. In the revised ms, the pore conductances (which is a measure of pore size) are compared for the pores formed by only SNAREs (in the absence or presence of DAG/PIP2) and SNAREs+Munc18-1/Munc13-1.

Please see the response #10 of reviewer #2.

10. The authors need to be cautious about the bulk liposome-nanodisc fusion data. Munc13-1 is reported to directly bind to lipids and membrane structures. The C2C domain of Munc13-1 associates with membrane through hydrophobic insertion with no lipid preference (Quade et al., 2019, PMID: 30816091), which means that in vitro membrane fusion assays in the presence of Munc13-1 may be perturbed by content leaking. The authors should perform control assays for bulk liposome-nanodisc fusion assays and nanodisc-bilayer electrophysiology assays to confirm the effects were genuinely caused membrane fusion, rather than content leakage (e.g., use NDS0 or NDLO).

>> We appreciate the reviewer's comment. Please note that, a ~0.5 μM Munc13-1 addition did not alter the baseline current of the ND-BLM recording, hence not involved in the leakage of the bilayer membrane. The data is shown below for the ready reference.

11. It is strange that Munc13-1H567K mutant displayed markedly larger pores compared to Munc13-1WT. The authors concluded that 'These results indicated that the Munc13-1, while rescuing the pores from their DAG-induced inhibitory state, keeps the pores in a narrow-constricted conformation. Pores are kinetically stable in open states despite such structural constraints (Figs. 3c-h). This structural arrest is mediated through Munc13-1's interaction with DAG in the membrane lipids.' The interpretations were vague and incomprehensible. I suppose that the authors meant that the 'narrow pore' is caused by Munc13-1 binding to DAG. However, this is literally not a 'mechanism' but an experimental observation. If the authors believed that the binding of Munc13-1 to DAG instead of the membrane (I mean, not specific to DAG) is responsible for the effect, they should confirm whether Munc13-1H567K would still attach to membrane, since Munc13-1 C2B could also interact with PIP2. Alternatively, they may construct a chimeric Munc13-1H567K with an N-terminal membrane binding module that does not bind to DAG and check the function of the chimera in pore formation.

>> We thank the reviewer for pointing this out. In the revised ms, we have included data to show Munc13-1H567K's membrane binding ability using a co-flotation assay. The relevant data is provided below for ready reference of the reviewer.

Fig. S6

Fig. S6: a,b, Cartoon showing coflotation assay (a, left) where protein interacting with liposomes will be in the top layer (T) and unbound proteins would be in bottom layer (B). Representative immunoblots (a, right) and the quantification (b) of co-flotation assay showing the interaction of WT and mutants of Munc13-1 (as indicated) with liposomes. Molecular weight in kDa of nearest protein ladder marker is shown in all the immunoblots. Antibody used has been mentioned. N = 3 independent experiments. Data represented as mean±SEM.

12. It is also incomprehensible that over-dose Munc18-1 would affect pore formation. The authors still did not present a mechanism and potentially reasonable explanation (and they did not show the conductance comparison between 5 and 10 uM Munc18-1). Does over-dose Munc18-1 impair SNARE assembly? According to Fig. 5a, left panel, it seems not. Does over-dose Munc18-1 affect the lipid dynamics? The authors need to carry out additional experiments to further interpret the results, otherwise the data using different [Munc18-1] make no sense.

>> Thank you for pointing this out. In the revised ms, we have now included a series of additional experiments with Munc18-1 mutants and titration of Munc18-1 (WT). These experiments helped us to reach a detailed mechanistic insight into how this Munc protein organizes the functional fusion pore assembly, supported by the existing literature.

>> Please see our response #7 above for reviewer #1;

>> Please see our response #7 above for reviewer #2; and

>> Please see our response #11 above for reviewer #2.

Incidentally, does over-dose Munc13-1 affect pore formation as well?

>> We have titrated few [Munc13-1] and found that the concentration does not affect the pore formation by Munc13-1. The raw traces for two additional [Munc13-1] are shown below as a response to the reviewer.

Figure legend: a, Representative trace of single ND5_s pore triggered open by different [Munc13-1]. BLM lipids used is mentioned above the trace. Two epochs (beginning and end) of the trace are shown. Closed (C) and open states (O), (O₁) and (O₂) of the individual pores are indicated with the respective currents. f, b, Current histogram of the ND5_s pore formed in the presence of indicated [Munc13-1]. Closed (C) and open (O, O₁ and O₂) states are indicated by red and blue arrows respectively.

13. Given that the Hill coefficient (n) of ND5S is larger than ND3S, the authors did not present the parameter k for comparison (also lacking representative traces, see minor concerns). It is important to judge the cooperative effect with full fitting parameters. Similar issues were found in Fig.5c.

>> Thank you for pointing this out. In the revised ms, we have now included the full fitting parameters, as the reviewer suggested. We have also included the representative traces in the supplementary. The relevant text from the revised ms is provided below.

14. I would doubt about the interpretation of T1/2. The authors wrote that (e.g. Fig. 2b) '...indicate fold change in time at which % glutamate released is half of the maximum (T1/2)'. If the authors meant to compare the thermodynamical efficiencies among different groups, the T1/2 value may overlap with fold change in maximum percent (%) and be meaningless. If the authors meant to compare the kinetic

efficiencies, there are more appropriate ways, e.g., fit the scatters with exponential association functions and use kinetic parameters to interpret the data.

>> Thank you for pointing this out. In the revised ms, we have plotted the half-maximum of % glu released, instead of T1/2. Because we have performed in-depth kinetic measurements in single molecule resolution (using ND-BLM system), we haven't done the kinetic fitting on this bulk fusion assay. However, if it is deemed necessary, we will provide that.

Additionally, in Fig.5e, it is clear that the 3 traces displayed multiple exponential kinetics, it is hard to conclude whether Munc18-1/Munc13-1 accelerate C-terminal assembly kinetics or simply facilitate SNARE pairing. The authors are obligated to check the above issues and carefully interpret the data since these may be important to the central conclusion.

>> Thank you for pointing this out. Here we agree with the reviewer. From the bulk C-terminal FRET data, we cannot conclude whether Munc18-1/Munc13-1 accelerate C-terminal assembly kinetics or simply facilitate SNARE pairing. Hence, we have incorporated that interpretation in explaining our model in the revised ms. The revised text is provided below for the ready reference.

"This, however, does not exclude the possibility that the observed results were due to enhanced SNARE pairing by Munc proteins."

15. A critical concern: in Fig.3, the authors concluded that 'Munc13-1 stimulated the pore to interconvert between two different open states, a smaller conducting O1 and a larger conducting dilated O2 state.' and 'These pores, however, were restricted to a low-conducting narrow conformation. Munc18-1 also rescued the pores from their DAG mediated inhibitory state and the pores were significantly larger in size in comparison to those triggered open by Munc13-1.' However, in Fig. 6e and g, the pore sizes stimulated by Munc13-1 before Munc18-1 addition were literally larger than that shown in Fig.3, and Munc18-1 addition did not induce any change in sizes. This seems to be an apparent contradiction. Please check and comment on the issue.

>> Thanks for pointing this out. In all the figures the current histograms are shown only for the representative raw traces, shown in the figures.

Please note – we have plotted conductance of all the Munc13-1 pores in Figure 3G, which does show trace-to-trace variation in the currents. That, however, does not change the central conclusions from this figure.

16. Also, I noticed that the data of Fig.3g (green scatters) and Fig.6g (green scatters) are identical if I am not mistaken. In my view, the data in Fig.6e is independent to that in Fig.3a, therefore, the statistical data should be different. Please carefully check the issue and present reasonable explanations.

>> Yes, the reviewer is right. The data of Fig.3g (green scatters) and Fig.6g (green scatters) are identical. The starting points of both the experiments, in Fig.6e and 3a are the same, in both cases Munc13-1(WT) was added at the beginning of ND5 pore opening. In a subset of 3a experiments, Munc18-1 was subsequently added after a considerable amount of recording. Hence we combined all the Munc13-1(WT) only data together in both the Figures 3g and 6g. If it is deemed necessary, we can certainly modify the Fig. 6g. Nevertheless, that will not change the central conclusion from this figure.

17. In Fig.S7a, the authors concluded that 'The Ca²⁺ addition downstream to Munc13-1 had no appreciable effect on stabilizing the pore open states.' According to the CDF curve, Ca²⁺ addition after Munc13-1 even decrease the open state dynamic of the fusion pore. In contrast, in Fig.S7b, the authors concluded that 'The Ca²⁺ addition downstream to Munc18-1 shifted the multi exponential kinetics of pore opening to largely single exponential (Fig. S7b), suggesting a role of Ca²⁺ in stimulated fusion pore opening.' Judging from the data, Ca²⁺ addition after Munc18-1 also decrease the open state dynamic of the fusion pore. Theoretically, if I am not mistaken, nanodisc-bilayer fusion would never produce a post-fusion state as liposome-bilayer fusion. Thus, it is unlikely that Ca²⁺ addition leads to a post-fusion state. Then, why Ca²⁺ addition decrease the open state dynamic of the fusion pore? Munc13-1 C2B domain is the only one that could response to calcium ions in this system, is Munc13-1 C2B responsible for the effect? If so, what is the underlying mechanism? Would disruption of calcium-binding sites of C2B rescue the effect? The authors also interpreted that the 'largely single exponential kinetics' suggests 'a role of Ca²⁺ in stimulated fusion pore opening', which puzzles me a lot. Could the 'kinetics' of close dwell time CDF trace be used for interpreting pore opening/dilation kinetics?

>> Again, we agree with the reviewer that the ND-BLM assay, under the current experimental condition, would not produce a post-fusion state. The role of Ca²⁺ in Munc13-1-mediated fusion pore assembly is more complicated than what we expected. Hence, from the current version of the manuscript, we have omitted that supplementary Figure, for a future submission. This does not change the central conclusion and results of our current ms, as all other studies were done in the presence of BAPTA.

18. I do not know why the authors explored the effect of Ca²⁺ here in the absence of the calcium sensor Synaptotagmin-1. Given that Munc13-1 C2B domain is proposed to be a potential calcium sensor (Xu et al., 2017, PMID: 28177287), physiological fusion pore opening and dilation may be dominated by Synaptotagmin-1. Please comment on this.

>> Because we have omitted the Ca²⁺ data from the current version of the ms, we have not considered including the synaptotagmin-1 here. That is a separate study, which we are currently pursuing in the lab for future submission.

19. The major conclusion that Munc18-1 and Munc13-1 function in fusion pore opening and dynamics largely relies on the system build on DAG-inhibitory effect. Unfortunately, the underlying mechanism of DAG-inhibitory effect was not fully addressed by the authors. Therefore, additional data were needed to confirm the effect of Munc18-1 and Munc13-1 in fusion pore opening and dilation. Given that the fusion pores were efficiently and stably formed without DAG and PIP₂, the authors may utilize a C-terminal truncated SNAP-25 ($\Delta 9$, BoNT/A cleaved product) to clamp the final fusion (Lu et al., 2015, PMID: 26157630; Li et al., 2018, PMID: 29485200). In fact, similar function (i.e., driving the terminal stage of SNARE-dependent membrane fusion) was found in HOPS-regulated vacuolar SNAREs-mediated fusion (Song et al., 2021, PMID: 33944780; D'Agostino et al., 2017, PMID: 29088698). The authors need to discuss these analogous results in the discussion section.

>> Thank you for pointing this. In the revised ms, our described model for Munc13-1/Munc18-1 mediated fusion pore opening/dilation is built on a series of new experiments, unrelated to DAG-inhibitory effect. Additionally, we have performed experiments to address the DAG mediated inhibitory effects.

We have included the above references in the discussion section of the revised ms.

Please see our response 6, for reviewer 3 above.

20. Finally, I would express my concern about the title. The title was partially overstated and not clear. On one hand, it feels like that the authors analyzed multiple SNARE chaperones more than Munc18-1 and Munc13-1, but in fact, they mainly focused on Munc18-1 and Munc13-1; On the other hand, the authors did not fully resolve the 'activity' of Munc18-1 and Munc13-1 in the fusion pore formation during SNARE-mediated membrane fusion, thus, to say '...activity determines...' should be inappropriate. The authors may utilize words that reflect their ground truth data, instead of making the title vague.

>> We have modified the current title of the ms. The revised title now reads "Differential SNARE chaperoning by Munc13-1 and Munc18-1 dictates fusion pores fate at the release site".

Minor concerns:

1. The fractions (M1 and M2) of Manders' colocalization coefficient are not very friendly to the readers and may cause potential misunderstandings. The authors are suggested to use 'A to B' and 'B to A' to interpret the colocalization data.

>> We have modified the nomenclature of MCC as suggested by the reviewer.

2. The CDFs of closed state dwell times were not shown for different concentration of Munc18-1 in Fig. 4. Please check.

>> We have revised the Fig. 4.

3. Some of the representative traces were missing (e.g., Fig.5b, the traces for ND3S; Fig.5c, the trace for PS/PE/PC; Fig.5g, all; Fig.S2b, all; Fig.S5c, the traces for ND25L; Fig.S7, all). It is important to present raw representative traces to meet the requirement of transparency process.

>> We appreciate the reviewer's comment. We have now included all the raw traces in the revised ms. The relevant raw traces are provided below for ready reference.

Fig. S8: e, Representative immunoblot showing trans SNARE complexes during t-SNARE liposomes' fusion with ND3_s with increasing [Munc13-1] (top) and [Munc18-1] (bottom). Scatter plots fitted with modified hill equation showing fold change in band intensity for trans-SNARE complexes formed as [Munc13-1] (bottom, left) and [Munc18-1] (bottom, right) increases for ND3_s and ND5_s. The membrane lipid composition for the t-SNARE liposomes was PS/PE/PC/PIP₂/DAG. f, Representative immunoblot showing trans SNARE complexes during t-SNARE liposomes' fusion with ND5_s with increasing [Munc13-1] (top) and [Munc18-1] (bottom), using membrane lipid composition: PS/PE/PC. Scatter plots fitted as in (e) for increasing [Munc13-1] (bottom, left) and [Munc18-1] (bottom, right) for different membrane lipid compositions as indicated. Molecular weight in kDa of nearest protein ladder marker is shown in all the immunoblots.

Fig. S9: c, Representative time course of glutamate release through the fusion pores during fusion reaction as ND3_s, ND5_s and ND7_s react with glutamate entrapped t-SNARE liposomes, in separate set of experiments, in the absence (left) and presence of SNARE chaperones: Munc13-1 (middle) and Munc18-1 (right).

4. In pg.8, texts cited Fig.5h, however, Fig.5h could not be found in the figure c

>> We apologies for the confusion. We have corrected the mistake in the revised ms.

REVIEWER COMMENTS

Reviewer #2 (Remarks to the Author):

My Major concerns have been well addressed. Significant improvement was made in the revised manuscript. There are still some minor inconsistencies that need attention.

1. To explain the lower fraction-close of 5uM than 10uM Munc18-1, the author stated that “a multi-exponential fusion pore closure driven by Munc18-1, yielding slow and fast components for pore closure”. This non-linear effect of Munc18-1 is interesting. Fast component increased while slow component decreased with the concentration increase of Munc18. In Fig. 4d, the 2.5 and 5 uM concentrations of Munc18 were missing for two components in Fig. 4d while their CDF are plotted in Fig. S7. Any reason to delete these two points?
2. In Fig. 5a, the gel of 0.5 uM Munc13-1 was shown without intensity measurement in the accompanying intensity plot.
3. The shapes (round, square or triangle) of the label (e.g., Fig. 2a middle and 2a right; Fig. 2b; Fig. 5d and Fig. 5e.) should be consistent to avoid confusion if the shape doesn't mean any specific condition.

Reviewer #3 (Remarks to the Author):

It is appreciated that the authors have made several improvements for the manuscript, including new data, additional discussions, and corrections to some issues, in response to the reviewers' concerns. I have also browsed through the review comments by the other two reviewers. The authors could clearly see that some of the critical concerns among the three reviewers are largely overlapped, which indicates that these concerns are pivotal issues hindering the manuscript from being published. But it is regrettable that some critical issues raised in the first reviewing stage are still not fully resolved by the authors. Related comments are listed below:

1. The usage of PMA, related experiments, and related interpretations are still incomprehensible, especially after I read the concerns raised by the other reviewers. It is not enough to supplied some eclectic discussion for the issue. As emphasized before, PMA would cause various cellular effects. This means that the colocalization between SNAREs and SNARE-chaperones may not fairly indicate physiological association. Also, according to the revised manuscript, the pore-opening/dilation ability of Munc18 and Munc13 is independent on the specific lipid DAG (in PA experiments). Therefore, it is hardly difficult to understand the data in Fig.1, especially for its relationship to the results below.
2. The supplemented ND-BLM experiments with PA have directly addressed my previous concerns. Nevertheless, the interpretations and discussions made by the authors are not convincing. On one hand, if the inhibition-stimulation of pore opening is not DAG-specific (instead, it is caused by membrane dynamics), then there is only one explanation that Munc18 and Munc13 could induce lipid rearrangements and this ability is not specific to any lipids, which detracts the significance of the manuscript; on the other hand, even if the above hypothesis is true, the authors did not present any further analysis for the hint. I'm pretty sure that the mechanism of Munc18 and Munc13 in pore-

opening/dilation is the core of the manuscript. Nevertheless, the data presented here is, to some extent, incomplete.

3. I could understand that the authors removed the Ca^{2+} -related data, since they could not present a rational explanation for the results. But one should acknowledge that physiological fusion pore formation in synaptic exocytosis is accompanied with calcium influx. It was not my intention to request the authors to remove the related data. Instead, the authors should try to conclude a potential mechanism. Although their system did not contain the canonical calcium sensor Syt1, Munc13 would also respond to calcium, let alone calcium influx may be required for fusion pore opening in physiological context. The authors seemed to avoid the important and dwell on the trivial.

4. The authors seemed to be overconfident for the previous established ND-BLM method. I must admit that it is indeed a quite precise and elegant approach to analyze fusion pore dynamics. However, it doesn't mean that this approach could override any other ensemble or single-molecule approaches in proving the same thing or mutual corroboration. I am regret that the authors rejected some requests for additional experiments.

5. The action of Munc18 and Munc13 in SNARE pairing and C-terminal assembly is not clearly elucidated by the authors. Actually, in the revised manuscript with supplemented data, I do not even know what the authors is going to conclude the mechanism of Munc18 and Munc13 in pore-opening/dilation. Induce lipid rearrangement? Facilitate SNARE C-terminal assembly? Or both? Again, I will express my opinion, the underlying mechanism of Munc18 and Munc13 in pore-opening/dilation should be the core of the manuscript, instead of briefly describing the experimental phenomenon with noneffective discussions.

REVIEWER COMMENTS

Reviewer #2 (Remarks to the Author):

My Major concerns have been well addressed. Significant improvement was made in the revised manuscript. There are still some minor inconsistencies that need attention.

1. To explain the lower fraction-close of 5uM than 10uM Munc18-1, the author stated that “a multi-exponential fusion pore closure driven by Munc18-1, yielding slow and fast components for pore closure”. This non-linear effect of Munc18-1 is interesting. Fast component increased while slow component decreased with the concentration increase of Munc18. In Fig. 4d, the 2.5 and 5 uM concentrations of Munc18 were missing for two components in Fig. 4d while their CDF are plotted in Fig. S7. Any reason to delete these two points?

>> We thank the reviewer here. The reason to delete the two points (2.5 and 5 uM) was that we could see the non-linear trend without those data points and we just wanted to keep a very high concentration of 10 uM after 1 uM to emphasize the increase of fast component and decrease of slow component.

To resolve the issue, in the revised figure, we have now incorporated the two concentrations; providing below the revised Figures for ready reference.

Figure 4.

a. Membrane lipids: PS/PE/PC/PIP₂/DAG

Fig. 4: Munc18-1 differentially alters fusion pore properties compared to Munc13-1

a, Representative trace for ND5s pore triggered open by ~ 1 μM Munc18-1. Three epochs (beginning, middle and end) of the trace are shown, membrane composition mentioned; closed (C) and open (O) states are indicated with the respective currents. **b**, Current histogram of the trace shown in **a**, closed (C) and open (O) states are marked by red and blue arrows respectively. **c**, Scatter plot shows comparison of pore conductances between Munc13-1 (~ 0.3 μM) and Munc18-1 (~ 1 μM) triggered pores. **d**, Stacked column plots show amplitudes of fast and slow kinetic components (as indicated) for pore closures, derived from

open dwell time CDFs exponential fit. These individual pores were triggered open by different [Munc18-1], as shown.

2. In Fig. 5a, the gel of 0.5 μM Munc13-1 was shown without intensity measurement in the accompanying intensity plot.

>> We again appreciate the reviewer to point this out. To resolve the issue, in the revised Figure 5a, we are now showing only those concentrations in the gel whose intensity measurements are provided below. The revised Figure is shown here for the ready reference.

Figure 5.

Fig. 5: Munc13-1 and Munc18-1 synchronize vesicular secretion by differentially organizing trans-SNARE assembly

a, Representative immunoblot showing trans SNARE complexes during t-SNARE liposomes' fusion with ND5_s with increasing [Munc13-1] (top, left) and [Munc18-1] (top, right). Scatter plots fitted with modified hill equation showing fold change in band intensity for trans-SNARE complexes formed as a function of increasing [Munc13-1] (bottom, left) and [Munc18-1] (bottom, right). The hill coefficient (n) and EC₅₀ (k) are indicated in the plot. The membrane lipid composition for the t-SNARE liposomes was PS/PE/PC/PIP₂/DAG. **b**, Scatter plot showing pooled data for hill coefficients (top panel – Munc13-1, bottom panel – Munc18-1) obtained from different trials of **a**, using ND3_s and ND5_s.

3. The shapes (round, square or triangle) of the label (e.g., Fig. 2a middle and 2a right; Fig. 2b; Fig. 5d and Fig. 5e.) should be consistent to avoid confusion if the shape doesn't mean any specific condition.

>> We appreciate the reviewer to point this out. In the revised Figures, we have now modified the shapes of the label. The revised Figures are shown here for the ready reference.

Figure 2.

Fig. 2: SNARE chaperones overcome the inhibitory effect of phorbol ester DAG during membrane fusion.

a, Left panel, illustration shows the glutamate release assay through the fusion pores as ND5s interacts with the glutamate entrapped t-SNARE liposomes. The glutamate released into reaction medium is measured by an increase in the fluorescence signal of the glutamate sensor (iGluSnFR). Middle panel, representative time course of glutamate release through the fusion pores during fusion reaction between ND5s and glutamate entrapped t-SNARE liposomes, in absence and presence of PIP₂/DAG lipids in t-SNARE liposomes. Right panel, box plots showing pooled results to indicate half maximum percent (%) of glutamate (glu) released for the indicated conditions. **b,** Left, representative time course as shown in **a,** in the absence and presence of SNARE chaperones (Munc13-1 and Munc18-1) keeping membrane lipid composition for t-SNARE liposomes (PS/PE/PC/PIP₂/DAG) and ND5s fixed. Middle, box plots showing pooled results to indicate half maximum percent (%) of glutamate (glu) released for the indicated conditions. Data are presented as mean \pm SEM from seven (for panel **a,**) and ten (for panel **b,**) independent trials under each of the conditions mentioned; three independent sets of NDs were used. Student's T test was performed to compare the two means; * $p < 0.05$, n.s. - non-significance.

Fig. 5: Munc13-1 and Munc18-1 synchronize vesicular secretion by differentially organizing trans-SNARE assembly

d, Left panel, representative time course of corrected acceptor (N terminal cysteine - Cy5) emission, as ND5^{Cy3N} reacts with t-SNARE liposome^{Cy5N} under indicated conditions. Right panel, scatter plot shows fold change in time at which FRET intensity is half of the maximum ($T_{1/2}$) for the indicated conditions. **e**, Left panel, representative time course of corrected acceptor (C terminal cysteine - Cy5) emission, as ND5^{Cy3C} reacts with t-SNARE liposome^{Cy5C} under indicated conditions. Right panel, scatter plot shows fold change in time at which FRET intensity is half of the maximum ($T_{1/2}$) for the indicated conditions.

Reviewer #3 (Remarks to the Author):

It is appreciated that the authors have made several improvements for the manuscript, including new data, additional discussions, and corrections to some issues, in response to the reviewers' concerns. I have also browsed through the review comments by the other two reviewers. The authors could clearly see that some of the critical concerns among the three reviewers are largely overlapped, which indicates that these concerns are pivotal issues hindering the manuscript from being published. But it is regrettable that some critical issues raised in the first reviewing stage are still not fully resolved by the authors. Related comments are listed below:

1. The usage of PMA, related experiments, and related interpretations are still incomprehensible, especially after I read the concerns raised by the other reviewers. It is not enough to supplied some eclectic discussion for the issue. As emphasized before, PMA would cause various cellular effects. This means that the colocalization between SNAREs and SNARE-chaperones may not fairly indicate physiological association. Also, according to the revised manuscript, the pore-opening/dilation ability of Munc18 and Munc13 is independent on the specific lipid DAG (in PA experiments). Therefore, it is hardly difficult to understand the data in Fig.1, especially for its relationship to the results below.

>> Thanks to the reviewer for pointing this out. We agree that the PMA treatment would cause various cellular effects. Previous literature suggests that the exogenous PMA treatment enhances SNARE phosphorylation (PMID: 26721188, 8662851), which has a significant impact on secretion from various cell types (PMID: 10481193, 26721188, 29101310). It was also indicated that exogenous PMA treatment enhances SNARE complex assembly in PC12 cells (PMID: 26721188). The effect of phorbol ester is debated to have PKC dependent and independent role in exocytosis. Our study aimed to corroborate the previous studies and inspect whether PMA treatment affects the co-localization of individual SNAREs with Munc13-1 and Munc18-1, irrespective of the mode of action. Our reductionist approach then followed up, to show that the phorbol ester has a direct role in inhibitory-stimulatory secretion, involving SNAREs and SNARE chaperones.

Additionally, this data suggests that SNAP-25B's cellular localization is majorly affected by the PMA treatment. This follows the previous observations that the recruitment of Munc13-1 and SNAP-25B to syntaxin1a-Munc18-1 clusters is a prerequisite for priming event (PMID: 33222163, 24835618, 12859899, 37343100).

Our ND-BLM studies (Figures 4 and 7) also indicate that the stable association of SNAP-25B with other SNAREs is mediated through Munc13-1. Please see our revised discussion (our response to reviewer#3, comment #5).

Please also see our response to reviewer#3, comment#2, which shows the specificity of Munc13-1 to DAG.

However, if the reviewer still thinks it is deemed necessary, this data can be moved to supplementary.

In the revised ms, the updated discussion section now reads:

"This set of experiments, however, did not provide direct evidence of how these Munc proteins organize fusion pores at the release site, during regulated secretion."

2. The supplemented ND-BLM experiments with PA have directly addressed my previous concerns. Nevertheless, the interpretations and discussions made by the authors are not convincing. On one hand, if the inhibition-stimulation of pore opening is not DAG-specific (instead, it is caused by membrane dynamics), then there is only one explanation that Munc18 and Munc13 could induce lipid rearrangements and this ability is not specific to any lipids, which detracts the significance of the manuscript; on the other hand, even if the above hypothesis is true, the authors did not present any further analysis for the hint. I'm pretty sure that the mechanism of Munc18 and Munc13 in pore-opening/dilation is the core of the manuscript. Nevertheless, the data presented here is, to some extent, incomplete.

>> We thank the reviewer to point this out. To resolve the lipid specificity issue, we have now compared the Munc13-1 mediated pore-opening kinetics, in the presence of DAG and PA in BLM. The motivation for this set of experiments was the presence of DAG binding domain in Munc13-1.

The revised Figure S5 now shows that, while Munc13-1 can overcome the inhibitory effect of both the DAG and PA, the probability of appearance of long closures was more in the presence of PA than in the presence of DAG. This set of analysis suggests that the inhibitory stimulatory mechanism, is, indeed DAG specific. Additionally, please note that our detailed kinetic analysis, shown in table S2 and Figure S8c indicates that Munc18-1 alone is not efficient in kinetically overcoming the DAG mediated inhibitory effect; while Munc13-1 (either alone or in the presence of Munc18-1) can kinetically overcome the DAG mediated inhibitory effect significantly more, further emphasizing the specific role of Munc13-1/DAG in the inhibitory stimulatory mechanism.

Also, we believe that Munc13-1's insertion into the DAG-containing membrane is crucial for its pore opening activity. The Munc13-1(H567K) mutation, which mimics the DAG-bound activated state of Munc13-1 (PMID: 17267576) was not efficient in triggering stable pore opening, further suggesting that membrane-bound Munc13-1's interaction with DAG is important in regulating pore properties.

The revised figures and the text are shown below for ready reference.

Fig. S5

a. Membrane lipids: PS/PE/PC/PA

Fig. S5: Effect of Phosphatidic acid (PA) in ND5s pores

a,b, Representative trace (a) and the corresponding current histograms (b) of ND5s pore, before and after Munc13-1 addition. The BLM lipid composition is mentioned. The close (C) and open (O) states are indicated. N = 3 independent BLM recordings. **c**, Cumulative distribution functions (CDF) of closed dwell times for each experimental condition as indicated.

The revised result now reads:

“When we replaced DAG with PA, Munc13-1 could overcome the inhibitory effect of PA (Fig. S5). We compared the probabilities of the appearance of long closures under these conditions by analysing the closed state dwell time CDFs (Fig. S5); the mean closed-state dwell times ($\langle t_{c-obs} \rangle$) for DAG and PA were 0.54(± 0.17) s and 1.1 (± 0.3) s, respectively. The probability of appearance of long closures was more in the presence of PA than in the presence of DAG (Fig. S5). This indicates a stabilizing function of Munc13-1/DAG interaction in pore opening and suggests that the inhibitory stimulatory mechanism, is, indeed DAG specific. Overall, Munc13-1’s interaction with the membrane lipids and SNAREs helps recover the pores from their inhibitory states.”

3. I could understand that the authors removed the Ca²⁺-related data, since they could not present a rational explanation for the results. But one should acknowledge that physiological fusion pore formation in synaptic exocytosis is accompanied with calcium influx. It was not my intention to request the authors to remove the related data. Instead, the authors should try to conclude a potential mechanism. Although their system did not contain the canonical calcium sensor Syt1, Munc13 would also respond to calcium, let alone calcium influx may be required for fusion pore opening in physiological context. The authors seemed to avoid the important and dwell on the trivial.

>> We again thank the reviewer for the suggestion. To resolve the issue, in the revised manuscript, we have now incorporated the Ca²⁺ related data, which indicates that, Ca²⁺-Munc13-1 induces significant dilation of the individual pores, as evidenced by the pore conductance, while those pores showed kinetic destabilization in their open states, as indicated by the closed state CDF analysis. This indicates that the secondary messengers like, Ca²⁺ and DAG, can affect the function of Munc13-1 in regulating fusion pore properties. The revised Figure S7 and the text are provided for ready reference.

We would like to mention, however, that the extensive study in the presence of Ca²⁺ will be published elsewhere with various experimental conditions.

Fig. S7

Fig. S7: Effect of Ca²⁺ in Munc13-1 triggered pores

a, Cumulative distribution functions (CDF) of closed dwell times for each experimental condition as indicated. N = 3 independent BLM recordings, for (Munc13-1 + Ca²⁺) condition. **b**, Scatter plots show comparison of pore conductance between apo-/Munc13-1 and Ca²⁺/Munc13-1 triggered pores, as indicated. Students T test was performed to compare the two means; * $p < 0.05$.

Text:

“Because this Munc protein possess Ca²⁺ binding domain, we tested the effect of Ca²⁺ on Munc13-1 triggered open pores. Interestingly, a 500 μ M Ca²⁺ significantly enhanced the pore conductance, while those pores showed kinetic destabilization in their open states, as indicated by the closed state CDF analysis (Fig. S7). The same amount of Ca²⁺ in absence of regulatory protein did not alter pore properties, observed previously⁴³. This indicates that the secondary messengers like, Ca²⁺ and DAG, can affect the function of Munc13-1 in regulating fusion pore properties.”

4. The authors seemed to be overconfident for the previous established ND-BLM method. I must admit that it is indeed a quite precise and elegant approach to analyze fusion pore dynamics. However, it doesn't mean that this approach could override any other ensemble or single-molecule approaches in proving the same thing or mutual corroboration. I am regret that the authors rejected some requests for additional experiments.

>> We apologize for not incorporating the suggested ensemble assays by this reviewer. In the revised manuscript, now we have included ensemble lipid mixing experiments (well-accepted ensemble assay for membrane fusion, PMID: 23598444, 9529252, 22422984), which further confirms our conclusions drawn from the single molecule ND-BLM experiments. The new results and the relevant text are shown below for ready reference:

Text:

"The direct action of Munc13-1 or Munc18-1 on membrane lipids did not contribute to the enhanced content release, as none of them showed considerable SRB (sulforhodamine B) release when SRB entrapped t-SNARE liposomes were separately treated with these Munc proteins (Fig. S2b)."

"The ensemble lipid mixing assay, demonstrated previously^{11,18,52} also indicated that the inclusion of DAG in the PE,PC, PS containing membrane is responsible for reducing the membrane fusion efficiency (Fig. S3d)."

"Similarly, lipid mixing percent also reduced gradually as we increased the DAG content of the acceptor membrane (Fig. S4c)."

Fig. S2: BAPTA titration and leakage assay.

b, Representative plot for leakage assay shows percent SRB released from t-SNARE liposomes in the absence and presence of SNARE chaperones and during membrane fusion in the presence of ND5s. Inset (right) bar plot with SEM indicating average percent of SRB released at 3600s for different conditions.

Fig. S3: Effect of PIP2 on ND5s pore

d, Scatter plot showing time course of averaged percentage lipid mixed during fusion reaction between ND5s and t-SNARE liposomes containing PIP2 and DAG lipids as indicated. Light colour area under the curve indicates SEM for each time point. N=4 independent trials were done for each condition.

Fig. S4: Effect of DAG titration on membrane fusion.

c, Scatter plot showing time course of averaged percentage lipid mixed during fusion reaction between ND5s and t-SNARE liposomes containing membrane lipids as PE/PS/PC and various [DAG], as indicated. Light colour area under the curve indicates SEM for each time point. N=4 independent trials were done for each condition.

5. The action of Munc18 and Munc13 in SNARE pairing and C-terminal assembly is not clearly elucidated by the authors. Actually, in the revised manuscript with supplemented data, I do not even know what the authors is going to conclude the mechanism of Munc18 and Munc13 in pore-opening/dilation. Induce lipid rearrangement? Facilitate SNARE C-terminal assembly? Or both? Again, I will express my opinion, the underlying mechanism of Munc18 and Munc13 in pore-opening/dilation should be the core of the manuscript, instead of briefly describing the experimental phenomenon with noneffective discussions.

>> We apologize for not being clear in writing the mechanism.

In the updated manuscript, we have included explanations of how Munc18-1 and Munc13-1 act in SNARE pairing and C-terminal assembly. We emphasize how these events would trigger the fusion pores to open in structurally large and kinetically stable open states. We have also incorporated the specific role of Munc13-1-DAG interaction in regulating pore properties.

The key interpretations of this study are –

- a)** The localized presence of DAG in the membrane can serve as the membrane fusion clamp, through its ability to alter membrane curvature.
- b)** Munc13-1 and Munc18-1 can differentially control the release probabilities by differently altering the fusion pore assemblies at the release sites.
- c)** The Munc13-1 cooperatively acts to cluster multiple SNARE complexes, which has ramifications in pore properties.
- d)** Munc18-1 is less efficient in kinetically stabilizing the pore open states, presumably due to its action in displacing SNAP-25B.
- e)** The enhanced C-terminal assembly of syntaxin1a and syb2 in the presence of both the Munc proteins indicate their contribution in kinetically overcoming the energetic barrier at the onset of membrane fusion.
- f)** During the fusion pore opening, the co-existence of both the Munc proteins is required to stabilize the trans-SNARE complex. This allows fusion pores to open in structurally large and kinetically stable open states.

The revised discussion is provided below for ready reference:

"Spatiotemporal dynamics of membrane fusion control the amount of chemical messengers released from cells. The SNAREs serve as the minimal machinery for fusion^{10,62} and catalyze the fusion pore formation, through which chemical messengers escape in a timely and precise manner. The regulated fusion pore assembly is the key to synchronizing overall human body function, which gets disrupted under disease conditions¹. Several regulatory proteins either act on SNAREs and/or on membrane lipids to directly control membrane fusion²¹. It is unclear how these regulators organize the SNAREs and regulate the functional fusion pore assembly. Here we have investigated the role of SNARE chaperones Munc13-1 and Munc18-1 in arranging fusion pores at the release site.

The membrane association of Munc13-1 was reported to be DAG-dependent⁵³. Here we first investigated phorbol ester's effect in localization of SNARE chaperones with the individual SNAREs in rat cortical neurons. Our results indicated that SNAP-25B is recruited at the presynapse by the phorbol ester binding protein Munc13-1 (Fig. 7). This event presumably brings SNAP-25B adjacent to Munc18-1, which is already present in the syntaxin clusters at the release site (Fig. 7)^{47-49,63}. Hence, the exogenous PMA treatment enhanced localization of Munc18-1 with SNAP-25B, although Munc18-1 does not contain any phorbol ester binding domain. The PMA treatment in our experiments might also activate PKC (Protein Kinase C) pathways⁶⁴, which has been shown to phosphorylate Munc18-1^{65,66} which may not affect synaptic transmission⁶⁷. This set of experiments, however, did not provide direct evidence of how these Munc proteins organize fusion pores at the release site, during regulated secretion.

To understand the role of Munc13-1 and Munc18-1 in fusion pore assembly, we traced glu release through the pores in the reconstituted system and serendipitously discovered the inhibitory role of phorbol ester DAG in glutamate efflux. The presence of DAG as a membrane lipid constituent significantly reduced the rate of individual fusion pores' opening, yielding long closures (Fig.2). Hence, the localized presence of DAG in the membrane can serve as the membrane fusion clamp. Because we could functionally replace DAG with PA (Fig.S5), presumably, the inhibitory action of DAG could be a consequence of its ability to alter membrane curvature. Interestingly, SNARE chaperones Munc13-1 and Munc18-1 assisted in overcoming this inhibitory effect and stimulated the glu release during membrane fusion.

Munc13-1 affects both the size and kinetic properties of individual nascent fusion pores. It stimulates pore opening through its specific interaction with DAG. Munc13-1 translocates to the plasma membrane via its interaction with DAG, which is a crucial step to stably open the fusion pores, but restricts the pores in structurally small conformation. This action of Munc13-1 is dependent on its ability to cooperatively cluster multiple SNARE complexes at the release site, which we observed is also DAG dependent. This SNARE chaperone altered the required number of *cd-syb2* molecules to titrate out the dynamic trans-SNARE complexes. Because an increasing number of SNARE complexes at the release site increase the probability of cargo release through the fusion pores^{13,18,20}, it was unknown how this heterogeneous release gets attuned during sustained release activities. Munc13-1 synchronizes the release probabilities by cooperatively organizing multiple SNARE complexes at the release site. The Munc13-1 nano-assemblies were observed at the active zone which in turn can control the number of release sites⁶⁸⁻⁷⁰. The direct pore modulatory action of Munc13-1 is also dependent on its self-oligomerization^{71,72} at the release site and on its interaction with the N-terminal regulatory domain of syntaxin1a⁵⁵.

The other SNARE chaperone Munc18-1 stoichiometrically interacts with trans-SNARE complexes, and stimulates individual SNARE complexes' zippering from N- to C-terminus. This Munc protein triggered pores showed larger conductance than the Munc13-1 triggered pores, however, those pores were kinetically less stable than Munc13-1 triggered pores. The Munc18-1 mediated long closures presumably appeared due to a dynamic open/close transition of SNAREs within the fusion pore assembly. This Munc protein has been shown to disassemble t-SNARE heterodimers and stabilized the closed conformation of syntaxin1a^{41,73}. This could lead to transient SNAP-25B displacement from the SNARE complexes, hence resulting in long pore closures (Fig. 7). The Munc18-1 mutation (D326K) that showed reduced SNAP-25B displacement⁷³, significantly enhanced the open state stability of pores, with the appearance of multiple open states.

The amount of cargo released during membrane fusion is dependent on both the size and the duration of fusion pore opening, which has ramifications in differentially activating the downstream receptors. The above results indicated that Munc13-1 and Munc18-1 can differentially control the release probabilities by differently altering the fusion pore assemblies at the release sites. When both the SNARE chaperones engage at the membrane fusion site, fusion pores open in a structurally large conformation and in a kinetically stable open state. We also observed enhanced C-terminal assembly of syntaxin1a and *syb2* in presence of these Munc proteins, indicating their contribution in kinetically overcoming the energetic barrier of membrane fusion. The co-existence of these Munc proteins together allows stable association of SNAP-25B with other two SNAREs to hold the pores open at the release sites. How these properties dictate the cellular physiology under normal and pathological conditions, will be the subject of future studies."

REVIEWERS' COMMENTS

Reviewer #2 (Remarks to the Author):

The authors have satisfactorily addressed my concerns.

Reviewer #3 (Remarks to the Author):

It is praiseworthy that the authors took the concerns and criticisms to heart and made significant improvements of the manuscript. At this stage I do not have any concerns about the data and the major arguments. I would raise only some minor comments to the authors for further polishing the manuscript.

1. About the PMA-related results. I recommend the authors to transfer the PMA-related results (mainly Fig.1) to supplementary materials. In this manner, the authors may firstly present the data showing DAG inhibits SNARE-mediated liposome-ND fusion (Fig.2), which is a solid and clear observation as far as I am concerned. Nevertheless, this is just my side of the story. The authors have the choice about whether or not to do so.

2. It is recommended to show SDs or SEMs for some traces plotted with scatters (e.g., Fig.2a middle panel, Fig. 2b left panel, Fig.5d,e left panel, Fig.S2b left panel, Fig.S10c) as the authors did in, for example, Fig.S3d and Fig.S4c.

3. Should the inset index of Fig.5 be lower-cased (i.e., a, b,c,..)? Please check.

4. In pg.6, the authors may need to be cautious about the interpretation of '... is, indeed DAG specific', since the data showed that PA also displayed inhibition-stimulation behavior, although the kinetics is different to that of DAG. As far as I am concerned, a more neutral tone to interpret the data may be better.

5. Pg.10, the third paragraph, 'This Munc protein triggered pores showed larger conductance than the Munc13-1 triggered pores', replacing 'This Munc protein' with 'Munc18' would be better.

6. About the working model, the intermediates in the dashed box. According to the citation [Jiao et al., 2018, eLife (Fig.1)], there may exist two potential pathways. The authors seemed to omit the 'template complex' that contains Munc18, Munc13, Syx1, and VAMP2.

REVIEWER COMMENTS

Reviewer #2 (Remarks to the Author):

The authors have satisfactorily addressed my concerns.

>> We thank Reviewer #2 for reviewing the manuscript and providing astute comments and improving our manuscript.

Reviewer #3 (Remarks to the Author):

It is praiseworthy that the authors took the concerns and criticisms to heart and made significant improvements of the manuscript. At this stage I do not have any concerns about the data and the major arguments. I would raise only some minor comments to the authors for further polishing the manuscript.

>> We thank Reviewer #3 for appreciating our efforts in improving the manuscripts with the insightful comments.

1. About the PMA-related results. I recommend the authors to transfer the PMA-related results (mainly Fig.1) to supplementary materials. In this manner, the authors may firstly present the data showing DAG inhibits SNARE-mediated liposome-ND fusion (Fig.2), which is a solid and clear observation as far as I am concerned. Nevertheless, this is just my side of the story. The authors have the choice about whether or not to do so.

>> We really appreciate the reviewer's suggestion. Nevertheless, we have decided to leave the PMA results in Fig.1 as it is instead of moving it to the supplementary file.

2. It is recommended to show SDs or SEMs for some traces plotted with scatters (e.g., Fig.2a middle panel, Fig. 2b left panel, Fig.5d,e left panel, Fig.S2b left panel, Fig.S10c) as the authors did in, for example, Fig.S3d and Fig.S4c.

>> Thanks for the suggestion. Because we have provided comparative analysis between indicated conditions using a box/dot plot with statistics in Figs.2a,b and 5d,e; we choose to show representative traces for them.

3. Should the inset index of Fig.5 be lower-cased (i.e., a, b,c,..)? Please check.

>> We thank the reviewer for noticing this formatting error. We have modified the Fig.5 as needed.

4. In pg.6, the authors may need to be cautious about the interpretation of '... is, indeed DAG specific', since the data showed that PA also displayed inhibition-stimulation behavior, although the kinetics is different to that of DAG. As far as I am concerned, a more neutral tone to interpret the data may be better.

>> We agree with the reviewer and have modified the sentence in the manuscript: "This indicates a stabilizing function of Munc13-1/DAG interaction in pore opening and suggests that the inhibitory stimulatory mechanism, is, perhaps DAG specific."

5. Pg.10, the third paragraph, 'This Munc protein triggered pores showed larger conductance than the Munc13-1 triggered pores', replacing 'This Munc protein' with 'Munc18' would be better.

>> We have modified the sentence as per reviewer's suggestions. It now reads:

'The Munc18-1 protein triggered pores showed larger conductance than the Munc13-1 triggered....'

6. About the working model, the intermediates in the dashed box. According to the citation [Jiao et al., 2018, eLife (Fig.1)], there may exist two potential pathways. The authors seemed to omit the 'template complex' that contains Munc18, Munc13, Syx1, and VAMP2.

>> We again thank the reviewer for the comment. We agree with the reviewer that the cited paper hypothesized two potential pathways. We have modified our current Figure 7 (working model), to incorporate a template complex formed by Munc18, Syx1 and VAMP2, which does trigger fusion pore to open, but in a kinetically unstable state (possibly due to transient displacement of SNAP25). The presence of Munc13-1 confers stability to this complex and triggers stable fusion pore opening. We cannot further comment on the existence of the two pathways mentioned, from the current set of experiments.

** See Nature Portfolio's author and referees' website at www.nature.com/authors for information about policies, services and author benefits.